# AutoTSAugment: Model-Agnostic Automated Data Augmentation for Unsupervised Contrastive-based Time Series Representation Learning

## Abstract

Contrastive-based time series representation learning methods excel at extracting high-quality representations from raw time series data. The performance of these methods depends, in unsupervised settings, on the data augmentation strategy used from generating similar and dissimilar samples. Configuring an augmentation strategy composed of various transformation is a time-consuming process, typically performed by trial and error, and dependent on extensive domain knowledge. In this work, we propose Automated Time Series Augmentation (AutoTSAugment), a modular, model-agnostic time series augmentation framework for generating augmented time series that can be used within any contrastive-based time series representation learning method, and define a novel search objective that allows evaluating the quality of augmentations in unsupervised settings. The proposed framework is designed as an unsupervised AutoML framework composed of a search space that contains a diverse range of augmentation methods and a search strategy that automatically navigates the sampling of these methods. We evaluated if this model-agnostic framework can replace augmentation strategies in existing contrastive learning methods using three baseline time series representation learning methods and three search strategies (random sampling, random search, and Bayesian optimisation search) on the downstream tasks of univariate and multivariate time series classification and forecasting across 164 datasets. Our empirical results demonstrate that the methods trained within the AutoTSAugment framework achieve similar, or better, results than those obtained using manually tailored augmentation methods, eliminating the need for labour-intensive manual experimentation of augmentations. The results also demonstrate that employing random sampling within this framework achieves similar results to that of dedicated search algorithms while having up to 63.25% faster training times on average compared with other search strategies. Furthermore, we studied the effect of an adaptive search space, which recommends augmentation methods based on dataset characteristics, showing results equivalent to a fixed search space in downstream tasks while being up to 58.21% faster.

## 1 Introduction

Time series analysis forms a prominent research area with practical applications in various domains, such as healthcare (Skaf et al., 2023), where it plays an important role in various downstream tasks, including forecasting (De Gooijer & Hyndman, 2006), clustering (Ma et al., 2019), classification (Middlehurst et al., 2024; Ismail-Fawaz et al., 2025), and anomaly detection (Skaf & Horváth, 2022).

Compared with other data modalities such as images or text, time series data present greater challenges for human annotation. The increased difficulty stems from the temporal dependencies, high dimensionality, and abstract patterns inherent in time series, which are less intuitively recognisable by annotators compared with visual or linguistic features. This complicates the adoption of deep learning algorithms as they require a large amount of labelled data for training (Eldele et al., 2021). To solve this problem, unsupervised learning from unlabelled data has emerged as a viable approach

for machine learning based on time series data; in particular, contrastive learning has shown good results in multiple research areas, including time series analysis (Meng et al., 2023; Zhang et al., 2024). These contrastive-based methods learn by using positive and negative examples to generate representations that are as close as possible in the embedding space for positive examples and as far as possible for negative examples, with data augmentation being the most popular approach for creating these positive and negative examples in the time series domain (Zhang et al., 2024).

Selecting an augmentation strategy is a crucial step that can be more impactful than the network architecture (Goodfellow et al., 2016; Lemley et al., 2017) and it has been demonstrated that the use of an incorrect augmentation method can lead to inferior results (Meng et al., 2023). Common augmentation strategies for time series data include scaling, warping, permutation, and noise addition (Meng et al., 2023). Configuring the data augmentation strategy, which includes the selection of a number of transformations and their hyperparameters, is typically approached in an ad-hoc manner—by trial and error—and is limited to the imagination, experience, and effort of the researcher (Lemley et al., 2017). This manual design process impedes scaling to new applications and downstream tasks (involving time series as well as other types of data). This is due to the abundance of potential transformations with their corresponding hyperparameters (Iwana & Uchida, 2021).

To address the limitations introduced by manual design, learning augmentation strategies from data, typically by leveraging Automated Machine Learning (AutoML) techniques in supervised settings, has emerged as a new paradigm for automating the design of augmentation methods primarily in the computer vision domain (see, e.g., (Lemley et al., 2017; Cubuk et al., 2019)). This paradigm has been adapted to times series domain in supervised settings (see, e.g., (Cheung & Yeung, 2020; Jing et al., 2024)). However, one core problem remains open within the proposed strategies: Existing approaches (Lemley et al., 2017; Cubuk et al., 2019; Cheung & Yeung, 2020; Jing et al., 2024) formulate a supervised learning AutoML problem, where the augmentation methods are selected and optimised using validation accuracy and task-specific performance metrics. This limits the scope of these approaches in unsupervised settings where such metrics are not available. There have been attempts to address this by proposing randomised or deep learning-based methods that work in unsupervised settings (see. e.g., (Woo et al., 2022; Yue et al., 2022; Zheng et al., 2024)). However, existing augmentation solutions for unsupervised time series representation learning are monolithic as data augmentation is tightly coupled with the architecture; this leads to redundancies as instances are augmented each time the model is refined (see Figure 1, Left).

Given this, it would be beneficial to have a model-agnostic strategy that carries out the automatic selection of transformations or sequences of transformations with their corresponding parameters to perform data augmentation on time series in unsupervised settings without the need to manually choose based on trial and error or domain knowledge—a topic that has been referred to in the literature (Meng et al., 2023; Zhang et al., 2024; Annaki et al., 2024). Furthermore, generating high-quality augmentations in an unsupervised setting and independently of the model objective allows reuse of the augmented data and can further speed up model training (see Figure 1, Right).

To fill this gap, we introduce Automated Time Series Augmentation (AutoTSAugment), the first modular, model-agnostic AutoML framework tailored for time series augmentations in unsupervised settings. AutoTSAugment is based on a search space that includes the most common augmentation methods from the literature on time series augmentation (see Section 3.1) and is used to obtain a combination of augmentation methods. The main contributions of this study are as follows:

1. Building the first modular, model-agnostic augmentation framework tailored for unsupervised contrastive-based time series representation learning, which consists of a comprehensive search space for time series augmentation methods, allowing the selection of a variable number augmentation strategies, and an effective search strategy.

2. Introducing a search objective that extend beyond that of standard AutoML systems to evaluate the quality of a selection of augmentation methods for contrastive learning in unsupervised settings.

3. Selecting a search strategy by studying the effectiveness of sophisticated and simple search algorithms (i.e., Bayesian optimisation, random sampling, and random search) in addition to showing a significant speed-up possibility by adaptively narrowing the search space based on the dataset characteristics.

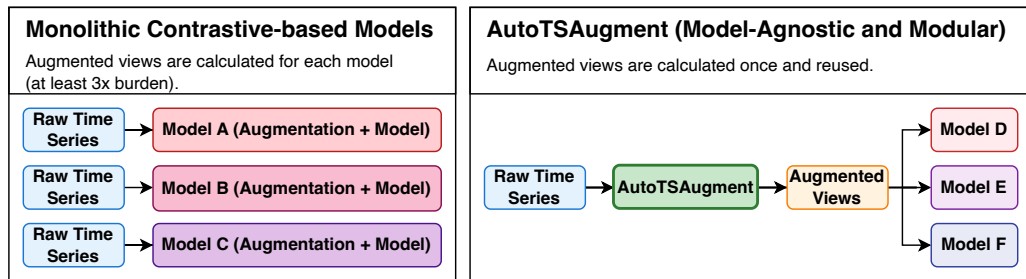

Figure 1: Comparison between our proposed AutoTSAugment framework and monolithic contrastive-based models. **Left:** Monolithic approaches require each model (A, B, C) to independently perform augmentation, resulting in redundant computation and at least triple computational burden for the same number of models. **Right:** AutoTSAugment adopts a modular, model-agnostic design, with augmented views computed once and reused across multiple models (D, E, and F).

4. Evaluating the effectiveness of the AutoTSAugment framework in combination with three baseline contrastive learning methods on four different downstream tasks, namely classification (univariate and multivariate) and forecasting (univariate and multivariate), on a total of 164 widely-used datasets.

The remainder of the paper is structured as follows: First, in Section 2, we discuss related work. Next, in Section 3, we describe the proposed framework, followed by the experimental setup in Section 4. After that, in Section 5, we report our empirical results; finally, in Section 6, we summarise our main findings, followed by a discussion of limitations and potential future work.

## 2 RELATED WORK

Data augmentation is a common approach to improve the generalisation performance of neural networks by teaching the model about invariances in the data (Goodfellow et al., 2016). This approach has shown improvements in performance across various tasks, including image classification (Krizhevsky et al., 2012; Shorten & Khoshgoftaar, 2019) and time series classification (Um et al., 2017; Ismail Fawaz et al., 2019; Foumani et al., 2024). In what follows, we review automated data augmentation methods for computer vision followed by methods tailored to time series.

**Automated Data Augmentation Methods for Computer Vision.** SmartAugment (Lemley et al., 2017) was one of the first automated augmentation approaches for computer vision tasks, implemented in the form of a neural network that automatically combines two or more samples from the same class to create augmented images. AutoAugment (Cubuk et al., 2019) is an automated augmentation approach with a search space defined over image augmentation methods that utilises Reinforcement Learning (RL) as a search strategy to find augmentation policies that are transferable between datasets. Despite the positive results of AutoAugment in obtaining good augmentations, the use of RL as a search algorithm introduces long running times. Cubuk et al. (2020) proposed RandAugment, a simple automated augmentation approach that samples a set of transformations from a group of $K$ image augmentation methods with an equal probability $\frac{1}{K}$ for each mini-batch in the training phase. Other studies (Lim et al., 2019) have attempted to overcome the expense of the search phase by using a smaller proxy task, such as training a smaller model or training the model on a reduced dataset size. All of the aforementioned augmentation methods are designed to work with any computer vision model (model-agnostic). However, these methods have a limited scope as they only work in computer vision tasks (not designed to work with different tasks) and only in supervised settings with evaluation criteria coupled with a downstream task (they require labels).

**Automated Data Augmentation Methods for Time Series.** In the context of automated data augmentation for time series data, related work falls into two main categories: (i) supervised, and (ii) unsupervised approaches. Examples of supervised approaches are: MODALS (Cheung & Yeung, 2020), W- and $\alpha$-Augment (Fons et al., 2021), and AutoCL (Jing et al., 2024). MODALS is an augmentation framework that operates in the latent space rather than the input space, which allows it to be applicable to multiple data modalities, including text, tabular, time series and image data

(modality-agnostic). In this framework, an encoder-classifier pair is used as the backbone for the transition from the input space to the latent space. Moreover, MODALS utilises population-based augmentation to discover augmentation policies. Fons et al. (2021) proposed two automatic augmentation policies for time series data. The first one learns to weight the contribution of the augmented samples to the loss, and the second selects a subset of transformations based on the ranking of the predicted training loss. The policies collect several augmentation operations during the training phase, which is achieved either through the selection of augmented samples based on their loss, or by adjusting the loss contributions of each augmented sample. AutoCL is a framework that utilises RL to search for a complete contrastive pipeline by jointly optimising for the augmentation pair, encoder and projection head, as well as loss hyperparameters. Each pipeline is pretrained contrastively and evaluated using validation accuracy, which serves as reward signal. AutoCL is not model-agnostic, as its search space couples augmentation with architecture and loss choices. The major limits of these approaches is the need for labelled instances, making them limited to supervised settings.

The second group of studies has targeted learning time series augmentation strategies in unsupervised settings, which allows benefiting from massive unlabelled data. Examples of work in this category are CoST (Woo et al., 2022), TS2Vec (Yue et al., 2022), and AutoTCL (Zheng et al., 2024). CoST employs an augmentation method that randomly composes three augmentation methods in the following order: jitter, scale, and shift. Each method is applied based on a user-determined probability $p$, which leads to multiple combinations of these methods during training. TS2Vec employs an augmentation method that extracts augmented views by performing random cropping and temporal shifting. AutoTCL has a neural network-based augmentation method consisting of a factorisation head and transformation head and operates as follows: First, the informative part is extracted from the input time series using the factorisation head. Random noise is then injected into this informative part using the transformation head. This neural network-based augmentation method, unlike the previous methods that apply clear transformations, is inherently unexplainable, which might not be suitable for applications where it is necessary to understand how an augmented view is derived. All the aforementioned augmentation methods are monolithic as they are tightly coupled with their respective model architectures, making them difficult to use with other models (none of them are proven to be both modular and model-agnostic).

As evident from the literature, while significant progress has been made in automated augmentation for domains such as computer vision, the time series domain—particularly in unsupervised learning—remains comparatively underexplored. To fill the gaps in the literature, it is necessary to develop an automated augmentation methods that satisfies the following requirements: (i) tailored to time series, (ii) works in unsupervised settings, (iii) model-agnostic, (iv) modular, and (v) interpretable. To the best of our knowledge, the development of a framework that satisfies all the aforementioned requirements has not been previously explored, making our study the first dedicated effort in this direction.

## 3 PROPOSED FRAMEWORK

The main goal of our work is to provide a modular, model-agnostic augmentation framework that can be used to obtain positive and negative views of an input time series to assist in training contrastive-based models for time series representation learning in unsupervised settings. The resulting AutoTSAugment framework consists of two main components: (i) search space, and (ii) search strategy. The search space is a set of time series augmentation methods along with their hyperparameters, and the search algorithm is a method based on which augmentation methods are sampled from the search space. The following sections provide details of these components.

### 3.1 SEARCH SPACE

The search space of AutoTSAugment, denoted by $\mathcal{S}$, represents all valid combinations of augmentation methods and their hyperparameters which can be sampled during the training process. Formally, we define the search space as:

$$\mathcal{S} = \{f_k \circ f_{k-1} \circ \cdots \circ f_1 \mid f_i \in \mathcal{M}, \boldsymbol{\lambda}_i \in \Lambda_i, 1 \le k \le K\}, \tag{1}$$

where $\mathcal{M}$ is the set of available augmentation functions, $\Lambda_i$ represents the hyperparameter space for augmentation function $f_i$, and $\boldsymbol{\lambda}_i \in \Lambda_i$ denotes the specific hyperparameters for $f_i$. Each $f_i$

is selected from $\mathcal{M}$ and parameterised by $\boldsymbol{\lambda}_i$. The composition $f_k \circ f_{k-1} \circ \cdots \circ f_1$ represents the sequential application of $k$ augmentation functions in a specific order (i.e., $f_1$ is applied first, followed by $f_2$, etc.), with $1 \leq k \leq K$, where $K$ is the maximum allowed number of augmentation functions in a single composition. In this study, we considered a set of time series augmentation methods frequently used in the literature (Um et al., 2017; Meng et al., 2023), namely: (i) Magnitude Warp, (ii) Time Warp, (iii) Window Slice, (iv) Jitter, (v) Scaling, and (vi) Permutation. Detailed description of the aforementioned methods and their hyperparameters is available in Appendix B.

## 3.2 SEARCH STRATEGY

Contrastive-based methods learn by using positive and negative examples (examples similar to and dissimilar to an anchor, respectively) to generate representations that are as close as possible in the embedding space for positive examples and as far as possible for negative examples. This is achieved when (i) the positive pairs are similar but not identical, and (ii) the negative pairs are dissimilar, but still lie on the same data manifold (Saunshi et al., 2019). This necessitates using distance-based measures with a threshold ($\tau$) to control the distance between the pairs and ensures that the extracted augmentation adheres to the requirements. The threshold prevents the search algorithms from extracting augmentation that result in augmented samples that are too close (i.e identical) or too far (i.e., very different) from the anchor (see Equation 2 for details).

**Search Objective.** The search objective in the AutoTSAugment framework is responsible for guiding the search within the defined search space $\mathcal{S}$ to identify an augmentation composition $f \in \mathcal{S}$ that extracts augmented views for each sample in a batch of samples $\mathcal{X}$. Formally, the search objective of is expressed as (Baratchi et al., 2024): $f^* \in \arg\max_{f \in \mathcal{S}} \mathcal{P}(f; \mathcal{X})$, where $f^*$ denotes the optimal augmentation composition and $\mathcal{P}(f; \mathcal{X})$ represents the performance metric based on a time series-related distance metric that measures the distance between the input batch of samples $\mathcal{X}$ and the augmented views $\tilde{\mathcal{X}}$ generated by the composition $f$. The performance metric $\mathcal{P}(f; \mathcal{X})$ is designed to achieve either the closest possible augmented view above a certain threshold (in the case of a positive view) or the furthest possible augmented view below a certain threshold (in the case of a negative view). We formalise $\mathcal{P}(f; \mathcal{X})$ as:

$$\mathcal{P}(f; \mathcal{X}) = \Phi_\tau \left( \mathcal{O}(\mathcal{D}) \right), \mathcal{D} = \left\{ d(\mathbf{x}, \tilde{\mathbf{x}}) \mid \mathbf{x} \in \mathcal{X}, \tilde{\mathbf{x}} \in \tilde{\mathcal{X}}, \tilde{\mathcal{X}} = f(\mathcal{X}) \right\}, \tag{2}$$

where $\mathcal{D}$ is the set of calculated distances between original and augmented samples, and $\mathcal{O}$ is an operator that can take different forms, depending on the desired data aggregation choice; for example, $\mathcal{O}_{\text{avg}}(\mathcal{D}) = \frac{1}{|\mathcal{D}|} \sum_{d \in \mathcal{D}} d$ (average distance), $\mathcal{O}_{\text{min}}(\mathcal{D}) = \min(\mathcal{D})$ (minimum distance), and $\mathcal{O}_{\text{max}}(\mathcal{D}) = \max(\mathcal{D})$ (maximum distance). $\Phi_\tau(\cdot)$ is a scoring function that incentivises the aggregated distance to be either near or far from a threshold $\tau$, depending on whether a positive or negative view is required. We define the scoring function $\Phi_\tau(\cdot)$ as: $\Phi_\tau(d) = s \cdot |d - \tau|$, where $s \in \{-1, +1\}$ represents the direction of the scoring objective and is determined based on the desired view in contrastive learning: $s = -1$ corresponds to a positive view, where the function penalises distances far from the target value $\tau$ and encourages solutions where $d$ is close to $\tau$; $s = +1$ corresponds to a negative view, where the function rewards distances far from $\tau$ and pushes solutions away from it.

**Search Algorithms.** In our experiments, we used three search algorithms: Random Sampling, Random Search, and Bayesian Optimisation Search (the details of these search algorithms are presented in Section 4.1). While other automated augmentation methods in the literature utilised RL in domains such as computer vision (Cubuk et al., 2019), we chose to avoid using it as a search algorithm for the following reasons: (i) RL, in general, is resource-intensive and requires longer running times, and (ii) follow up research (Cubuk et al., 2020) demonstrated that random sampling can achieve similar results as RL while being significantly faster.

## 4 EXPERIMENTAL SETUP

In this section, we outline and describe the experiments along with the search algorithms, datasets, and baseline methods, in addition to the research questions that we aimed to answer.

Table 1: Hyperparams of BOSearch and RandSearch and the values considered in the experiments.

| HP | Description | Values[2] |
|---|---|---|
| $n_{min}$ | The minimum number of augmentation methods to sample. $n_{min} \geq 1$. | 1 |
| $n_{max}$ | The maximum number of augmentation methods to sample. $n_{max}. \geq n_{min}$. | 3 |
| $m$ | The number of iterations the search algorithm carries out. | 100 |
| $\mathcal{G}$ | The search goal: determines whether to obtain a positive or negative view of an input sample. | Positive |
| $d$ | The distance metric used to calculate the similarity between two samples, such as Euclidean distance (Euc) and SoftDTW (SDTW). | Euc; SDTW |
| $\mathcal{O}$ | The aggregation strategy of the batches during training. | Min |
| $\tau$ | The threshold, where obtaining configurations with a score below it (in the case of positive $\mathcal{G}$) or above it (in the case of negative $\mathcal{G}$) is not desirable. | 2.5; 3.0 |
| $\omega$ | The penalty value applied for configurations that cross the threshold $\tau$. $0 \leq \omega < \tau$ for positive $\mathcal{G}$ and $\omega > \tau$ for negative $\mathcal{G}$. | 100 |

## 4.1 SEARCH SPACE AND ALGORITHMS

We built a search space that contains multiple methods with their parameters, as described in Section 3.1. Regarding the search algorithm, we aimed to determine whether a more sophisticated search algorithm brings substantial benefits compared to a random sampling approach for identifying high-quality augmentations. In this context, we considered three search algorithms that use the same search objective (presented in Section 3.2):

1. **Random Sampling (RandSampling):** This algorithm randomly samples a number of methods $n$, a user-specified hyperparameter, with their corresponding parameters from the search space, and then applies these to the given time series sample.

2. **Random Search (RandSearch):** This algorithm samples a number of augmentation methods $n \in [n_{min}, n_{max}]$, where both $n_{min}$ and $n_{max}$ are user-specified hyperparameters, for $m$ iterations, and then selects the most suitable configuration according to other user-specified hyperparameters.

3. **Bayesian Optimisation Search (BOSearch):** This algorithm was realised using Sequential Model-Based Optimisation for General Algorithm Configuration (SMAC) (Lindauer et al., 2022). The optimisation begins with an initial phase that evaluates $k$[1] randomly sampled configurations with combinations of $n \in [n_{min}, n_{max}]$ augmentation methods and their hyperparameters (conditioned on the augmentation method). After initialisation, $m$ iterations are carried out, in which the process alternates between: (1) training a random forest surrogate model on all previously evaluated configurations to predict augmentation quality (measured by batch distance thresholds presented in Table 1), and (2) selecting the next configuration to evaluate through the Expected Improvement acquisition function. At each iteration, the number of augmentation methods $n \in [n_{min}, n_{max}]$ is jointly optimised with the method-specific parameters. The surrogate model updates after each evaluation to balance exploration of novel combinations and exploitation of promising regions.

## 4.2 BASELINE METHODS

In our experiments, we considered multiple time series contrastive-based representation learning models as baseline methods, each with a specific augmentation strategy. We are interested in knowing if the model-specific augmentation strategy in the baseline can be replaced by our model-agnostic framework. We selected the baselines based on two main criteria: (i) the availability of open-source code, and (ii) the inclusion of a variety of augmentation method categories, namely neural network-based methods, randomised augmentation and random compositions of specific pre-selected methods. The models considered in the experiments are as follows: **TS2Vec (Yue et al., 2022)**, **CoST (Woo et al., 2022)**, and **AutoTCL (Zheng et al., 2024)**.

---

[1] $k = \mu \times nhp$, where $nhp$ is the total number of hyperparameters ($\mu = 10$ is the default value in SMAC3).

[2] Explanation behind selecting these values can be found in Appendix D.

### 4.3 EVALUATION PROCEDURE

Following the work of Yue et al. (2022) and Zheng et al. (2024), we considered the downstream tasks of time series classification (univariate and multivariate) and time series forecasting (univariate and multivariate) and used accuracy for classification and MSE loss for forecasting as evaluation metrics. Training was performed on the training and validation sets, where the validation set was used for early stopping, followed by extraction of representations using the trained model. The downstream tasks were then performed accordingly. Subsequently, a Friedman test and a Nemenyi post-hoc test were performed for each downstream task to acquire final rankings and assess the statistical significance of the results. The overall goal of the experiments was to evaluate the efficiency of our framework in replacing the original augmentation methods in the baseline time series representation learning models. Specifically, we aimed to answer the following research questions: (1) *How does changing the augmentation method and replacing it with AutoTSAugment affect the performance in downstream tasks? (2) How does the choice of search algorithm affect the performance when using AutoTSAugment, and what are the tradeoffs? (3) How does changing the search space affect the performance when using AutoTSAugment, and what are the tradeoffs?*

In total, 27 experiments were conducted. The first three experiments involved running the baseline methods using the original augmentation methods. In the other experiments, the original three augmentation methods were replaced with the AutoTSAugment framework and different variations of the three search methods and two search spaces. Further details of our experiments are presented in Table 2c. The hyperparameters of the search methods were set as shown in Table 1.

**Datasets.** We used 164 datasets that are widely used in the literature. For univariate time series classification, we used the UCR Dataset Archive (Dau et al., 2019) which contains 128 datasets. For multivariate time series classification, we used the UEA Dataset Archive (Bagnall et al., 2018), which contains 30 datasets. For univariate and multivariate forecasting, we used the following datasets: Electricity Load (Trindade, 2015), Weather Dataset (Zhou et al., 2021), and ETT dataset collection (Zhou et al., 2021), which consisted of four datasets: h1, h2, m1, and m2.

## 5 RESULTS

In this section, we present the results of our experiments in the context of the previously stated research questions. These results can be reproduced using the open-source-code available on GitHub[3].

**Q1. *How does changing the augmentation method and replacing it with AutoTSAugment affect the performance in downstream tasks?*** Table 2c summarises results on 164 datasets in terms of significant differences in performance between baseline models using the AutoTSAugment framework and the same baselines using their original augmentation methods (full results are available in Appendix F). It can be observed from these results that, generally, there is no significant difference in performance in any of the downstream tasks between using the original augmentation method of each baseline method and using the AutoTSAugment framework. In addition, the AutoTSAugment framework achieved better results with significant differences when it replaced a neural network-based augmentation method (AutoTCL (Zheng et al., 2024)) in the downstream task of univariate classification (see experiments $r2$-$1$, $r2$-$3$ in Table 2c). Achieving on par or better results using AutoTSAugment against tailored augmentation methods in representation learning algorithms is positive, demonstrating the general applicability of AutoTSAugment in combination with different time series representation learning algorithms.

**Q2. *How does the choice of search algorithm affect the performance when using AutoTSAugment and what are the tradeoffs?*** From Table 2c, it can be observed that changing the search algorithm of the AutoTSAugment framework did not lead to significant differences in performance, with Rand-Sampling achieving better results than variants of dedicated search algorithms in some cases (see Experiments $r2$-$1$, $r2$-$3$ in Table 2c). However, as presented in Table 2a, the training time for Rand-Sampling was 47.34% faster than RandSearch and 63.25% faster than BOSearch on average. This is due to the fact that RandSampling randomly samples a configuration from the search space without any iteration, unlike RandSearch and BOSearch, where iteration is required (see Section 4.1 for details). These results align with RandAugment (Cubuk et al., 2020), where the authors conducted

---

[3]https://gitfront.io/r/anonydev/tKfVphfW2iL7/AutoTSAugment/

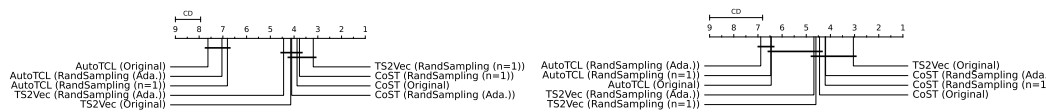

(a) Univariate Time Series Classification          (b) Multivariate Time Series Classification

Figure 2: **Critical Difference (CD) Diagrams for Adaptive Search Space Experiments** that illustrate the ranking of variants of the baseline models when using their original augmentation methods compared to variants employing the AutoTSAugment framework with and without adaptive search space. The rankings were determined using a Friedman test followed by a Nemenyi posthoc test.

a comparable comparison in the computer vision domain between random sampling and dedicated search algorithms, demonstrating that the search space itself is more important than the search algorithm in terms of accuracy—in their experiments, random sampling achieved the same results with significantly reduced training time.

***Q3. How does changing the search space affect the performance when using AutoTSAugment, and what are the tradeoffs?*** To inspect the effect of the search space and, specifically, to determine whether modifying the search based on the input data has an effect on the (i) performance of downstream tasks, and (ii) makes the search more efficient, we incorporated a recommendation component into our framework. This step narrows the search space based on the input dataset, making the search space adaptive rather than keeping it constant. The design of the recommendation component follows exactly the method of Liu et al. (2024), who introduced a recommendation system to suggest augmentation methods based on trend and seasonality analysis of the input dataset. This method compares the input dataset with 12 synthetically generated datasets each with different combinations of trend and seasonality components (i.e., trend-dominant, seasonality-dominant, and even trend and seasonality) with a total of 72 000 time series samples. It then detects the datasets most similar to the input dataset, and recommends augmentation methods (without specific hyperparameters) based on this detection and a group of pre-computed and pre-ranked augmentations for each synthetic dataset (for more details about the method, refer to Liu et al. (2024)).

In our integration, we used their method and their pre-computed results to narrow the search space. Given that their approach recommends methods that can be applied individually and does not consider the composition of multiple methods, it was possible to use RandSampling with $n = 1$ as a search algorithm in combination with their method—setting $n$ to values $n > 1$ leads to a composition of augmentation methods. Other search algorithms considered in our work, RandSearch and BOSearch, optimise the number of augmentation methods as part of the search; this also leads to composition of augmentation methods and therefore cannot be used in combination with this recommendation component. Furthermore, their method is specifically tailored for time series classification and has not been tested on other downstream tasks; therefore, it was only possible to test this additional component with time series classification. The results from our analysis (full results are available in Tables 9 and 10 in Appendix F) demonstrate that changing the search space leads to a more efficient search process: using AutoTSAugment with the refined search space achieved similar results to that of the original augmentation method of each baseline while being faster than using the full search space by 34.42% on univariate time series classification and 58.21% on multivariate time series classification (see Table 2b, Figure 2, and experiments $ar1$, $ar2$, and $ar3$ in Table 2c).

## 6  CONCLUSIONS

In this study, we introduced Automated Time Series Augmentation (AutoTSAugment), a modular, model-agnostic data augmentation framework for obtaining augmented views of samples for unsupervised contrastive-based time series representation learning We evaluated the performance of AutoTSAugment using the downstream tasks of time series classification (univariate and multivariate) and time series forecasting (univariate and multivariate), achieving performance on par with the baseline models using their original augmentation methods across all evaluated downstream tasks. Furthermore, our experiments provided insights into how to set up this framework and its hyperparameters for different search algorithms. Notably, we found that using RandSampling can obtain results similar to those obtained from dedicated search algorithms, such as RandSearch and

Table 2: Results Analysis. (a) and (b) Comparison of training time (in hours) between the search algorithm. (c) Significance comparison between the original augmentation and other variants across all experiments using a Friedman testfollowed by a Nemenyi posthoc test. The table shows whether each augmentation variant performs significantly better ($\uparrow$) or no difference ($=$) than the original augmentation. Here, ID stands for the experiment ID, C for classification, F for forecasting, U for univariate, M for multivariate, Ada. for Adaptive Search Space, and ($\times$) for "not relevant".

(a) Main Experiments

| Task | Variant | Mean $\pm$ Std. |
|---|---|---|
| UC | RandSampling | **6.69 $\pm$ 9.89** |
| | RandSearch | 20.46 $\pm$ 17.74 |
| | BOSearch | 27.38 $\pm$ 21.25 |
| MC | RandSampling | **12.12 $\pm$ 17.37** |
| | RandSearch | 21.24 $\pm$ 17.61 |
| | BOSearch | 32.72 $\pm$ 24.03 |
| UF | RandSampling | **0.07 $\pm$ 0.10** |
| | RandSearch | 0.10 $\pm$ 0.10 |
| | BOSearch | 0.16 $\pm$ 0.12 |
| MF | RandSampling | **3.28 $\pm$ 10.97** |
| | RandSearch | 7.28 $\pm$ 16.20 |
| | BOSearch | 8.35 $\pm$ 18.77 |
| RandSampling is 47.34% and 63.25% faster than RandSearch and BOSearch, respectively. | | |

(b) Adaptive Search Space Experiments

| Task | Variant | Mean $\pm$ Std. |
|---|---|---|
| UC | RandSampling | 7.23 $\pm$ 10.67 |
| | RandSampling (Ada.) | **4.74 $\pm$ 7.16** |
| MC | RandSampling | 11.92 $\pm$ 16.83 |
| | RandSampling (Ada.) | **4.98 $\pm$ 10.27** |
| RandSampling (Ada.) is 34.42% and 58.21% faster than RandSampling on UC and MC, respectively. | | |

(c)

| | ID | Variant | C | | F | |
|---|---|---|---|---|---|---|
| | | | U | M | U | M |
| **TS2Vec** | s1 | BOSearch (Euc) | = | = | = | = |
| | s11 | BOSearch (SoftDTW) | = | = | = | = |
| | rs1 | RandSearch (Euc) | = | = | = | = |
| | rs11 | RandSearch (SoftDTW) | = | = | = | = |
| | r1-1 | RandSampling (n=1) | = | = | = | = |
| | r1-2 | RandSampling (n=2) | = | = | = | = |
| | r1-3 | RandSampling (n=3) | = | = | = | = |
| | ar1 | RandSampling (Ada.) | = | = | × | × |
| **AutoTCL** | s2 | BOSearch (Euc) | = | = | = | = |
| | s21 | BOSearch (SoftDTW) | = | = | = | = |
| | rs2 | RandSearch (Euc) | = | = | = | = |
| | rs21 | RandSearch (SoftDTW) | = | = | = | = |
| | r2-1 | RandSampling (n=1) | ↑ | = | = | = |
| | r2-2 | RandSampling (n=2) | = | = | = | = |
| | r2-3 | RandSampling (n=3) | ↑ | = | = | = |
| | ar2 | RandSampling (Ada.) | = | = | × | × |
| **CoST** | s3 | BOSearch (Euc) | = | = | = | = |
| | s31 | BOSearch (SoftDTW) | = | = | = | = |
| | rs3 | RandSearch (Euc) | = | = | = | = |
| | r31 | RandSearch (SoftDTW) | = | = | = | = |
| | r3-1 | RandSampling (n=1) | = | = | = | = |
| | r3-2 | RandSampling (n=2) | = | = | = | = |
| | r3-3 | RandSampling (n=3) | = | = | = | = |
| | ar3 | RandSampling (Ada.) | = | = | × | × |

BOSearch, whine being 47.34% and 63.25% faster on average, respectively. Furthermore, we found that refining the search space based on the characteristics of the input data can speed up the training by 58.21%.

By providing a modular, model-agnostic, and efficient approach to time series data augmentation, AutoTSAugment offers a fully automated augmentation strategy for any time series dataset. It facilitates the automatic generation of reusable augmented time series by decoupling augmentation from modelling, which in turn accelerates the training of contrastive learning models. Moreover, it establishes a standardised augmentation benchmark that enables consistent evaluation and refinement of different model architectures without relying on labour-intensive manual augmentation selection.

**Limitations and Future Work.** In our study, the robustness of the learned representations to noise and missing values was not explicitly addressed. The impact of such irregularities on time series representation learning has not yet been much explored in the literature (Zhang et al., 2024). When such robustness measures are established in the literature, future work could involve incorporating these measures into the framework to produce the most robust representations possible. Additionally, although initial experiments using an adaptive search space that tailors the available augmentation methods for the input dataset only showed changes in search speed, using more sophisticated augmentation methods recommendation systems may still have potential in obtaining better results. Future work could investigate designing time series augmentations recommendation systems beyond comparing with a static number of synthetic datasets by collecting performance data of state-of-the-art time series representation learning models across multiple downstream tasks then training a meta-learner using this data—a challenging task that requires the design of novel time series augmentation recommendation system and a large amount of computing power; therefore, it is beyond the scope of this paper.

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

## A  BASELINE METHODS DESCRIPTION

In this section, a detailed description of the baseline methods used in the paper is provided. The baseline methods used in the paper are as follows:

- **TS2Vec (Yue et al., 2022):** TS2Vec is a time series representation learning model that carries out augmentation by performing uniform random cropping of the original sample.

- **CoST (Woo et al., 2022):** CoST is a time series representation that carries out augmentation by randomly applying compositions of specific pre-selected methods.

- **AutoTCL (Zheng et al., 2024):** AutoTCL is a time series representation learning model that employs a neural network-based augmentation method.

## B  AUGMENTATION METHODS DESCRIPTION

Table 3 presents detailed descriptions of the augmentation methods considered in the search space of AutoTSAugment and their corresponding hyperparameters.

Table 3: Time series augmentation methods considered in the AutoTSAugment framework and their hyperparameters.

| Method | Method Description | Hyperarameters |
|---|---|---|
| **Magnitude Warp** | Applies a smooth, nonlinear transformation to the amplitude by multiplying it with a warping curve. | $\sigma \in [0.01, 0.5]$: Standard deviation controlling warping randomness. $k \in [2, 10]$: Number of knot points in the spline. |
| **Time Warp** | Applies a smooth, nonlinear transformation to the time axis of the time series. | $\sigma \in [0.01, 0.5]$: Standard deviation controlling time warping randomness. $k \in [2, 10]$: Number of knot points in the spline. |
| **Window Slice** | Extracts a continuous subsequence (window) from the original time series. | $r \in [0.5, 1.0]$: Reduce ratio of window length to original length. |
| **Jitter** | Introduces random noise into the time series by adding Gaussian-distributed noise. | $\sigma \in [0.01, 0.8]$: Standard deviation of Gaussian noise. $s \in [0.1, 2.0]$: Scaling factor that adjusts the amplitude before adding noise. |
| **Scaling** | Modifies the amplitude of the time series by multiplying it with a scaling factor. | $\alpha \in [0.1, 2.0]$: Scaling factor applied to the time series. |
| **Permutation** | Divides the time series into multiple segments and rearranges them. | $n_{\text{max\_segments}} \in [1, 10]$: Maximum number of segments. seg_mode: Mode of segmentation ('random', 'other'). |

## C  DATASETS SPLIT DETAILS

For the classification datasets (UCR and UEA Dataset Archives), 50% of the samples were used for training and 50% for testing, and 25% of the training samples were used for validation. We did not use the default split in these datasets for the following reasons: (i) a validation set was needed for our experiments, but was not provided in the default split, and (ii) the train/test ratios in the standard splits vary widely, and in some cases, there are not sufficiently many training samples to extract a validation set. Re-splitting all datasets in a consistent way ensures comparable sample sizes across datasets, which in turn enables a fair comparison. For the forecasting datasets, following the work of Yue et al. (2022), 12 months of data samples were used for training, 4 months for validation, and 4 months for testing for the ETT dataset collection, whereas for the Weather and Electricity Load datasets, 60% of the samples were used for training, 20% for validation, and 20% for testing.

# D  DETAILS ABOUT SETTING THE HYPERPARAMETERS

To gain insights into setting the parameters of both search algorithms used in the paper, we generated synthetic time series samples following the generation process used by Zaffran et al. (2022) according to Equation 3, and then conducted analysis using the generated samples.

$$Y_t = 10\sin(\pi X_{t,1} X_{t,2}) + 20(X_{t,3} - 0.5)^2 + 10X_{t,4} + 5X_{t,5} + 0X_{t,6} + \epsilon_t, \tag{3}$$

where $X_t$ is a vector of explanatory variables uniformly sampled from $[0, 1]$, with components $X_{t,1}, X_{t,2}, \ldots, X_{t,6}$, $\epsilon_t$ is noise generated from an ARMA(1,1) process, i.e., $\epsilon_t = \phi\epsilon_{t-1} + \xi_t + \theta\xi_{t-1}$, where $\xi_t$ is white noise, and $\phi$ and $\theta$ are parameters controlling the temporal dependence of the noise.

## D.1  RANDSAMPLING

RandSampling has only one hyperparameter to set, which is the number of augmentation methods to sample from the search space and use them to calculate the augmented samples based on the input sample ($n$). In order to have insights about the value on $n$ and its effect on the distance between the input and augmented samples, we conducted the following analysis:

1. Generate 10 000 time series sample according to the aforementioned process.

2. Calculate the augmented view for each sample using the following values for $n$: 1, 2, 3, 4, 5, 6, 7, and 8.

3. Calculate the distances between the samples and the augmented views according to both Euclidean distance and SoftDTW.

4. Inspect the statistics and distributions of these calculated distances and display a Empirical Cumulative Distribution Function (eCDF) for each distance metric.

As can be seen in Figure 4, where the eCDFs of distance metrics are displayed, the distance does not always increase by increasing $n$ (i.e., it is periodic) with some values of $n$ increasing the distance and other not—this phenomenon caused by some sampled augmentation methods cancelling each other. Based on this, $n$ values of 1, 2, and 3 can all be suitable for fetching positive samples for an input sample, as they produce samples that are not far on average, with values such as $n = 7$ being undesirable to test as they produce similar distances to $n = 2$.

## D.2  RANDSEARCH AND BOSEARCH

Both the RandSearch and BOSearch algorithms have a number of hyperparameters to be set.

On the one hand, both $\mathcal{G}$ and $\omega$ were straightforward to set, as the baseline methods use augmentation methods to fetch only a positive example to the input sample. Therefore, in order to align with this, $\mathcal{G}$ was set to value "Minimum" so that it looks for a sample that is as close as possible to the input sample, and $\omega$ was set to 100, which is the value to assign to augmentations configuration that produces samples too close to the input sample, i.e., below a threshold $\tau$, making it an undesirable choice for the search algorithm. $\mathcal{O}$ was set to "Minimum" in this case as well, which means that the optimisation process works to maintain a distance of at least $\tau$ between the original and augmented samples.

On the other hand, to set the other hyperparameters, we conducted further analysis. Based on the results of the analysis related to RandSampling (see Section D.1) $n_{min}$ and $n_{max}$ can be set to 1 and 3, respectively, as these values were shown to produce augmented views that are close to input samples. To set $\tau$, we generated two sets of 10 000 time series samples and studied the distribution of the distance metrics. Based on the distributions shown in Figure 3, $\tau$ was set to 2.5 and 3.0 for Euclidean distance and SoftDTW, respectively—these values prevent the framework from obtaining augmented views that are "too close" to the original samples and therefore defeating the whole purpose of contrastive learning.

Another method of setting the hyperparameters of RandSearch and BOSearch is to use automated analysis to dynamically optimise these hyperparameters for each dataset. However, this method has limitations and considerations, such as: (i) smaller datasets might not have enough samples for

conducting such analysis, and (ii) when training one model on multiple datasets, such as in the case of Time Series Foundation Models (TSFMs), it would be difficult to estimate it for multiple datasets. Therefore, we leave such automated adaptation for potential future work.

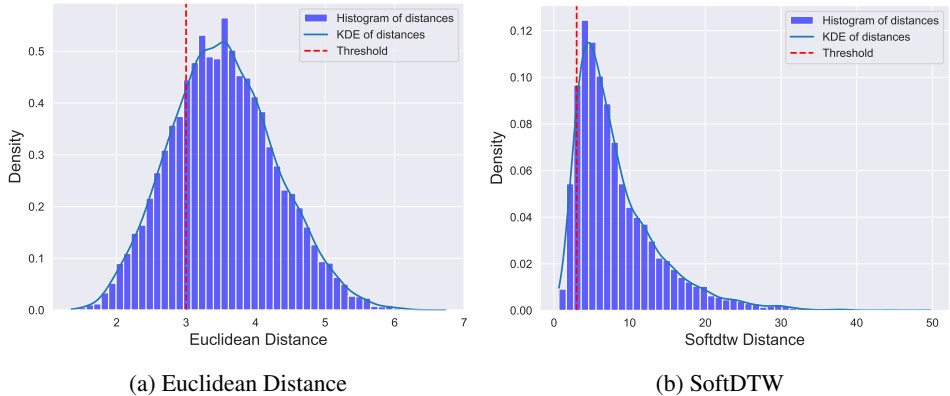

(a) Euclidean Distance         (b) SoftDTW

Figure 3: Distributions of distance metrics calculated between original random time series samples and their augmented counterparts.

# E  ADDITIONAL EXPERIMENT ON TIME SERIES FOUNDATION MODELS (TSFMS)

Considering the success of large language foundation models in Natural Language Processing (NLP), the time series community has become increasingly interested in the potential of Time Series Foundation Models (TSFMs) (Ye et al., 2024). However, there have been challenges in producing TSFMs as capable as the foundation models in NLP; these challenges include the relatively smaller available time series datasets compared to the vast corpus available in NLP. To address this challenge, some models, such as TEST (Sun et al., 2024) and TimeCLR (Yeh et al., 2023), have resorted to data augmentation to generate more data during training and utilised a contrastive learning-based loss function; however, as we discussed earlier, with this approach comes the challenge of selecting a combination of augmentation methods to use. This renders the proposed AutoTSAugment framework a potential solution for training TSFMs.

We therefore experimented with the AutoTSAugment on TEST using multivariate time series classification as the downstream task and 30 datasets from the UEA Archive[4]. We aimed at answering the following question: *How does replacing augmentation method in a contrastive-based TSFM with AutoTSAugment affect its performance in multivariate time series classification task?*

Towards this end, TEST was trained once using the original augmentation method and once using the AutoTSAugment framework; the difference in performance was measured using the Wilcoxon signed-rank test, resulting in a test statistic of 5.0 and a p-value of 0.0687. These results align with those of the experiments from the paper, making the AutoTSAugment framework suitable for training contrastive-based TSFMs.

---

[4]The experiments were conducted on TEST (Sun et al., 2024) as its code is open-sourced, and the experiments were limited to the UEA Archive due to limitations related to available computing resources—it was not possible to train using additional 128 datasets from the UCR Archive.

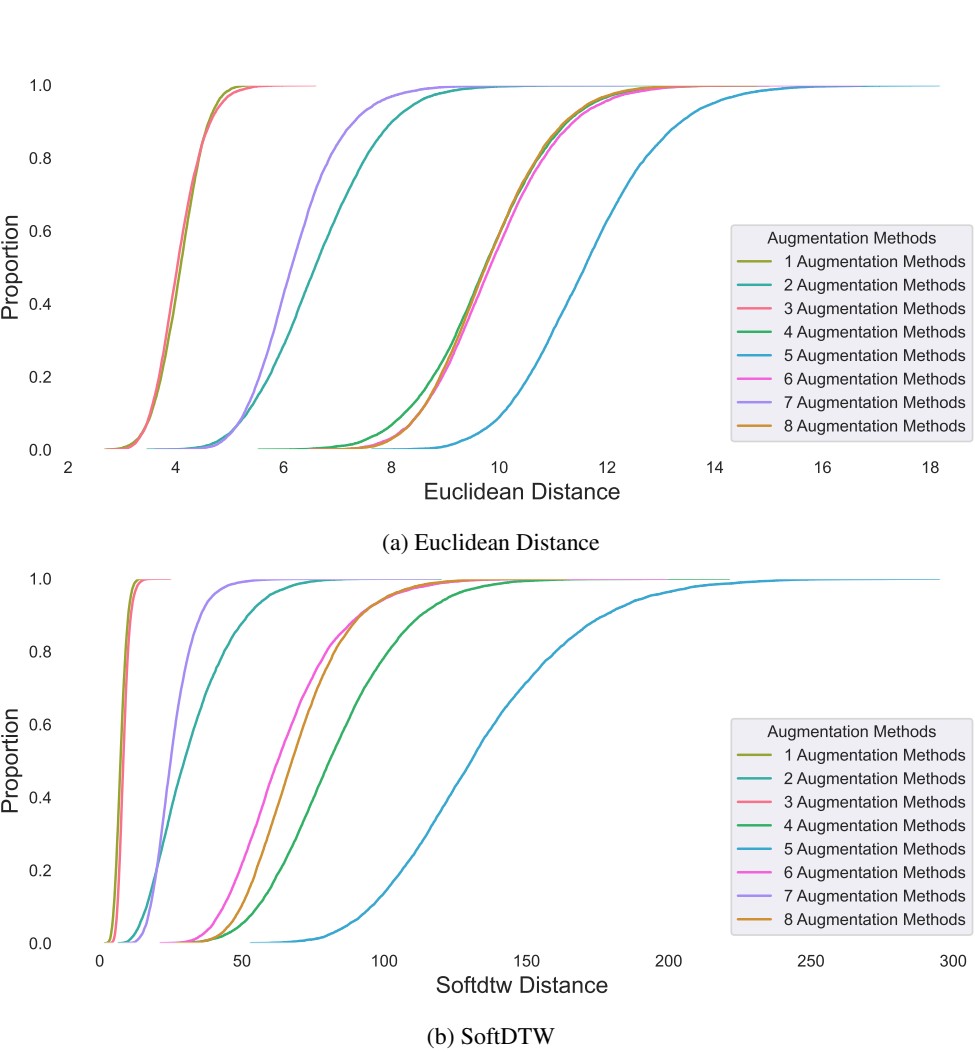

(a) Euclidean Distance

(b) SoftDTW

Figure 4: eCDFs of distances calculated between original random time series samples and their augmented counterparts, based on the number of augmentation methods applied. Each curve corresponds to a specific number of augmentation methods, illustrating how the distance distribution shifts as the number of applied augmentation method increases.

## F   DETAILED RESULTS

In this section, details of the results that are presented in the paper are listed in tables and are illustrated as CD diagrams for each experiment and each downstream task.

The CD diagrams are listed as follows:

- TS2Vec (Yue et al., 2022) experiments in Figure 7.
- AutoTCL (Zheng et al., 2024) experiments in Figure 8.
- CoST (Woo et al., 2022) experiments in Figure 9.

The detailed results of the main experiments are listed as follows:

- Univariate classification in Table 5.
- Multivariate classification in Table 6.
- Univariate forecasting in Table 7.
- Multivariate forecasting in Table 8

The detailed results of the adaptive search space experiments are listed as follows:

- Univariate classification in Table 9.
- Multivariate classification in Table 10.

The eCDF plots of the training time are displayed as follows:

- Training times of the main experiments in Figure 5.
- Training times of the adaptive search space experiments in Figure 6.

The detailed results of the experiments related to TEST (Sun et al., 2024), a TSFM, are listed in Table 4.

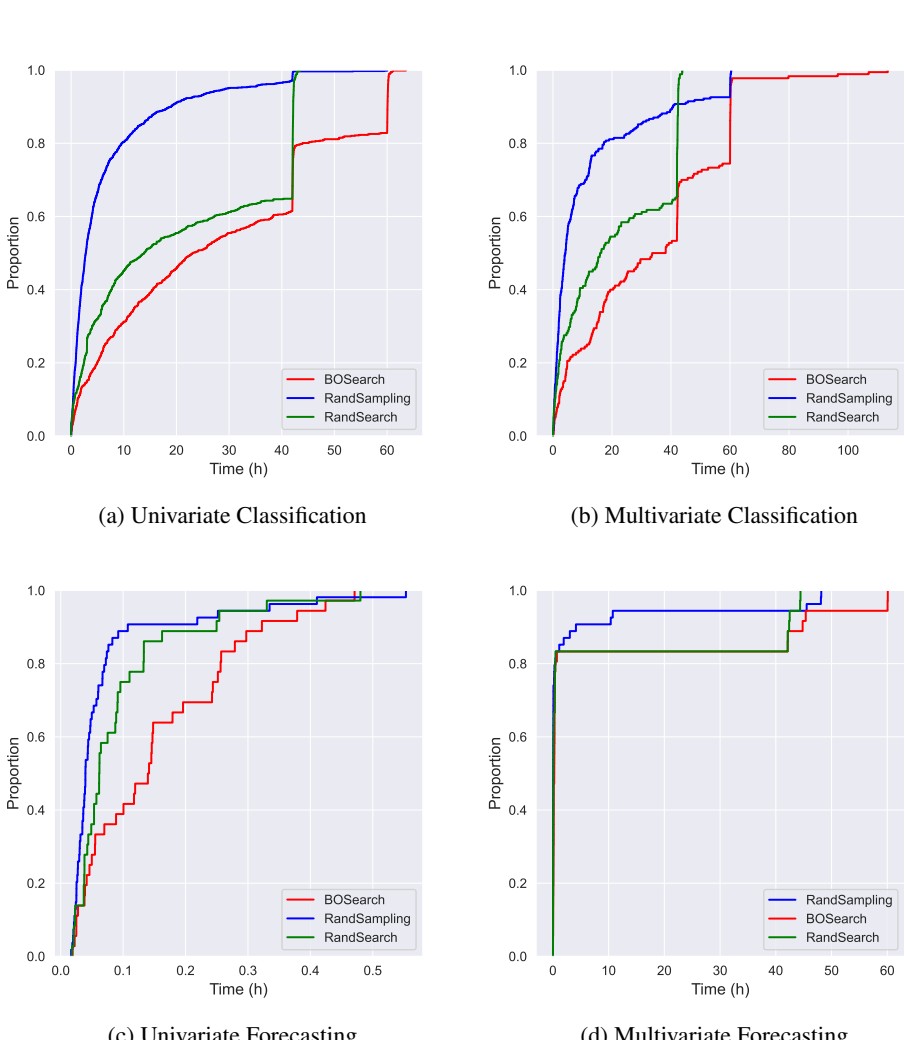

(a) Univariate Classification

(b) Multivariate Classification

(c) Univariate Forecasting

(d) Multivariate Forecasting

Figure 5: **Empirical Cumulative Distribution Function (eCDF) of Training Time for RandSampling, RandSearch, and BOSearch.** The x-axis represents the training time in hours (h), while the y-axis represents the proportion of runs (out of all experimental runs) that completed within a given training time These eCDF plots compare the training times of RandSampling, RandSearch, and BOSearch across various time series tasks, including Univariate Classification, Multivariate Classification, Univariate Forecasting, and Multivariate Forecasting. The comparisons were conducted using 164 datasets from UCR, UEA, ETT (ETTh1, ETTh2, ETTm1, and ETTm2), Electricity Load, and Weather datasets.

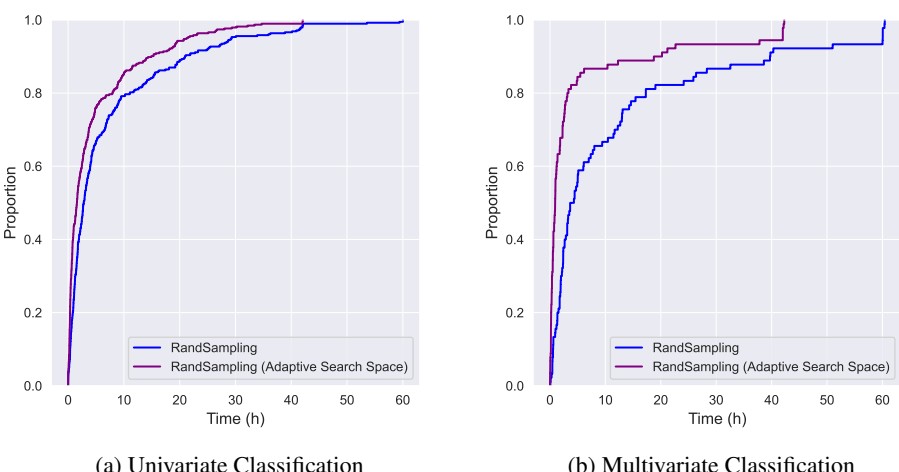

(a) Univariate Classification

(b) Multivariate Classification

Figure 6: **Empirical Cumulative Distribution Function (eCDF) of Training Time for RandSampling and RandSampling with Adaptive Search Space** The x-axis represents the training time in hours (h), while the y-axis represents the proportion of runs (out of all experimental runs) that completed within a given training time These eCDF plots compare the training times of RandSampling and RandSampling with Adaptive Search Space across Univariate Classification and Multivariate Classification downstream tasks. The comparisons were conducted using 158 datasets from the UCR and UEA Archives.

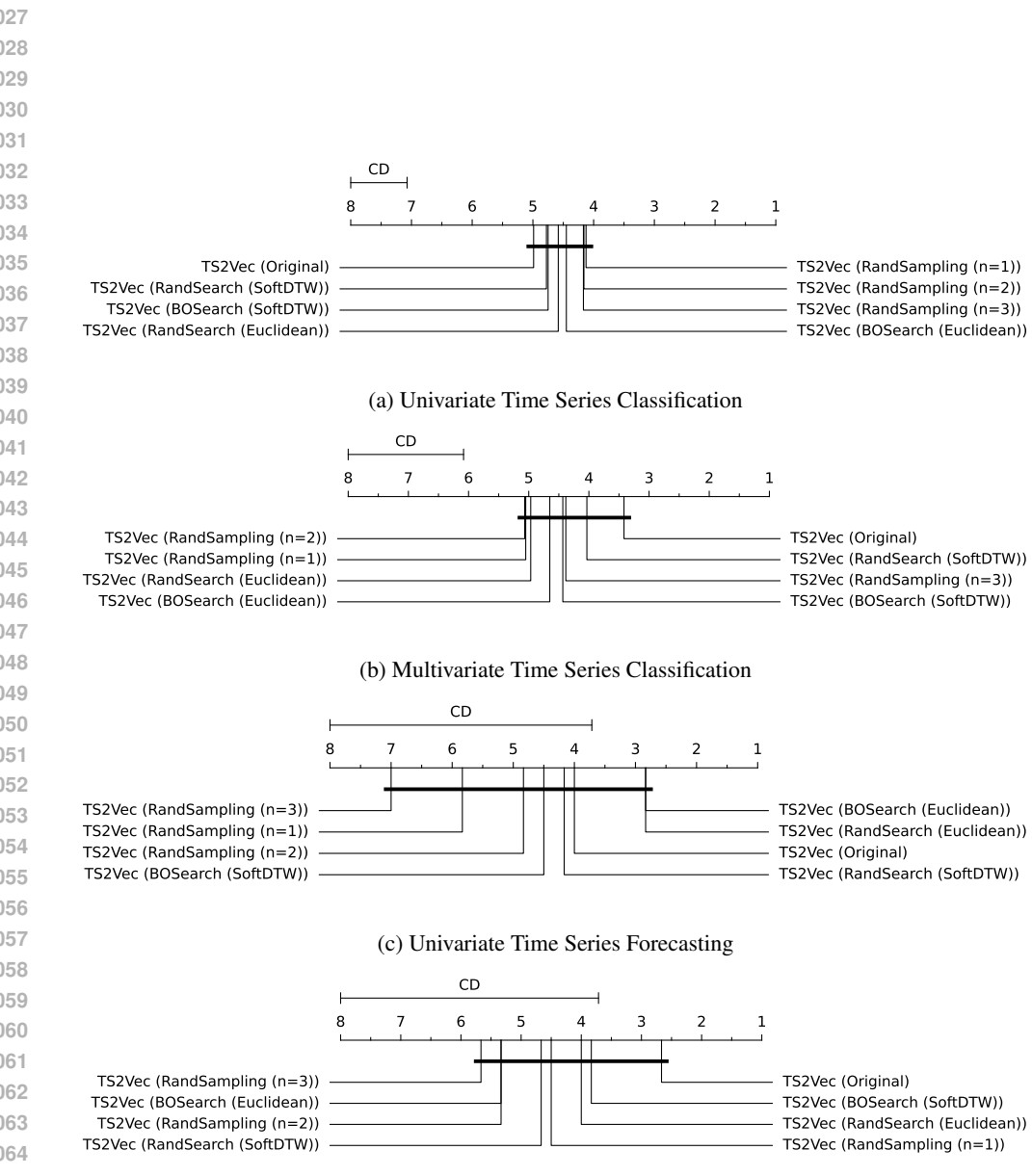

(a) Univariate Time Series Classification

(b) Multivariate Time Series Classification

(c) Univariate Time Series Forecasting

(d) Multivariate Time Series Forecasting

Figure 7: **CD Diagrams for Ts2Vec (Yue et al., 2022).** These CD diagrams illustrate the ranking of variants of the TS2Vec model when using its original augmentation methods compared to variants employing the AutoTSAugment framework. The comparisons were conducted across various time series classification downstream tasks, including univariate and multivariate classifications using UCR and UEA datasets, as well as time series forecasting tasks on ETT (ETTh1, ETTh2, ETTm1, and ETTm2), Electricity Load, and Weather datasets. The rankings were determined using a Friedman test followed by a Nemenyi posthoc test, highlighting statistically significant differences between the methods.

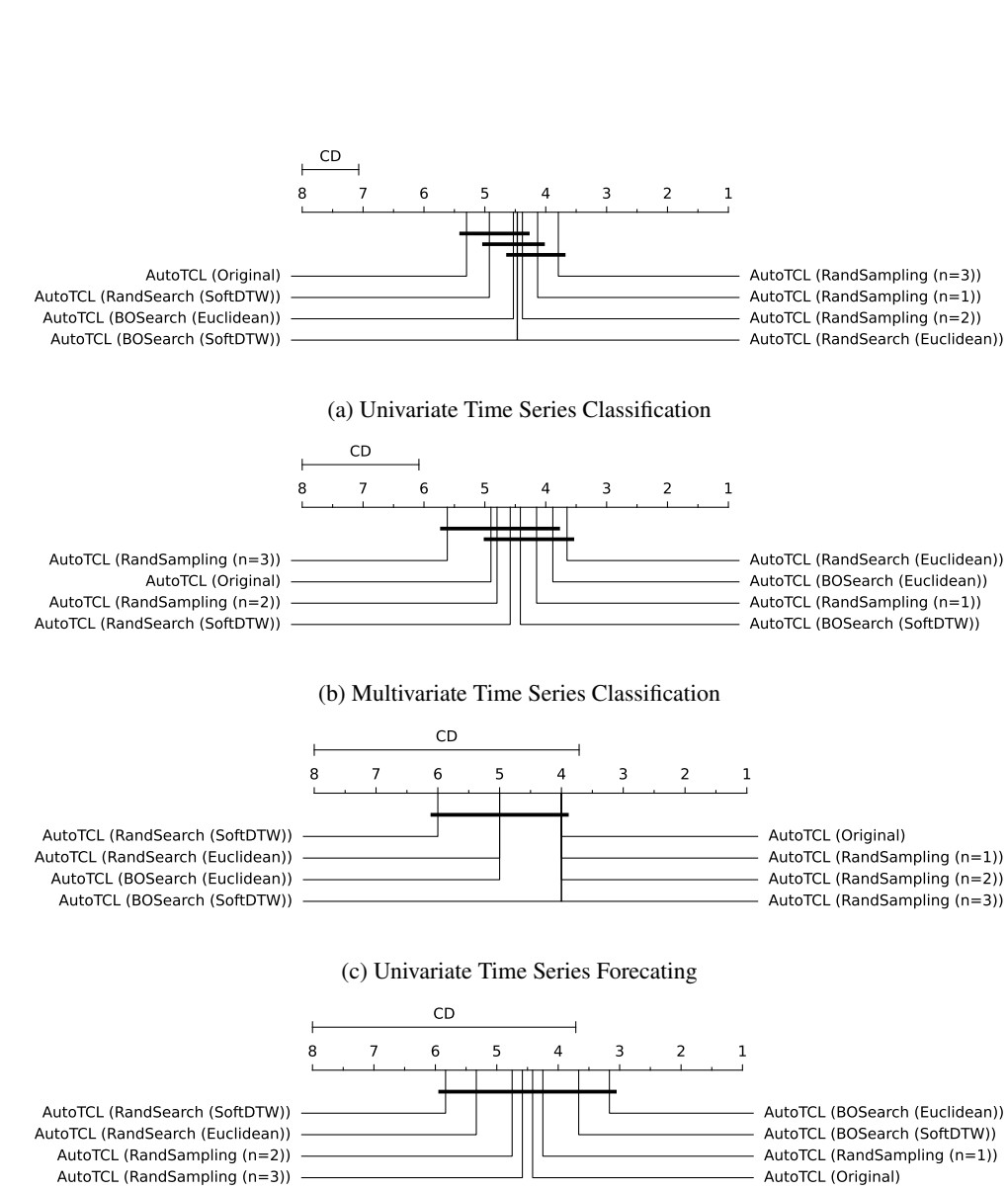

(a) Univariate Time Series Classification

(b) Multivariate Time Series Classification

(c) Univariate Time Series Forecating

(d) Multivariate Time Series Forecating

Figure 8: **CD Diagrams for AutoTCL (Zheng et al., 2024).** These CD diagrams illustrate the ranking of variants of the AutoTCL model when using its original augmentation methods compared to variants employing the AutoTSAugment framework. The comparisons were conducted across various time series classification downstream tasks, including univariate and multivariate classifications using UCR and UEA datasets, as well as time series forecasting tasks on ETT (ETTh1, ETTh2, ETTm1, and ETTm2), Electricity Load, and Weather datasets. The rankings were determined using a Friedman test followed by a Nemenyi posthoc test, highlighting statistically significant differences between the methods.

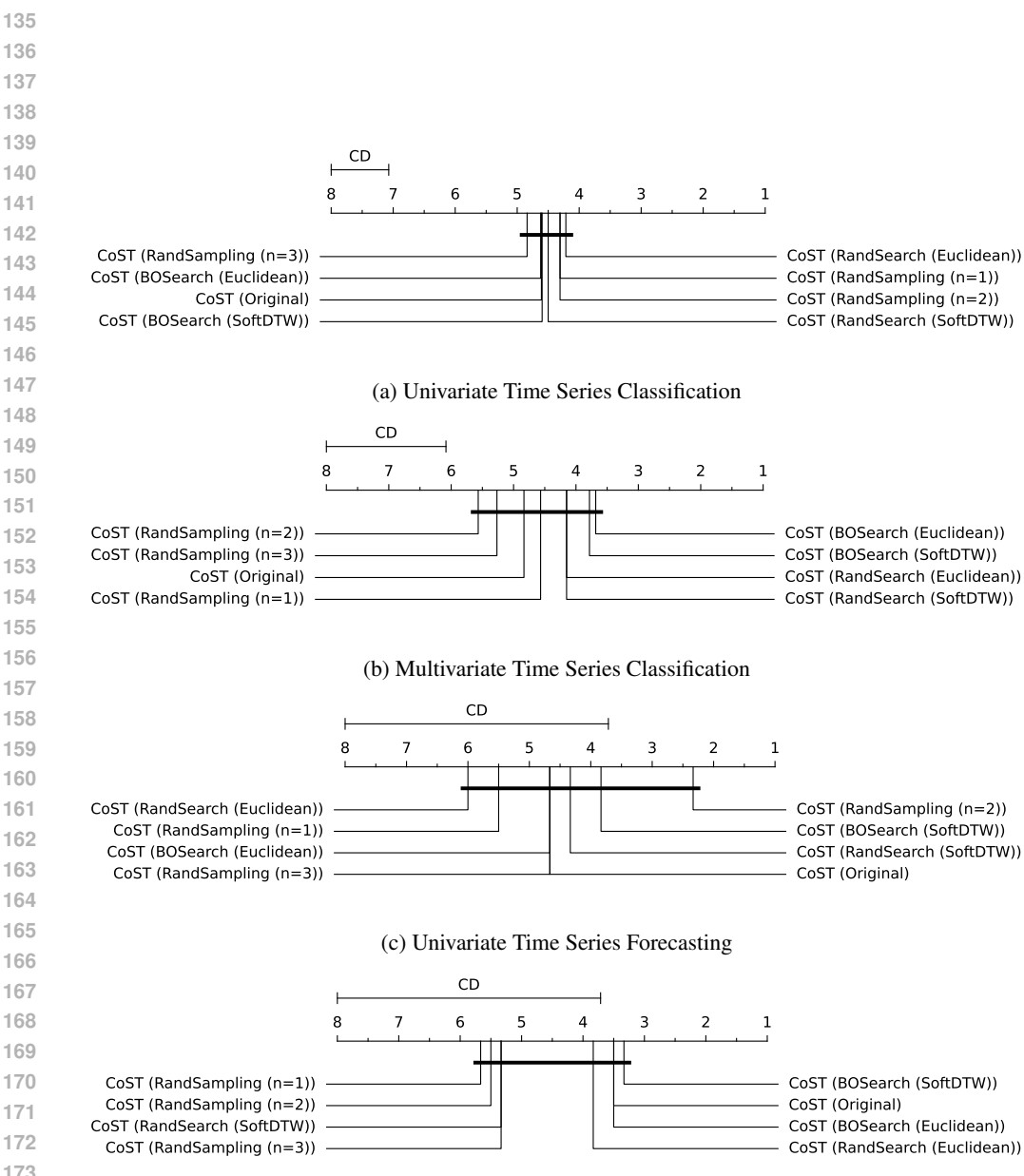

(a) Univariate Time Series Classification

(b) Multivariate Time Series Classification

(c) Univariate Time Series Forecasting

(d) Multivariate Time Series Forecasting

Figure 9: **CD Diagrams for CoST (Woo et al., 2022).** These CD diagrams illustrate the ranking of variants of the CoST model when using its original augmentation methods compared to variants employing the AutoTSAugment framework. The comparisons were conducted across various time series classification downstream tasks, including univariate and multivariate classifications using UCR and UEA datasets, as well as time series forecasting tasks on ETT (ETTh1, ETTh2, ETTm1, and ETTm2), Electricity Load, and Weather datasets. The rankings were determined using a Friedman test followed by a Nemenyi posthoc test, highlighting statistically significant differences between the methods.

Table 4: Detailed results of the multivariate time series classification task for TEST (Sun et al., 2024). Each row corresponds to a specific dataset, each column represents an augmentation method, and each cell contains the classification accuracy for the respective dataset and augmentation method.

| Augmentation Method
Dataset | Original | AutoTSAugment
(RandSampling) |
|---|---|---|
| ArticularyWordRecognition | 0.04 | 0.04 |
| AtrialFibrillation | 0.33 | 0.33 |
| BasicMotions | 0.25 | 0.25 |
| CharacterTrajectories | 0.05 | 0.04 |
| Cricket | 0.08 | 0.08 |
| DuckDuckGeese | 0.20 | 0.20 |
| ERing | 0.17 | 0.17 |
| EigenWorms | 0.42 | 0.42 |
| Epilepsy | 0.27 | 0.27 |
| EthanolConcentration | 0.25 | 0.25 |
| FaceDetection | 0.50 | 0.50 |
| FingerMovements | 0.50 | 0.50 |
| HandMovementDirection | 0.41 | 0.20 |
| Handwriting | 0.05 | 0.03 |
| Heartbeat | 0.50 | 0.50 |
| InsectWingbeat | 0.10 | 0.10 |
| JapaneseVowels | 0.09 | 0.08 |
| LSST | 0.32 | 0.32 |
| Libras | 0.07 | 0.07 |
| MotorImagery | 0.50 | 0.50 |
| NATOPS | 0.17 | 0.17 |
| PEMS-SF | 0.12 | 0.13 |
| PenDigits | 0.10 | 0.10 |
| PhonemeSpectra | 0.03 | 0.03 |
| RacketSports | 0.28 | 0.22 |
| SelfRegulationSCP1 | 0.50 | 0.50 |
| SelfRegulationSCP2 | 0.50 | 0.50 |
| SpokenArabicDigits | 0.10 | 0.10 |
| StandWalkJump | 0.33 | 0.33 |
| UWaveGestureLibrary | 0.12 | 0.12 |

Table 5: Detailed results of the univariate time series classification task across all conducted experiments. Each row corresponds to a specific dataset, each column represents an experiment ID, and each cell contains the classification accuracy for the respective dataset and experiment.

| ExpID
Dataset | b1 | b2 | b3 | r1-1 | r1-2 | r1-3 | r2-1 | r2-2 | r2-3 | r3-1 | r3-2 | r3-3 | s1 | s11 | s2 | s21 | s3 | s31 | rs1 | rs11 | rs2 | rs21 | rs3 | rs31 |
|---|---|---|---|---|---|---|---|---|---|---|---|---|---|---|---|---|---|---|---|---|---|---|---|---|
| ACSF | 0.61 | 0.62 | 0.54 | 0.61 | 0.66 | 0.56 | 0.62 | 0.49 | 0.49 | 0.59 | 0.63 | 0.51 | 0.65 | 0.65 | 0.70 | 0.59 | 0.59 | 0.60 | 0.62 | 0.64 | 0.58 | 0.63 | 0.52 | 0.50 |
| Adiac | 0.70 | 0.34 | 0.76 | 0.64 | 0.76 | 0.79 | 0.45 | 0.44 | 0.51 | 0.74 | 0.74 | 0.73 | 0.77 | 0.79 | 0.63 | 0.40 | 0.77 | 0.78 | 0.78 | 0.76 | 0.70 | 0.50 | 0.77 | 0.79 |
| All
Gesture
Wiimote | 0.67 | 0.10 | 0.37 | 0.49 | 0.50 | 0.46 | 0.10 | 0.10 | 0.10 | 0.40 | 0.40 | 0.41 | 0.48 | 0.48 | 0.10 | 0.10 | 0.42 | 0.44 | 0.52 | 0.44 | 0.10 | 0.10 | 0.43 | 0.39 |
| All
Gesture
Wiimote | 0.70 | 0.10 | 0.38 | 0.56 | 0.57 | 0.51 | 0.10 | 0.10 | 0.10 | 0.39 | 0.40 | 0.37 | 0.50 | 0.54 | 0.10 | 0.10 | 0.41 | 0.43 | 0.51 | 0.46 | 0.10 | 0.10 | 0.42 | 0.42 |
| All
Gesture
Wiimote | 0.59 | 0.10 | 0.31 | 0.49 | 0.45 | 0.34 | 0.10 | 0.10 | 0.10 | 0.37 | 0.36 | 0.38 | 0.44 | 0.48 | 0.10 | 0.10 | 0.32 | 0.34 | 0.50 | 0.41 | 0.10 | 0.10 | 0.38 | 0.38 |
| Arrow
Head | 0.91 | 0.63 | 0.89 | 0.89 | 0.89 | 0.87 | 0.75 | 0.85 | 0.65 | 0.85 | 0.82 | 0.84 | 0.88 | 0.88 | 0.77 | 0.75 | 0.84 | 0.86 | 0.86 | 0.85 | 0.65 | 0.76 | 0.86 | 0.86 |
| BME | 0.92 | 0.50 | 0.96 | 1.00 | 0.99 | 0.99 | 0.84 | 0.86 | 0.88 | 0.98 | 0.98 | 0.98 | 0.98 | 0.99 | 0.72 | 0.84 | 0.98 | 0.98 | 0.99 | 0.99 | 0.79 | 0.79 | 0.99 | 0.98 |
| Beef | 0.47 | 0.47 | 0.53 | 0.67 | 0.67 | 0.60 | 0.40 | 0.40 | 0.40 | 0.67 | 0.67 | 0.57 | 0.67 | 0.57 | 0.50 | 0.40 | 0.70 | 0.53 | 0.63 | 0.63 | 0.50 | 0.47 | 0.70 | 0.73 |

Table 5 – *Continued from previous page*

| | b1 | b2 | b3 | r1-1 | r1-2 | r1-3 | r2-1 | r2-2 | r2-3 | r3-1 | r3-2 | r3-3 | s1 | s11 | s2 | s21 | s3 | s31 | rs1 | rs11 | rs2 | rs21 | rs3 | rs31 |
|---|---|---|---|---|---|---|---|---|---|---|---|---|---|---|---|---|---|---|---|---|---|---|---|---|
| Beetle Fly | 0.50 | 0.55 | 0.85 | 0.75 | 0.65 | 0.75 | 0.65 | 0.65 | 0.70 | 0.95 | 0.90 | 0.85 | 0.85 | 0.90 | 0.65 | 0.80 | 0.90 | 0.50 | 0.50 | 0.50 | 0.50 | 0.65 | 0.80 | 0.50 |
| Bird Chicken | 0.75 | 0.55 | 0.60 | 0.70 | 0.65 | 0.65 | 0.65 | 0.50 | 0.70 | 0.60 | 0.70 | 0.65 | 0.80 | 0.70 | 0.50 | 0.50 | 0.60 | 0.60 | 0.75 | 0.70 | 0.50 | 0.55 | 0.60 | 0.70 |
| CBF | 0.98 | 0.91 | 1.00 | 1.00 | 1.00 | 1.00 | 1.00 | 1.00 | 0.92 | 1.00 | 1.00 | 1.00 | 1.00 | 1.00 | 0.99 | 1.00 | 1.00 | 1.00 | 1.00 | 1.00 | 1.00 | 0.99 | 1.00 | 0.99 |
| Car | 0.63 | 0.45 | 0.58 | 0.60 | 0.73 | 0.60 | 0.50 | 0.47 | 0.48 | 0.62 | 0.62 | 0.58 | 0.53 | 0.53 | 0.42 | 0.47 | 0.58 | 0.62 | 0.57 | 0.25 | 0.47 | 0.42 | 0.60 | 0.60 |
| Chinatown | 0.96 | 0.96 | 0.97 | 0.97 | 0.98 | 0.97 | 0.96 | 0.88 | 0.95 | 0.97 | 0.97 | 0.97 | 0.98 | 0.97 | 0.95 | 0.98 | 0.97 | 0.97 | 0.96 | 0.97 | 0.95 | 0.96 | 0.97 | 0.97 |
| Chlorine Concentration | 0.99 | 0.87 | 1.00 | 0.97 | 0.99 | 1.00 | 0.68 | 0.76 | 0.73 | 1.00 | 1.00 | 1.00 | 1.00 | 1.00 | 0.72 | 0.71 | 1.00 | 1.00 | 1.00 | 1.00 | 0.74 | 0.74 | 1.00 | 1.00 |
| Cin CECG Torso | 0.99 | 0.90 | 1.00 | 0.99 | 1.00 | 1.00 | 0.81 | 0.84 | 0.89 | 0.99 | 0.99 | 0.99 | 1.00 | 0.99 | 0.66 | 0.79 | 0.99 | 0.99 | 0.99 | 0.99 | 0.86 | 0.79 | 0.99 | 0.99 |
| Coffee | 0.86 | 0.43 | 0.96 | 0.96 | 0.93 | 0.89 | 0.64 | 0.75 | 0.50 | 1.00 | 0.96 | 1.00 | 0.89 | 0.89 | 0.50 | 0.50 | 0.96 | 0.96 | 0.89 | 0.89 | 0.43 | 0.50 | 1.00 | 0.96 |
| Computers | 0.62 | 0.52 | 0.56 | 0.60 | 0.57 | 0.52 | 0.62 | 0.55 | 0.62 | 0.56 | 0.56 | 0.56 | 0.57 | 0.59 | 0.65 | 0.64 | 0.58 | 0.56 | 0.59 | 0.58 | 0.66 | 0.57 | 0.56 | 0.59 |
| Cricket | 0.68 | 0.41 | 0.62 | 0.61 | 0.66 | 0.62 | 0.33 | 0.41 | 0.39 | 0.62 | 0.60 | 0.61 | 0.61 | 0.62 | 0.34 | 0.29 | 0.57 | 0.58 | 0.65 | 0.66 | 0.26 | 0.33 | 0.58 | 0.58 |
| Cricket | 0.68 | 0.48 | 0.56 | 0.61 | 0.62 | 0.61 | 0.44 | 0.42 | 0.31 | 0.56 | 0.44 | 0.45 | 0.63 | 0.65 | 0.47 | 0.37 | 0.43 | 0.37 | 0.60 | 0.62 | 0.31 | 0.32 | 0.45 | 0.46 |
| Cricket | 0.73 | 0.33 | 0.64 | 0.67 | 0.67 | 0.64 | 0.39 | 0.33 | 0.37 | 0.62 | 0.61 | 0.55 | 0.63 | 0.63 | 0.35 | 0.28 | 0.59 | 0.58 | 0.60 | 0.64 | 0.23 | 0.32 | 0.60 | 0.57 |
| Crop | 0.74 | 0.73 | 0.77 | 0.75 | 0.74 | 0.75 | 0.67 | 0.69 | 0.68 | 0.77 | 0.77 | 0.77 | 0.75 | 0.75 | 0.65 | 0.68 | 0.79 | 0.78 | 0.76 | 0.75 | 0.66 | 0.66 | 0.78 | 0.77 |
| Diatom Size Reduction | 0.96 | 0.93 | 0.97 | 0.97 | 0.97 | 1.00 | 0.95 | 0.94 | 0.98 | 1.00 | 1.00 | 0.96 | 0.97 | 0.97 | 0.96 | 0.96 | 1.00 | 1.00 | 0.97 | 0.97 | 0.96 | 0.84 | 1.00 | 1.00 |
| Distal Phalanx Outline Age Group | 0.77 | 0.67 | 0.78 | 0.80 | 0.77 | 0.80 | 0.81 | 0.76 | 0.77 | 0.81 | 0.79 | 0.81 | 0.79 | 0.78 | 0.79 | 0.80 | 0.83 | 0.82 | 0.76 | 0.77 | 0.75 | 0.79 | 0.83 | 0.80 |
| Distal Phalanx Outline Correct | 0.83 | 0.75 | 0.79 | 0.79 | 0.81 | 0.83 | 0.78 | 0.80 | 0.80 | 0.80 | 0.81 | 0.81 | 0.81 | 0.81 | 0.76 | 0.79 | 0.80 | 0.80 | 0.80 | 0.81 | 0.78 | 0.80 | 0.80 | 0.80 |
| Distal Phalanx TW | 0.73 | 0.71 | 0.74 | 0.76 | 0.75 | 0.75 | 0.72 | 0.70 | 0.71 | 0.73 | 0.76 | 0.73 | 0.71 | 0.72 | 0.71 | 0.75 | 0.74 | 0.76 | 0.72 | 0.70 | 0.76 | 0.70 | 0.76 | 0.74 |
| Dodger Loop Day | 0.43 | 0.15 | 0.47 | 0.57 | 0.53 | 0.56 | 0.15 | 0.15 | 0.15 | 0.53 | 0.62 | 0.52 | 0.52 | 0.58 | 0.15 | 0.15 | 0.58 | 0.61 | 0.56 | 0.59 | 0.15 | 0.15 | 0.62 | 0.65 |
| Dodger Loop Game | 0.72 | 0.52 | 0.89 | 0.90 | 0.85 | 0.80 | 0.52 | 0.52 | 0.52 | 0.84 | 0.84 | 0.77 | 0.85 | 0.85 | 0.52 | 0.52 | 0.84 | 0.80 | 0.86 | 0.85 | 0.52 | 0.52 | 0.84 | 0.82 |
| Dodger Loop Weekend | 0.92 | 0.71 | 1.00 | 0.99 | 0.99 | 1.00 | 0.71 | 0.71 | 0.71 | 0.97 | 0.99 | 0.99 | 0.97 | 0.96 | 0.71 | 0.71 | 0.97 | 1.00 | 0.99 | 0.97 | 0.71 | 0.71 | 0.99 | 0.99 |
| ECG | 0.83 | 0.82 | 0.91 | 0.86 | 0.87 | 0.87 | 0.77 | 0.75 | 0.75 | 0.89 | 0.86 | 0.88 | 0.88 | 0.88 | 0.84 | 0.81 | 0.86 | 0.89 | 0.87 | 0.91 | 0.82 | 0.80 | 0.89 | 0.89 |
| ECG | 0.94 | 0.93 | 0.96 | 0.95 | 0.95 | 0.95 | 0.94 | 0.92 | 0.95 | 0.96 | 0.95 | 0.95 | 0.95 | 0.95 | 0.95 | 0.95 | 0.96 | 0.95 | 0.95 | 0.95 | 0.95 | 0.93 | 0.96 | 0.95 |
| ECG Five Days | 0.98 | 0.90 | 0.99 | 0.98 | 0.99 | 1.00 | 0.90 | 0.88 | 0.91 | 1.00 | 1.00 | 0.96 | 0.99 | 0.99 | 0.90 | 0.94 | 0.99 | 0.99 | 0.99 | 0.98 | 0.93 | 0.92 | 0.99 | 1.00 |
| EOG Horizontal Signal | 0.54 | 0.61 | 0.60 | 0.65 | 0.63 | 0.47 | 0.49 | 0.42 | 0.51 | 0.62 | 0.60 | 0.60 | 0.67 | 0.57 | 0.51 | 0.44 | 0.60 | 0.60 | 0.59 | 0.54 | 0.52 | 0.54 | 0.60 | 0.63 |
| EOG Vertical Signal | 0.42 | 0.21 | 0.57 | 0.55 | 0.60 | 0.49 | 0.35 | 0.38 | 0.44 | 0.56 | 0.58 | 0.55 | 0.60 | 0.52 | 0.42 | 0.33 | 0.56 | 0.54 | 0.60 | 0.47 | 0.42 | 0.41 | 0.53 | 0.54 |
| Earthquakes | 0.80 | 0.80 | 0.80 | 0.81 | 0.80 | 0.80 | 0.80 | 0.80 | 0.71 | 0.80 | 0.80 | 0.80 | 0.80 | 0.80 | 0.80 | 0.80 | 0.80 | 0.80 | 0.80 | 0.79 | 0.80 | 0.80 | 0.80 | 0.80 |
| Electric Devices | 0.80 | 0.66 | 0.73 | 0.74 | 0.73 | 0.73 | 0.76 | 0.71 | 0.70 | 0.73 | 0.74 | 0.73 | 0.72 | 0.75 | 0.70 | 0.66 | 0.73 | 0.74 | 0.73 | 0.74 | 0.69 | 0.70 | 0.74 | 0.73 |
| Ethanol Level | 0.38 | 0.31 | 0.72 | 0.43 | 0.72 | 0.61 | 0.31 | 0.31 | 0.34 | 0.69 | 0.66 | 0.63 | 0.42 | 0.52 | 0.31 | 0.29 | 0.65 | 0.63 | 0.47 | 0.57 | 0.31 | 0.31 | 0.65 | 0.64 |
| Face All | 0.91 | 0.72 | 0.94 | 0.94 | 0.95 | 0.94 | 0.67 | 0.53 | 0.83 | 0.95 | 0.95 | 0.95 | 0.93 | 0.94 | 0.89 | 0.81 | 0.94 | 0.93 | 0.93 | 0.93 | 0.50 | 0.70 | 0.95 | 0.94 |
| Face Four | 0.88 | 0.70 | 0.89 | 0.91 | 0.91 | 0.93 | 0.73 | 0.86 | 0.73 | 0.89 | 0.89 | 0.91 | 0.95 | 0.93 | 0.61 | 0.88 | 0.89 | 0.91 | 0.95 | 0.93 | 0.77 | 0.73 | 0.91 | 0.88 |
| Faces UCR | 0.92 | 0.72 | 0.94 | 0.95 | 0.95 | 0.95 | 0.74 | 0.49 | 0.71 | 0.94 | 0.94 | 0.95 | 0.93 | 0.93 | 0.67 | 0.59 | 0.92 | 0.91 | 0.94 | 0.95 | 0.69 | 0.64 | 0.93 | 0.94 |
| Fifty Words | 0.51 | 0.50 | 0.71 | 0.70 | 0.71 | 0.72 | 0.49 | 0.49 | 0.36 | 0.70 | 0.69 | 0.71 | 0.69 | 0.72 | 0.29 | 0.39 | 0.70 | 0.69 | 0.70 | 0.69 | 0.43 | 0.43 | 0.70 | 0.68 |
| Fish | 0.81 | 0.59 | 0.87 | 0.86 | 0.80 | 0.87 | 0.39 | 0.57 | 0.51 | 0.86 | 0.89 | 0.78 | 0.85 | 0.88 | 0.51 | 0.54 | 0.87 | 0.88 | 0.85 | 0.87 | 0.37 | 0.30 | 0.89 | 0.89 |
| Ford | 0.89 | 0.65 | 0.81 | 0.85 | 0.79 | 0.80 | 0.84 | 0.65 | 0.70 | 0.75 | 0.78 | 0.77 | 0.86 | 0.79 | 0.67 | 0.61 | 0.71 | 0.71 | 0.86 | 0.79 | 0.85 | 0.55 | 0.79 | 0.77 |
| Ford | 0.88 | 0.62 | 0.78 | 0.79 | 0.75 | 0.77 | 0.73 | 0.64 | 0.73 | 0.70 | 0.70 | 0.74 | 0.86 | 0.76 | 0.83 | 0.71 | 0.65 | 0.66 | 0.84 | 0.77 | 0.80 | 0.58 | 0.75 | 0.75 |
| Freezer Regular Train | 0.94 | 0.85 | 0.99 | 0.98 | 0.99 | 1.00 | 0.97 | 0.98 | 0.99 | 0.99 | 0.99 | 0.99 | 0.99 | 0.99 | 0.95 | 0.92 | 0.99 | 0.99 | 0.99 | 0.99 | 0.98 | 0.98 | 0.99 | 0.99 |
| Freezer Small Train | 0.95 | 0.93 | 0.99 | 0.99 | 0.99 | 1.00 | 0.98 | 0.94 | 0.93 | 1.00 | 0.99 | 0.99 | 0.99 | 0.98 | 0.98 | 0.97 | 1.00 | 0.99 | 0.99 | 0.99 | 0.98 | 0.96 | 1.00 | 0.99 |

Table 5 – *Continued from previous page*

| | b1 | b2 | b3 | r1-1 | r1-2 | r1-3 | r2-1 | r2-2 | r2-3 | r3-1 | r3-2 | r3-3 | s1 | s11 | s2 | s21 | s3 | s31 | rs1 | rs11 | rs2 | rs21 | rs3 | rs31 |
|---|---|---|---|---|---|---|---|---|---|---|---|---|---|---|---|---|---|---|---|---|---|---|---|---|
| Fungi | 0.94 | 0.40 | 0.96 | 0.97 | 0.97 | 0.97 | 0.72 | 0.52 | 0.75 | 0.99 | 0.99 | 0.98 | 0.96 | 0.96 | 0.87 | 0.39 | 0.85 | 0.99 | 0.95 | 0.95 | 0.56 | 0.79 | 0.98 | 0.98 |
| Gesture Mid Air | 0.65 | 0.04 | 0.58 | 0.57 | 0.52 | 0.53 | 0.04 | 0.04 | 0.04 | 0.56 | 0.46 | 0.55 | 0.56 | 0.55 | 0.04 | 0.04 | 0.55 | 0.56 | 0.56 | 0.55 | 0.04 | 0.04 | 0.47 | 0.51 |
| Gesture Mid Air | 0.44 | 0.04 | 0.53 | 0.45 | 0.50 | 0.52 | 0.04 | 0.04 | 0.04 | 0.45 | 0.49 | 0.46 | 0.44 | 0.50 | 0.04 | 0.04 | 0.52 | 0.51 | 0.46 | 0.49 | 0.04 | 0.04 | 0.47 | 0.54 |
| Gesture Mid Air | 0.22 | 0.04 | 0.37 | 0.27 | 0.29 | 0.27 | 0.04 | 0.04 | 0.04 | 0.34 | 0.37 | 0.34 | 0.30 | 0.30 | 0.04 | 0.04 | 0.30 | 0.32 | 0.28 | 0.31 | 0.04 | 0.04 | 0.25 | 0.36 |
| Gesture Pebble | 0.87 | 0.17 | 0.84 | 0.93 | 0.90 | 0.81 | 0.18 | 0.18 | 0.18 | 0.81 | 0.86 | 0.82 | 0.90 | 0.80 | 0.18 | 0.18 | 0.84 | 0.87 | 0.95 | 0.91 | 0.18 | 0.18 | 0.85 | 0.87 |
| Gesture Pebble | 0.93 | 0.17 | 0.93 | 0.94 | 0.93 | 0.77 | 0.18 | 0.18 | 0.18 | 0.93 | 0.91 | 0.85 | 0.93 | 0.88 | 0.18 | 0.18 | 0.91 | 0.89 | 0.95 | 0.89 | 0.18 | 0.18 | 0.90 | 0.91 |
| Gun Point | 0.97 | 0.70 | 0.95 | 0.96 | 0.95 | 0.94 | 0.89 | 0.91 | 0.92 | 0.95 | 0.97 | 0.96 | 0.97 | 0.98 | 0.97 | 0.91 | 0.99 | 0.95 | 0.96 | 0.96 | 0.88 | 0.94 | 0.96 | 0.98 |
| Gun Point Age Span | 0.93 | 0.93 | 0.99 | 0.98 | 0.92 | 0.96 | 0.86 | 0.96 | 0.92 | 0.92 | 0.92 | 0.92 | 0.92 | 0.92 | 0.85 | 0.86 | 0.92 | 0.91 | 0.98 | 0.93 | 0.83 | 0.82 | 0.91 | 0.92 |
| Gun Point Male Versus Female | 1.00 | 0.97 | 1.00 | 1.00 | 1.00 | 1.00 | 0.96 | 0.99 | 0.99 | 0.99 | 0.96 | 0.98 | 1.00 | 1.00 | 0.98 | 0.98 | 0.96 | 0.99 | 1.00 | 1.00 | 0.94 | 0.97 | 1.00 | 1.00 |
| Gun Point Old Versus Young | 1.00 | 1.00 | 1.00 | 1.00 | 1.00 | 1.00 | 1.00 | 1.00 | 0.99 | 1.00 | 1.00 | 1.00 | 1.00 | 1.00 | 1.00 | 1.00 | 1.00 | 1.00 | 1.00 | 1.00 | 1.00 | 1.00 | 1.00 | 1.00 |
| Ham | 0.64 | 0.51 | 0.82 | 0.81 | 0.73 | 0.81 | 0.56 | 0.51 | 0.62 | 0.81 | 0.83 | 0.81 | 0.79 | 0.76 | 0.53 | 0.56 | 0.82 | 0.80 | 0.81 | 0.81 | 0.55 | 0.58 | 0.82 | 0.79 |
| Hand Outlines | 0.89 | 0.82 | 0.88 | 0.91 | 0.91 | 0.90 | 0.73 | 0.88 | 0.86 | 0.89 | 0.87 | 0.90 | 0.90 | 0.91 | 0.88 | 0.89 | 0.89 | 0.87 | 0.87 | 0.91 | 0.82 | 0.90 | 0.85 | 0.91 |
| Haptics | 0.41 | 0.34 | 0.43 | 0.41 | 0.47 | 0.45 | 0.39 | 0.37 | 0.38 | 0.45 | 0.44 | 0.45 | 0.44 | 0.41 | 0.24 | 0.26 | 0.45 | 0.45 | 0.39 | 0.43 | 0.28 | 0.31 | 0.45 | 0.42 |
| Herring | 0.61 | 0.59 | 0.61 | 0.61 | 0.61 | 0.61 | 0.61 | 0.61 | 0.61 | 0.61 | 0.61 | 0.61 | 0.61 | 0.61 | 0.61 | 0.61 | 0.61 | 0.61 | 0.61 | 0.61 | 0.61 | 0.61 | 0.61 | 0.61 |
| House Twenty | 0.96 | 0.89 | 0.72 | 0.81 | 0.80 | 0.81 | 0.62 | 0.75 | 0.89 | 0.71 | 0.72 | 0.70 | 0.93 | 0.88 | 0.70 | 0.56 | 0.75 | 0.72 | 0.91 | 0.84 | 0.90 | 0.81 | 0.74 | 0.71 |
| Inline Skate | 0.41 | 0.29 | 0.43 | 0.45 | 0.48 | 0.50 | 0.23 | 0.29 | 0.32 | 0.46 | 0.47 | 0.46 | 0.46 | 0.46 | 0.32 | 0.26 | 0.48 | 0.46 | 0.44 | 0.46 | 0.31 | 0.28 | 0.46 | 0.53 |
| Insect EPG Regular Train | 1.00 | 1.00 | 1.00 | 1.00 | 1.00 | 1.00 | 1.00 | 1.00 | 1.00 | 1.00 | 1.00 | 1.00 | 1.00 | 1.00 | 1.00 | 1.00 | 1.00 | 1.00 | 1.00 | 1.00 | 1.00 | 1.00 | 1.00 | 1.00 |
| Insect EPG Small Train | 1.00 | 1.00 | 1.00 | 1.00 | 1.00 | 1.00 | 1.00 | 1.00 | 1.00 | 1.00 | 1.00 | 1.00 | 1.00 | 1.00 | 1.00 | 1.00 | 1.00 | 1.00 | 1.00 | 1.00 | 1.00 | 1.00 | 1.00 | 1.00 |
| Insect Wingbeat Sound | 0.60 | 0.46 | 0.67 | 0.68 | 0.67 | 0.69 | 0.58 | 0.44 | 0.61 | 0.67 | 0.68 | 0.63 | 0.68 | 0.67 | 0.63 | 0.50 | 0.67 | 0.67 | 0.67 | 0.69 | 0.50 | 0.42 | 0.66 | 0.67 |
| Italy Power Demand | 0.96 | 0.96 | 0.96 | 0.96 | 0.95 | 0.96 | 0.93 | 0.94 | 0.95 | 0.95 | 0.97 | 0.97 | 0.95 | 0.95 | 0.95 | 0.95 | 0.97 | 0.97 | 0.95 | 0.95 | 0.95 | 0.94 | 0.50 | 0.97 |
| Large Kitchen Appliances | 0.72 | 0.54 | 0.50 | 0.48 | 0.48 | 0.46 | 0.57 | 0.60 | 0.50 | 0.50 | 0.49 | 0.53 | 0.58 | 0.50 | 0.72 | 0.57 | 0.50 | 0.49 | 0.51 | 0.49 | 0.73 | 0.58 | 0.47 | 0.50 |
| Lightning | 0.67 | 0.67 | 0.62 | 0.69 | 0.74 | 0.74 | 0.64 | 0.67 | 0.59 | 0.72 | 0.66 | 0.70 | 0.67 | 0.70 | 0.70 | 0.69 | 0.64 | 0.69 | 0.70 | 0.66 | 0.64 | 0.64 | 0.62 | 0.69 |
| Lightning | 0.69 | 0.50 | 0.69 | 0.65 | 0.62 | 0.61 | 0.44 | 0.47 | 0.58 | 0.65 | 0.71 | 0.68 | 0.67 | 0.65 | 0.39 | 0.26 | 0.62 | 0.68 | 0.69 | 0.65 | 0.35 | 0.43 | 0.68 | 0.71 |
| Mallat | 1.00 | 0.94 | 0.98 | 0.99 | 0.98 | 0.99 | 1.00 | 0.97 | 0.95 | 0.98 | 0.97 | 0.98 | 0.98 | 0.98 | 0.95 | 0.94 | 0.98 | 0.98 | 0.98 | 0.99 | 0.96 | 0.95 | 0.98 | 0.98 |
| Meat | 0.98 | 0.98 | 0.98 | 0.98 | 0.98 | 0.98 | 0.93 | 0.97 | 0.97 | 0.97 | 0.98 | 0.95 | 0.98 | 0.98 | 0.90 | 0.87 | 0.97 | 0.98 | 0.33 | 0.98 | 0.78 | 0.90 | 0.98 | 0.98 |
| Medical Images | 0.68 | 0.65 | 0.75 | 0.76 | 0.71 | 0.77 | 0.70 | 0.61 | 0.68 | 0.74 | 0.74 | 0.73 | 0.74 | 0.74 | 0.69 | 0.65 | 0.73 | 0.75 | 0.73 | 0.71 | 0.64 | 0.64 | 0.77 | 0.74 |
| Melbourne Pedestrian | 0.96 | 0.10 | 0.94 | 0.96 | 0.96 | 0.96 | 0.10 | 0.10 | 0.10 | 0.96 | 0.95 | 0.94 | 0.95 | 0.95 | 0.10 | 0.10 | 0.94 | 0.94 | 0.94 | 0.95 | 0.10 | 0.10 | 0.95 | 0.94 |
| Middle Phalanx Outline Age Group | 0.74 | 0.70 | 0.73 | 0.75 | 0.73 | 0.74 | 0.76 | 0.75 | 0.74 | 0.73 | 0.73 | 0.72 | 0.71 | 0.74 | 0.73 | 0.75 | 0.74 | 0.74 | 0.73 | 0.73 | 0.75 | 0.75 | 0.74 | 0.74 |
| Middle Phalanx Outline Correct | 0.79 | 0.73 | 0.80 | 0.79 | 0.80 | 0.81 | 0.73 | 0.72 | 0.74 | 0.83 | 0.84 | 0.80 | 0.78 | 0.78 | 0.76 | 0.76 | 0.79 | 0.80 | 0.79 | 0.80 | 0.78 | 0.76 | 0.82 | 0.80 |
| Middle Phalanx TW | 0.61 | 0.50 | 0.61 | 0.61 | 0.63 | 0.61 | 0.58 | 0.60 | 0.61 | 0.61 | 0.61 | 0.62 | 0.59 | 0.59 | 0.58 | 0.56 | 0.58 | 0.59 | 0.59 | 0.59 | 0.60 | 0.61 | 0.59 | 0.59 |

Table 5 – *Continued from previous page*

| | b1 | b2 | b3 | r1-1 | r1-2 | r1-3 | r2-1 | r2-2 | r2-3 | r3-1 | r3-2 | r3-3 | s1 | s11 | s2 | s21 | s3 | s31 | rs1 | rs11 | rs2 | rs21 | rs3 | rs31 |
|---|---|---|---|---|---|---|---|---|---|---|---|---|---|---|---|---|---|---|---|---|---|---|---|---|
| Mixed Shapes Regular Train | 0.93 | 0.87 | 0.93 | 0.95 | 0.95 | 0.95 | 0.86 | 0.89 | 0.89 | 0.94 | 0.93 | 0.94 | 0.94 | 0.94 | 0.71 | 0.83 | 0.94 | 0.94 | 0.94 | 0.94 | 0.73 | 0.69 | 0.94 | 0.92 |
| Mixed Shapes Small Train | 0.92 | 0.88 | 0.95 | 0.95 | 0.95 | 0.95 | 0.84 | 0.90 | 0.88 | 0.94 | 0.94 | 0.93 | 0.94 | 0.94 | 0.61 | 0.78 | 0.94 | 0.94 | 0.95 | 0.94 | 0.77 | 0.69 | 0.95 | 0.94 |
| Mote Strain | 0.96 | 0.90 | 0.94 | 0.95 | 0.95 | 0.94 | 0.91 | 0.91 | 0.93 | 0.95 | 0.95 | 0.94 | 0.94 | 0.94 | 0.92 | 0.90 | 0.95 | 0.96 | 0.93 | 0.94 | 0.89 | 0.88 | 0.95 | 0.91 |
| Non Invasive Fetal ECG Thorax | 0.77 | 0.60 | 0.90 | 0.83 | 0.92 | 0.91 | 0.71 | 0.65 | 0.72 | 0.88 | 0.88 | 0.89 | 0.84 | 0.86 | 0.59 | 0.60 | 0.88 | 0.89 | 0.89 | 0.86 | 0.55 | 0.56 | 0.90 | 0.89 |
| Non Invasive Fetal ECG Thorax | 0.78 | 0.61 | 0.89 | 0.86 | 0.89 | 0.94 | 0.62 | 0.61 | 0.78 | 0.91 | 0.92 | 0.91 | 0.92 | 0.89 | 0.69 | 0.59 | 0.92 | 0.91 | 0.90 | 0.89 | 0.67 | 0.59 | 0.91 | 0.91 |
| OSU Leaf | 0.75 | 0.42 | 0.65 | 0.64 | 0.67 | 0.67 | 0.45 | 0.51 | 0.51 | 0.66 | 0.58 | 0.57 | 0.62 | 0.69 | 0.49 | 0.30 | 0.57 | 0.62 | 0.60 | 0.65 | 0.48 | 0.45 | 0.59 | 0.61 |
| Olive Oil | 0.63 | 0.40 | 0.87 | 0.93 | 0.87 | 0.87 | 0.67 | 0.83 | 0.73 | 0.83 | 0.87 | 0.87 | 0.83 | 0.87 | 0.57 | 0.63 | 0.87 | 0.87 | 0.83 | 0.87 | 0.80 | 0.50 | 0.87 | 0.87 |
| PLAID | 0.48 | 0.16 | 0.42 | 0.42 | 0.36 | 0.45 | 0.16 | 0.16 | 0.16 | 0.42 | 0.45 | 0.44 | 0.47 | 0.38 | 0.16 | 0.16 | 0.45 | 0.46 | 0.43 | 0.45 | 0.16 | 0.16 | 0.47 | 0.46 |
| Phalanges Outlines Correct | 0.81 | 0.71 | 0.80 | 0.81 | 0.82 | 0.83 | 0.80 | 0.77 | 0.80 | 0.83 | 0.82 | 0.82 | 0.80 | 0.82 | 0.80 | 0.80 | 0.81 | 0.83 | 0.80 | 0.81 | 0.78 | 0.82 | 0.81 | 0.83 |
| Phoneme | 0.32 | 0.15 | 0.20 | 0.19 | 0.19 | 0.19 | 0.22 | 0.17 | 0.19 | 0.14 | 0.15 | 0.16 | 0.21 | 0.19 | 0.21 | 0.20 | 0.14 | 0.13 | 0.25 | 0.22 | 0.19 | 0.21 | 0.16 | 0.16 |
| Pickup Gesture Wiimote | 0.62 | 0.10 | 0.62 | 0.72 | 0.74 | 0.66 | 0.10 | 0.10 | 0.10 | 0.68 | 0.66 | 0.60 | 0.74 | 0.72 | 0.10 | 0.10 | 0.64 | 0.60 | 0.68 | 0.70 | 0.10 | 0.10 | 0.66 | 0.50 |
| Pig Airway Pressure | 0.40 | 0.19 | 0.17 | 0.17 | 0.17 | 0.08 | 0.12 | 0.12 | 0.15 | 0.18 | 0.19 | 0.08 | 0.18 | 0.17 | 0.02 | 0.14 | 0.14 | 0.18 | 0.16 | 0.17 | 0.15 | 0.06 | 0.17 | 0.18 |
| Pig Art Pressure | 0.81 | 0.32 | 0.25 | 0.42 | 0.35 | 0.31 | 0.17 | 0.24 | 0.25 | 0.22 | 0.22 | 0.23 | 0.53 | 0.56 | 0.12 | 0.20 | 0.22 | 0.23 | 0.59 | 0.33 | 0.37 | 0.17 | 0.24 | 0.26 |
| Pig CVP | 0.36 | 0.19 | 0.18 | 0.17 | 0.21 | 0.13 | 0.17 | 0.15 | 0.22 | 0.02 | 0.17 | 0.16 | 0.37 | 0.09 | 0.14 | 0.22 | 0.18 | 0.17 | 0.42 | 0.21 | 0.17 | 0.22 | 0.16 | 0.16 |
| Plane | 0.99 | 0.98 | 0.99 | 0.99 | 0.99 | 0.99 | 0.88 | 0.97 | 0.98 | 0.99 | 1.00 | 0.99 | 0.99 | 0.99 | 0.90 | 0.97 | 0.99 | 0.99 | 0.99 | 0.99 | 0.98 | 0.90 | 0.99 | 1.00 |
| Power Cons | 0.92 | 0.84 | 0.97 | 0.96 | 0.97 | 0.97 | 0.92 | 0.94 | 0.91 | 0.98 | 0.97 | 0.99 | 0.97 | 0.96 | 0.91 | 0.93 | 0.99 | 0.99 | 0.98 | 0.99 | 0.91 | 0.92 | 0.98 | 0.97 |
| Proximal Phalanx Outline Age Group | 0.81 | 0.79 | 0.81 | 0.82 | 0.83 | 0.82 | 0.81 | 0.81 | 0.82 | 0.82 | 0.81 | 0.81 | 0.81 | 0.82 | 0.82 | 0.81 | 0.82 | 0.82 | 0.82 | 0.82 | 0.80 | 0.81 | 0.83 | 0.82 |
| Proximal Phalanx Outline Correct | 0.84 | 0.75 | 0.84 | 0.84 | 0.82 | 0.84 | 0.76 | 0.72 | 0.81 | 0.84 | 0.86 | 0.87 | 0.82 | 0.83 | 0.78 | 0.82 | 0.84 | 0.85 | 0.82 | 0.84 | 0.77 | 0.83 | 0.85 | 0.85 |
| Proximal Phalanx TW | 0.78 | 0.76 | 0.83 | 0.81 | 0.80 | 0.81 | 0.80 | 0.80 | 0.81 | 0.83 | 0.82 | 0.82 | 0.81 | 0.80 | 0.78 | 0.81 | 0.82 | 0.82 | 0.80 | 0.81 | 0.80 | 0.77 | 0.80 | 0.83 |
| Refrigeration Devices | 0.62 | 0.42 | 0.42 | 0.50 | 0.42 | 0.50 | 0.49 | 0.46 | 0.40 | 0.42 | 0.39 | 0.40 | 0.59 | 0.47 | 0.54 | 0.62 | 0.39 | 0.38 | 0.58 | 0.44 | 0.55 | 0.56 | 0.37 | 0.39 |
| Rock | 0.71 | 0.54 | 0.71 | 0.69 | 0.71 | 0.74 | 0.71 | 0.74 | 0.63 | 0.69 | 0.80 | 0.77 | 0.71 | 0.66 | 0.57 | 0.63 | 0.77 | 0.77 | 0.66 | 0.69 | 0.60 | 0.69 | 0.74 | 0.77 |
| Screen Type | 0.47 | 0.34 | 0.46 | 0.45 | 0.45 | 0.47 | 0.42 | 0.43 | 0.45 | 0.45 | 0.44 | 0.45 | 0.47 | 0.42 | 0.42 | 0.40 | 0.43 | 0.43 | 0.44 | 0.44 | 0.42 | 0.41 | 0.42 | 0.42 |
| Semg Hand Gender Ch | 0.62 | 0.64 | 0.91 | 0.94 | 0.94 | 0.93 | 0.85 | 0.74 | 0.90 | 0.94 | 0.91 | 0.94 | 0.93 | 0.94 | 0.87 | 0.78 | 0.94 | 0.91 | 0.95 | 0.95 | 0.85 | 0.86 | 0.93 | 0.93 |
| Semg Hand Movement Ch | 0.43 | 0.33 | 0.67 | 0.85 | 0.86 | 0.84 | 0.73 | 0.69 | 0.71 | 0.74 | 0.75 | 0.77 | 0.85 | 0.85 | 0.66 | 0.72 | 0.77 | 0.76 | 0.83 | 0.87 | 0.64 | 0.67 | 0.74 | 0.76 |
| Semg Hand Subject Ch | 0.44 | 0.29 | 0.76 | 0.92 | 0.92 | 0.93 | 0.67 | 0.78 | 0.53 | 0.84 | 0.84 | 0.85 | 0.91 | 0.87 | 0.74 | 0.78 | 0.84 | 0.85 | 0.94 | 0.88 | 0.76 | 0.73 | 0.86 | 0.84 |
| Shake Gesture Wiimote | 0.76 | 0.10 | 0.64 | 0.70 | 0.70 | 0.70 | 0.10 | 0.10 | 0.10 | 0.66 | 0.56 | 0.64 | 0.64 | 0.68 | 0.10 | 0.10 | 0.70 | 0.62 | 0.74 | 0.64 | 0.10 | 0.10 | 0.66 | 0.58 |

Table 5 – *Continued from previous page*

| | b1 | b2 | b3 | r1-1 | r1-2 | r1-3 | r2-1 | r2-2 | r2-3 | r3-1 | r3-2 | r3-3 | s1 | s11 | s2 | s21 | s3 | s31 | rs1 | rs11 | rs2 | rs21 | rs3 | rs31 |
|---|---|---|---|---|---|---|---|---|---|---|---|---|---|---|---|---|---|---|---|---|---|---|---|---|
| Shapelet Sim | 0.99 | 0.54 | 0.45 | 0.50 | 0.50 | 0.50 | 0.50 | 0.50 | 0.50 | 0.57 | 0.50 | 0.50 | 0.64 | 0.77 | 0.48 | 0.50 | 0.50 | 0.50 | 0.50 | 0.50 | 0.58 | 0.47 | 0.50 | 0.50 |
| Shapes All | 0.73 | 0.59 | 0.76 | 0.77 | 0.76 | 0.78 | 0.59 | 0.54 | 0.58 | 0.76 | 0.77 | 0.76 | 0.77 | 0.77 | 0.43 | 0.41 | 0.78 | 0.77 | 0.76 | 0.77 | 0.34 | 0.26 | 0.77 | 0.78 |
| Small Kitchen Appliances | 0.60 | 0.54 | 0.57 | 0.33 | 0.57 | 0.56 | 0.66 | 0.55 | 0.59 | 0.58 | 0.57 | 0.62 | 0.53 | 0.59 | 0.64 | 0.58 | 0.52 | 0.57 | 0.55 | 0.33 | 0.54 | 0.61 | 0.55 | 0.58 |
| Smooth Subspace | 0.95 | 0.93 | 0.92 | 0.94 | 0.92 | 0.92 | 0.97 | 0.95 | 0.96 | 0.92 | 0.91 | 0.91 | 0.91 | 0.91 | 0.95 | 0.96 | 0.91 | 0.92 | 0.93 | 0.93 | 0.96 | 0.94 | 0.90 | 0.92 |
| Sony AIBO Robot Surface | 0.97 | 0.95 | 0.99 | 0.99 | 0.99 | 1.00 | 0.95 | 0.90 | 0.95 | 0.99 | 0.99 | 1.00 | 0.99 | 0.99 | 0.94 | 0.95 | 0.99 | 0.99 | 0.99 | 0.97 | 0.97 | 0.91 | 0.99 | 0.99 |
| Sony AIBO Robot Surface | 0.92 | 0.93 | 0.97 | 0.97 | 0.98 | 0.98 | 0.95 | 0.91 | 0.91 | 0.98 | 0.98 | 0.98 | 0.96 | 0.97 | 0.92 | 0.91 | 0.99 | 0.99 | 0.94 | 0.97 | 0.90 | 0.89 | 0.98 | 0.98 |
| Star Light Curves | 0.97 | 0.92 | 0.97 | 0.97 | 0.97 | 0.97 | 0.93 | 0.93 | 0.95 | 0.97 | 0.96 | 0.97 | 0.96 | 0.96 | 0.94 | 0.94 | 0.97 | 0.97 | 0.97 | 0.97 | 0.95 | 0.94 | 0.97 | 0.96 |
| Strawberry | 0.98 | 0.83 | 0.93 | 0.96 | 0.97 | 0.96 | 0.90 | 0.93 | 0.92 | 0.98 | 0.97 | 0.97 | 0.97 | 0.98 | 0.96 | 0.86 | 0.98 | 0.98 | 0.97 | 0.97 | 0.86 | 0.95 | 0.97 | 0.97 |
| Swedish Leaf | 0.88 | 0.74 | 0.89 | 0.88 | 0.90 | 0.88 | 0.79 | 0.73 | 0.70 | 0.91 | 0.91 | 0.91 | 0.90 | 0.91 | 0.70 | 0.78 | 0.90 | 0.90 | 0.90 | 0.91 | 0.59 | 0.61 | 0.91 | 0.90 |
| Symbols | 0.97 | 0.92 | 0.97 | 0.97 | 0.96 | 0.95 | 0.90 | 0.92 | 0.91 | 0.96 | 0.95 | 0.93 | 0.97 | 0.96 | 0.91 | 0.89 | 0.96 | 0.96 | 0.97 | 0.97 | 0.90 | 0.89 | 0.96 | 0.95 |
| Synthetic Control | 0.98 | 0.80 | 0.97 | 0.98 | 0.99 | 0.99 | 0.96 | 0.95 | 0.94 | 0.97 | 0.97 | 0.97 | 0.99 | 1.00 | 0.97 | 0.97 | 0.96 | 0.97 | 0.99 | 0.99 | 0.95 | 0.96 | 0.98 | 0.98 |
| Toe Segmentation | 0.94 | 0.52 | 0.70 | 0.68 | 0.75 | 0.74 | 0.56 | 0.64 | 0.69 | 0.67 | 0.67 | 0.66 | 0.69 | 0.65 | 0.73 | 0.72 | 0.64 | 0.64 | 0.75 | 0.77 | 0.74 | 0.75 | 0.69 | 0.66 |
| Toe Segmentation | 0.92 | 0.75 | 0.75 | 0.80 | 0.87 | 0.81 | 0.78 | 0.82 | 0.80 | 0.78 | 0.80 | 0.82 | 0.88 | 0.83 | 0.75 | 0.82 | 0.82 | 0.82 | 0.87 | 0.82 | 0.82 | 0.75 | 0.81 | 0.78 |
| Trace | 0.98 | 0.80 | 0.83 | 0.91 | 0.96 | 0.91 | 0.74 | 0.85 | 0.79 | 0.91 | 0.87 | 0.87 | 0.91 | 0.87 | 0.71 | 0.82 | 0.85 | 0.84 | 0.90 | 0.88 | 0.73 | 0.71 | 0.85 | 0.82 |
| Two Lead ECG | 0.99 | 0.92 | 0.99 | 0.97 | 0.97 | 0.98 | 0.94 | 0.93 | 0.90 | 0.99 | 0.99 | 1.00 | 0.97 | 0.98 | 0.93 | 0.93 | 0.98 | 0.99 | 0.98 | 0.97 | 0.89 | 0.91 | 0.99 | 0.99 |
| Two Patterns | 0.90 | 0.70 | 0.98 | 1.00 | 1.00 | 0.98 | 1.00 | 0.83 | 0.80 | 0.96 | 0.93 | 0.93 | 1.00 | 0.98 | 1.00 | 0.98 | 0.90 | 0.87 | 1.00 | 0.98 | 1.00 | 0.70 | 0.91 | 0.85 |
| UMD | 0.78 | 0.59 | 0.78 | 0.92 | 0.96 | 0.97 | 0.73 | 0.88 | 0.64 | 0.93 | 0.83 | 0.81 | 0.91 | 0.92 | 0.63 | 0.66 | 0.93 | 0.79 | 0.90 | 0.93 | 0.79 | 0.80 | 0.91 | 0.94 |
| Wave Gesture Library All | 0.95 | 0.82 | 0.97 | 0.97 | 0.97 | 0.97 | 0.94 | 0.95 | 0.95 | 0.97 | 0.97 | 0.97 | 0.97 | 0.97 | 0.64 | 0.78 | 0.97 | 0.96 | 0.97 | 0.97 | 0.76 | 0.66 | 0.97 | 0.97 |
| Wave Gesture Library | 0.79 | 0.72 | 0.81 | 0.82 | 0.80 | 0.80 | 0.77 | 0.74 | 0.75 | 0.80 | 0.79 | 0.78 | 0.81 | 0.80 | 0.74 | 0.71 | 0.80 | 0.80 | 0.80 | 0.81 | 0.74 | 0.73 | 0.80 | 0.80 |
| Wave Gesture Library | 0.70 | 0.57 | 0.76 | 0.76 | 0.74 | 0.73 | 0.69 | 0.68 | 0.68 | 0.74 | 0.75 | 0.74 | 0.77 | 0.75 | 0.67 | 0.70 | 0.75 | 0.74 | 0.74 | 0.75 | 0.70 | 0.67 | 0.75 | 0.74 |
| Wave Gesture Library | 0.70 | 0.65 | 0.75 | 0.75 | 0.76 | 0.75 | 0.69 | 0.70 | 0.71 | 0.76 | 0.75 | 0.75 | 0.75 | 0.74 | 0.69 | 0.70 | 0.75 | 0.74 | 0.74 | 0.74 | 0.70 | 0.70 | 0.75 | 0.75 |
| Wafer | 0.99 | 0.99 | 1.00 | 1.00 | 1.00 | 1.00 | 0.99 | 0.99 | 1.00 | 1.00 | 1.00 | 1.00 | 1.00 | 0.99 | 0.99 | 0.99 | 1.00 | 0.99 | 1.00 | 1.00 | 0.99 | 1.00 | 0.99 | 1.00 |
| Wine | 0.88 | 0.82 | 0.93 | 0.86 | 0.86 | 0.88 | 0.75 | 0.70 | 0.86 | 0.89 | 0.93 | 0.91 | 0.88 | 0.89 | 0.62 | 0.70 | 0.93 | 0.93 | 0.84 | 0.86 | 0.77 | 0.54 | 0.93 | 0.93 |
| Word Synonyms | 0.54 | 0.41 | 0.65 | 0.71 | 0.71 | 0.72 | 0.49 | 0.48 | 0.40 | 0.68 | 0.64 | 0.63 | 0.70 | 0.69 | 0.34 | 0.47 | 0.63 | 0.65 | 0.70 | 0.70 | 0.42 | 0.49 | 0.64 | 0.64 |
| Worms | 0.70 | 0.53 | 0.54 | 0.63 | 0.56 | 0.57 | 0.50 | 0.43 | 0.39 | 0.50 | 0.50 | 0.54 | 0.53 | 0.53 | 0.46 | 0.47 | 0.49 | 0.51 | 0.55 | 0.53 | 0.47 | 0.39 | 0.53 | 0.53 |
| Worms Two Class | 0.74 | 0.57 | 0.50 | 0.58 | 0.57 | 0.53 | 0.53 | 0.53 | 0.55 | 0.57 | 0.47 | 0.57 | 0.59 | 0.60 | 0.51 | 0.55 | 0.52 | 0.49 | 0.62 | 0.61 | 0.58 | 0.60 | 0.57 | 0.48 |
| Yoga | 0.87 | 0.86 | 0.95 | 0.95 | 0.95 | 0.95 | 0.88 | 0.88 | 0.90 | 0.94 | 0.95 | 0.95 | 0.95 | 0.96 | 0.88 | 0.89 | 0.96 | 0.95 | 0.96 | 0.95 | 0.87 | 0.81 | 0.96 | 0.96 |

Table 6: Detailed results of the multivariate time series classification task across all conducted experiments. Each row corresponds to a specific dataset, each column represents an experiment ID, and each cell contains the classification accuracy for the respective dataset and experiment.

| ExpID Dataset | b1 | b2 | b3 | r1-1 | r1-2 | r1-3 | r2-1 | r2-2 | r2-3 | r3-1 | r3-2 | r3-3 | s1 | s11 | s2 | s21 | s3 | s31 | rs1 | rs11 | rs2 | rs21 | rs3 | rs31 |
|---|---|---|---|---|---|---|---|---|---|---|---|---|---|---|---|---|---|---|---|---|---|---|---|---|
| Articulary Word Recognition | 0.95 | 0.92 | 0.97 | 0.97 | 0.95 | 0.97 | 0.89 | 0.92 | 0.83 | 0.96 | 0.94 | 0.92 | 0.98 | 0.97 | 0.94 | 0.92 | 0.97 | 0.98 | 0.93 | 0.97 | 0.96 | 0.94 | 0.97 | 0.97 |
| Atrial Fibrillation | 0.20 | 0.33 | 0.33 | 0.33 | 0.33 | 0.33 | 0.33 | 0.33 | 0.20 | 0.27 | 0.33 | 0.33 | 0.33 | 0.33 | 0.33 | 0.20 | 0.33 | 0.33 | 0.33 | 0.33 | 0.27 | 0.27 | 0.33 | 0.33 |

Table 6 – *Continued from previous page*

| | b1 | b2 | b3 | r1-1 | r1-2 | r1-3 | r2-1 | r2-2 | r2-3 | r3-1 | r3-2 | r3-3 | s1 | s11 | s2 | s21 | s3 | s31 | rs1 | rs11 | rs2 | rs21 | rs3 | rs31 |
|---|---|---|---|---|---|---|---|---|---|---|---|---|---|---|---|---|---|---|---|---|---|---|---|---|
| Basic Motions | 1.00 | 0.95 | 0.95 | 0.95 | 0.90 | 0.95 | 1.00 | 0.88 | 0.83 | 0.93 | 0.90 | 0.90 | 0.95 | 0.97 | 0.90 | 0.95 | 0.90 | 0.93 | 1.00 | 1.00 | 1.00 | 0.97 | 0.93 | 0.90 |
| Character Trajectories | 0.98 | 0.07 | 0.93 | 0.97 | 0.97 | 0.96 | 0.07 | 0.07 | 0.07 | 0.88 | 0.82 | 0.79 | 0.95 | 0.95 | 0.07 | 0.07 | 0.93 | 0.94 | 0.90 | 0.98 | 0.07 | 0.07 | 0.93 | 0.91 |
| Cricket | 1.00 | 0.94 | 0.94 | 0.80 | 0.90 | 0.92 | 0.86 | 0.97 | 0.61 | 0.96 | 0.92 | 0.90 | 0.90 | 0.88 | 0.97 | 0.96 | 0.94 | 0.94 | 0.84 | 0.90 | 0.99 | 0.99 | 0.94 | 0.93 |
| Duck Duck Geese | 0.50 | 0.34 | 0.32 | 0.28 | 0.26 | 0.28 | 0.30 | 0.32 | 0.34 | 0.40 | 0.36 | 0.34 | 0.36 | 0.42 | 0.36 | 0.32 | 0.28 | 0.30 | 0.46 | 0.20 | 0.30 | 0.28 | 0.32 | 0.32 |
| Ring | 0.94 | 0.89 | 0.95 | 0.95 | 0.93 | 0.96 | 0.85 | 0.89 | 0.94 | 0.95 | 0.97 | 0.97 | 0.96 | 0.96 | 0.86 | 0.80 | 0.97 | 0.97 | 0.96 | 0.97 | 0.92 | 0.83 | 0.95 | 0.96 |
| Eigen Worms | 0.76 | 0.45 | 0.56 | 0.65 | 0.70 | 0.67 | 0.55 | 0.58 | 0.45 | 0.56 | 0.50 | 0.52 | 0.66 | 0.71 | 0.62 | 0.64 | 0.55 | 0.58 | 0.64 | 0.72 | 0.54 | 0.53 | 0.54 | 0.58 |
| Epilepsy | 0.96 | 0.83 | 0.80 | 0.89 | 0.82 | 0.93 | 0.79 | 0.78 | 0.86 | 0.86 | 0.80 | 0.82 | 0.88 | 0.92 | 0.91 | 0.93 | 0.86 | 0.85 | 0.93 | 0.94 | 0.93 | 0.94 | 0.88 | 0.85 |
| Ethanol Concentration | 0.32 | 0.39 | 0.34 | 0.32 | 0.32 | 0.37 | 0.27 | 0.24 | 0.23 | 0.36 | 0.32 | 0.32 | 0.33 | 0.34 | 0.26 | 0.25 | 0.33 | 0.32 | 0.31 | 0.31 | 0.27 | 0.24 | 0.31 | 0.32 |
| Face Detection | 0.50 | 0.50 | 0.56 | 0.58 | 0.63 | 0.63 | 0.52 | 0.52 | 0.50 | 0.61 | 0.61 | 0.62 | 0.63 | 0.64 | 0.55 | 0.56 | 0.61 | 0.61 | 0.58 | 0.63 | 0.52 | 0.52 | 0.60 | 0.58 |
| Finger Movements | 0.58 | 0.49 | 0.44 | 0.53 | 0.50 | 0.51 | 0.50 | 0.50 | 0.50 | 0.50 | 0.51 | 0.50 | 0.49 | 0.45 | 0.50 | 0.52 | 0.54 | 0.55 | 0.50 | 0.50 | 0.53 | 0.53 | 0.55 | 0.51 |
| Hand Movement Direction | 0.30 | 0.30 | 0.34 | 0.30 | 0.30 | 0.30 | 0.30 | 0.25 | 0.30 | 0.43 | 0.40 | 0.38 | 0.30 | 0.43 | 0.25 | 0.25 | 0.43 | 0.37 | 0.31 | 0.30 | 0.30 | 0.24 | 0.30 | 0.35 |
| Handwriting | 0.64 | 0.12 | 0.45 | 0.46 | 0.47 | 0.44 | 0.18 | 0.23 | 0.12 | 0.55 | 0.53 | 0.40 | 0.46 | 0.49 | 0.25 | 0.17 | 0.54 | 0.35 | 0.53 | 0.42 | 0.28 | 0.08 | 0.49 | 0.35 |
| Heartbeat | 0.74 | 0.70 | 0.68 | 0.68 | 0.72 | 0.72 | 0.70 | 0.74 | 0.71 | 0.72 | 0.72 | 0.72 | 0.70 | 0.72 | 0.76 | 0.72 | 0.72 | 0.72 | 0.71 | 0.72 | 0.70 | 0.71 | 0.72 | 0.73 |
| Insect Wingbeat | 0.20 | 0.10 | 0.17 | 0.17 | 0.41 | 0.42 | 0.10 | 0.10 | 0.10 | 0.23 | 0.21 | 0.19 | 0.22 | 0.32 | 0.10 | 0.10 | 0.24 | 0.37 | 0.25 | 0.36 | 0.10 | 0.10 | 0.22 | 0.35 |
| Japanese Vowels | 0.95 | 0.18 | 0.91 | 0.92 | 0.90 | 0.89 | 0.18 | 0.18 | 0.18 | 0.88 | 0.86 | 0.87 | 0.89 | 0.91 | 0.18 | 0.18 | 0.91 | 0.93 | 0.91 | 0.92 | 0.18 | 0.18 | 0.92 | 0.87 |
| LSST | 0.49 | 0.38 | 0.42 | 0.42 | 0.40 | 0.40 | 0.41 | 0.39 | 0.41 | 0.43 | 0.40 | 0.43 | 0.44 | 0.42 | 0.40 | 0.35 | 0.41 | 0.43 | 0.44 | 0.42 | 0.35 | 0.40 | 0.42 | 0.41 |
| Libras | 0.88 | 0.75 | 0.84 | 0.76 | 0.73 | 0.74 | 0.61 | 0.74 | 0.58 | 0.76 | 0.77 | 0.78 | 0.77 | 0.75 | 0.64 | 0.59 | 0.75 | 0.73 | 0.75 | 0.78 | 0.79 | 0.59 | 0.78 | 0.76 |
| Motor Imagery | 0.52 | 0.46 | 0.50 | 0.51 | 0.52 | 0.56 | 0.56 | 0.46 | 0.56 | 0.50 | 0.50 | 0.51 | 0.52 | 0.54 | 0.55 | 0.52 | 0.51 | 0.50 | 0.51 | 0.54 | 0.56 | 0.49 | 0.50 | 0.51 |
| NATOPS | 0.85 | 0.80 | 0.78 | 0.80 | 0.81 | 0.79 | 0.82 | 0.81 | 0.83 | 0.89 | 0.78 | 0.79 | 0.79 | 0.76 | 0.87 | 0.82 | 0.83 | 0.81 | 0.77 | 0.68 | 0.79 | 0.82 | 0.79 | 0.82 |
| PEMS SF | 0.76 | 0.74 | 0.83 | 0.70 | 0.70 | 0.72 | 0.62 | 0.66 | 0.65 | 0.76 | 0.77 | 0.76 | 0.76 | 0.74 | 0.74 | 0.67 | 0.85 | 0.77 | 0.73 | 0.79 | 0.75 | 0.69 | 0.80 | 0.77 |
| Pen Digits | 0.98 | 0.97 | 0.99 | 0.99 | 0.99 | 0.99 | 0.99 | 0.99 | 0.98 | 0.99 | 0.99 | 0.99 | 0.99 | 0.99 | 0.98 | 0.98 | 0.99 | 0.99 | 0.99 | 0.99 | 0.99 | 0.98 | 0.99 | 0.99 |
| Phoneme Spectra | 0.24 | 0.14 | 0.12 | 0.13 | 0.13 | 0.11 | 0.10 | 0.08 | 0.08 | 0.10 | 0.10 | 0.11 | 0.12 | 0.10 | 0.09 | 0.09 | 0.12 | 0.10 | 0.13 | 0.13 | 0.12 | 0.09 | 0.12 | 0.11 |
| Racket Sports | 0.86 | 0.81 | 0.82 | 0.81 | 0.79 | 0.80 | 0.84 | 0.84 | 0.71 | 0.75 | 0.74 | 0.70 | 0.78 | 0.78 | 0.80 | 0.86 | 0.76 | 0.80 | 0.83 | 0.78 | 0.81 | 0.84 | 0.84 | 0.81 |
| Self Regulation SCP | 0.77 | 0.84 | 0.84 | 0.83 | 0.82 | 0.86 | 0.75 | 0.74 | 0.74 | 0.82 | 0.80 | 0.81 | 0.84 | 0.74 | 0.75 | 0.73 | 0.83 | 0.84 | 0.81 | 0.74 | 0.72 | 0.73 | 0.80 | 0.86 |
| Self Regulation SCP | 0.54 | 0.51 | 0.53 | 0.49 | 0.53 | 0.53 | 0.53 | 0.53 | 0.50 | 0.52 | 0.51 | 0.52 | 0.53 | 0.51 | 0.51 | 0.54 | 0.54 | 0.55 | 0.52 | 0.50 | 0.53 | 0.52 | 0.53 | 0.55 |
| Spoken Arabic Digits | 0.98 | 0.10 | 0.97 | 0.99 | 0.99 | 0.99 | 0.10 | 0.10 | 0.10 | 0.97 | 0.96 | 0.96 | 0.98 | 0.98 | 0.10 | 0.10 | 0.98 | 0.97 | 0.94 | 0.98 | 0.10 | 0.10 | 0.98 | 0.98 |
| Stand Walk Jump | 0.29 | 0.29 | 0.29 | 0.29 | 0.29 | 0.29 | 0.29 | 0.14 | 0.14 | 0.36 | 0.36 | 0.29 | 0.29 | 0.29 | 0.36 | 0.43 | 0.36 | 0.36 | 0.36 | 0.43 | 0.29 | 0.36 | 0.36 | 0.36 |
| Wave Gesture Library | 0.94 | 0.78 | 0.90 | 0.93 | 0.92 | 0.90 | 0.83 | 0.61 | 0.69 | 0.90 | 0.85 | 0.93 | 0.94 | 0.95 | 0.75 | 0.78 | 0.90 | 0.88 | 0.94 | 0.94 | 0.83 | 0.80 | 0.92 | 0.91 |

Table 7: Detailed results of the univariate forecasting task across all conducted experiments. Each row corresponds to a specific dataset and forecast horizon (h), each column represents an experiment ID, and each cell contains the forecasting accuracy for the respective dataset, forecast horizon, and experiment.

| Exp ID Dataset | h | b1 | b2 | b3 | r1-1 | r1-2 | r1-3 | r2-1 | r2-2 | r2-3 | r3-1 | r3-2 | r3-3 | s1 | s11 | s2 | s21 | s3 | s31 | rs1 | rs11 | rs2 | rs21 | rs3 | rs31 |
|---|---|---|---|---|---|---|---|---|---|---|---|---|---|---|---|---|---|---|---|---|---|---|---|---|---|
| ETTh1 | 24 | 0.05 | 0.05 | 0.06 | 0.06 | 0.05 | 0.02 | 0.05 | 0.05 | 0.05 | 0.06 | 0.06 | 0.06 | 0.05 | 0.04 | 0.05 | 0.05 | 0.06 | 0.06 | 0.04 | 0.06 | 0.05 | 0.05 | 0.06 | 0.06 |
| | 48 | 0.08 | 0.05 | 0.05 | 0.05 | 0.05 | 0.07 | 0.05 | 0.05 | 0.05 | 0.05 | 0.05 | 0.05 | 0.05 | 0.11 | 0.05 | 0.05 | 0.05 | 0.05 | 0.03 | 0.03 | 0.05 | 0.05 | 0.05 | 0.05 |
| | 96 | 0.13 | 0.05 | 0.05 | 0.11 | 0.10 | 0.39 | 0.05 | 0.05 | 0.05 | 0.05 | 0.05 | 0.05 | 0.04 | 0.04 | 0.05 | 0.05 | 0.05 | 0.05 | 0.04 | 0.03 | 0.05 | 0.04 | 0.05 | 0.05 |
| | 168 | 0.13 | 0.05 | 0.05 | 0.10 | 0.09 | 0.49 | 0.05 | 0.05 | 0.05 | 0.05 | 0.05 | 0.05 | 0.04 | 0.26 | 0.05 | 0.05 | 0.05 | 0.05 | 0.04 | 0.06 | 0.05 | 0.05 | 0.05 | 0.05 |
| | 288 | 0.13 | 0.05 | 0.05 | 0.08 | 0.07 | 0.46 | 0.05 | 0.05 | 0.05 | 0.05 | 0.05 | 0.05 | 0.04 | 0.28 | 0.05 | 0.05 | 0.05 | 0.05 | 0.04 | 0.06 | 0.05 | 0.05 | 0.05 | 0.05 |
| | 336 | 0.12 | 0.05 | 0.05 | 0.07 | 0.07 | 0.46 | 0.05 | 0.05 | 0.05 | 0.05 | 0.05 | 0.05 | 0.02 | 0.27 | 0.05 | 0.05 | 0.05 | 0.05 | 0.04 | 0.04 | 0.05 | 0.05 | 0.05 | 0.05 |
| | 672 | 0.08 | 0.03 | 0.05 | 0.05 | 0.05 | 0.45 | 0.03 | 0.03 | 0.03 | 0.05 | 0.05 | 0.05 | 0.03 | 0.27 | 0.03 | 0.03 | 0.05 | 0.05 | 0.04 | 0.10 | 0.03 | 0.03 | 0.05 | 0.05 |
| | 720 | 0.08 | 0.03 | 0.05 | 0.05 | 0.05 | 0.42 | 0.03 | 0.03 | 0.03 | 0.05 | 0.05 | 0.05 | 0.03 | 0.27 | 0.03 | 0.03 | 0.05 | 0.05 | 0.04 | 0.12 | 0.03 | 0.03 | 0.05 | 0.05 |

Table 7 – *Continued from previous page*

| ExpID Dataset | h | b1 | b2 | b3 | r1-1 | r1-2 | r1-3 | r2-1 | r2-2 | r2-3 | r3-1 | r3-2 | r3-3 | s1 | s11 | s2 | s21 | s3 | s31 | rs1 | rs11 | rs2 | rs21 | rs3 | rs31 |
|---|---|---|---|---|---|---|---|---|---|---|---|---|---|---|---|---|---|---|---|---|---|---|---|---|---|
| ETTh2 | 24 | 0.05 | 0.05 | 0.06 | 0.05 | 0.05 | 0.06 | 0.05 | 0.05 | 0.05 | 0.06 | 0.06 | 0.06 | 0.05 | 0.06 | 0.05 | 0.05 | 0.06 | 0.06 | 0.05 | 0.07 | 0.05 | 0.05 | 0.06 | 0.06 |
| | 48 | 0.05 | 0.05 | 0.06 | 0.06 | 0.06 | 0.07 | 0.05 | 0.05 | 0.05 | 0.06 | 0.06 | 0.06 | 0.05 | 0.07 | 0.05 | 0.05 | 0.06 | 0.06 | 0.05 | 0.07 | 0.05 | 0.05 | 0.06 | 0.06 |
| | 96 | 0.06 | 0.05 | 0.06 | 0.06 | 0.06 | 0.07 | 0.05 | 0.05 | 0.05 | 0.06 | 0.06 | 0.06 | 0.05 | 0.08 | 0.05 | 0.05 | 0.06 | 0.06 | 0.05 | 0.08 | 0.05 | 0.05 | 0.06 | 0.06 |
| | 168 | 0.06 | 0.06 | 0.06 | 0.07 | 0.06 | 0.07 | 0.06 | 0.06 | 0.06 | 0.06 | 0.06 | 0.06 | 0.05 | 0.08 | 0.06 | 0.06 | 0.06 | 0.06 | 0.05 | 0.08 | 0.06 | 0.06 | 0.06 | 0.06 |
| | 288 | 0.06 | 0.06 | 0.06 | 0.07 | 0.06 | 0.08 | 0.06 | 0.06 | 0.06 | 0.06 | 0.06 | 0.06 | 0.05 | 0.09 | 0.06 | 0.06 | 0.06 | 0.06 | 0.06 | 0.08 | 0.06 | 0.06 | 0.06 | 0.06 |
| | 336 | 0.06 | 0.06 | 0.06 | 0.07 | 0.06 | 0.07 | 0.06 | 0.06 | 0.06 | 0.06 | 0.06 | 0.06 | 0.06 | 0.09 | 0.06 | 0.06 | 0.06 | 0.06 | 0.05 | 0.08 | 0.06 | 0.06 | 0.06 | 0.06 |
| | 672 | 0.06 | 0.05 | 0.05 | 0.07 | 0.06 | 0.07 | 0.05 | 0.05 | 0.05 | 0.05 | 0.05 | 0.05 | 0.05 | 0.09 | 0.05 | 0.05 | 0.05 | 0.05 | 0.05 | 0.08 | 0.05 | 0.05 | 0.05 | 0.05 |
| | 720 | 0.06 | 0.05 | 0.05 | 0.06 | 0.06 | 0.07 | 0.05 | 0.05 | 0.05 | 0.05 | 0.05 | 0.05 | 0.05 | 0.09 | 0.05 | 0.05 | 0.05 | 0.05 | 0.05 | 0.08 | 0.05 | 0.05 | 0.05 | 0.05 |
| ETTm1 | 24 | 0.04 | 0.05 | 0.06 | 0.05 | 0.04 | 0.04 | 0.05 | 0.05 | 0.05 | 0.06 | 0.06 | 0.06 | 0.04 | 0.04 | 0.05 | 0.05 | 0.06 | 0.06 | 0.04 | 0.04 | 0.05 | 0.05 | 0.06 | 0.06 |
| | 48 | 0.06 | 0.05 | 0.06 | 0.21 | 0.04 | 0.02 | 0.05 | 0.05 | 0.05 | 0.06 | 0.06 | 0.06 | 0.02 | 0.02 | 0.05 | 0.05 | 0.06 | 0.06 | 0.02 | 0.02 | 0.05 | 0.05 | 0.06 | 0.06 |
| | 96 | 0.08 | 0.03 | 0.05 | 0.21 | 0.21 | 0.03 | 0.03 | 0.03 | 0.03 | 0.05 | 0.05 | 0.05 | 0.03 | 0.03 | 0.03 | 0.03 | 0.05 | 0.05 | 0.03 | 0.03 | 0.03 | 0.03 | 0.05 | 0.05 |
| | 168 | 0.13 | 0.03 | 0.05 | 0.16 | 0.25 | 0.05 | 0.03 | 0.03 | 0.03 | 0.05 | 0.05 | 0.05 | 0.04 | 0.05 | 0.03 | 0.03 | 0.05 | 0.05 | 0.06 | 0.06 | 0.03 | 0.03 | 0.05 | 0.05 |
| | 288 | 0.21 | 0.04 | 0.05 | 0.22 | 0.39 | 0.32 | 0.04 | 0.04 | 0.04 | 0.05 | 0.05 | 0.05 | 0.18 | 0.30 | 0.04 | 0.04 | 0.05 | 0.05 | 0.07 | 0.07 | 0.04 | 0.04 | 0.05 | 0.05 |
| | 336 | 0.24 | 0.04 | 0.05 | 0.27 | 0.51 | 0.45 | 0.04 | 0.04 | 0.04 | 0.05 | 0.05 | 0.05 | 0.28 | 0.46 | 0.04 | 0.04 | 0.05 | 0.05 | 0.07 | 0.07 | 0.04 | 0.04 | 0.05 | 0.05 |
| | 672 | 0.30 | 0.05 | 0.04 | 0.27 | 0.33 | 0.49 | 0.05 | 0.05 | 0.05 | 0.04 | 0.04 | 0.04 | 0.32 | 0.55 | 0.05 | 0.05 | 0.04 | 0.04 | 0.19 | 0.19 | 0.05 | 0.05 | 0.04 | 0.04 |
| | 720 | 0.31 | 0.05 | 0.04 | 0.27 | 0.33 | 0.48 | 0.05 | 0.05 | 0.05 | 0.04 | 0.04 | 0.04 | 0.32 | 0.54 | 0.05 | 0.05 | 0.04 | 0.04 | 0.19 | 0.19 | 0.05 | 0.05 | 0.04 | 0.04 |
| ETTm2 | 24 | 0.04 | 0.05 | 0.06 | 0.06 | 0.06 | 0.06 | 0.05 | 0.05 | 0.05 | 0.06 | 0.06 | 0.06 | 0.05 | 0.04 | 0.05 | 0.05 | 0.06 | 0.06 | 0.05 | 0.05 | 0.05 | 0.05 | 0.06 | 0.06 |
| | 48 | 0.05 | 0.05 | 0.06 | 0.06 | 0.06 | 0.06 | 0.05 | 0.05 | 0.05 | 0.06 | 0.06 | 0.06 | 0.05 | 0.05 | 0.05 | 0.05 | 0.06 | 0.06 | 0.05 | 0.05 | 0.05 | 0.05 | 0.06 | 0.06 |
| | 96 | 0.05 | 0.05 | 0.06 | 0.05 | 0.06 | 0.06 | 0.05 | 0.05 | 0.05 | 0.06 | 0.06 | 0.06 | 0.05 | 0.05 | 0.05 | 0.05 | 0.06 | 0.06 | 0.05 | 0.05 | 0.05 | 0.05 | 0.06 | 0.06 |
| | 168 | 0.05 | 0.05 | 0.06 | 0.06 | 0.06 | 0.07 | 0.05 | 0.05 | 0.05 | 0.06 | 0.06 | 0.06 | 0.06 | 0.05 | 0.05 | 0.05 | 0.06 | 0.06 | 0.06 | 0.06 | 0.05 | 0.05 | 0.06 | 0.06 |
| | 288 | 0.05 | 0.05 | 0.06 | 0.06 | 0.06 | 0.07 | 0.05 | 0.05 | 0.05 | 0.06 | 0.06 | 0.06 | 0.05 | 0.05 | 0.05 | 0.05 | 0.06 | 0.06 | 0.06 | 0.06 | 0.05 | 0.05 | 0.06 | 0.06 |
| | 336 | 0.05 | 0.05 | 0.06 | 0.06 | 0.06 | 0.08 | 0.05 | 0.05 | 0.05 | 0.06 | 0.06 | 0.06 | 0.06 | 0.05 | 0.05 | 0.05 | 0.06 | 0.06 | 0.06 | 0.06 | 0.05 | 0.05 | 0.06 | 0.06 |
| | 672 | 0.06 | 0.05 | 0.06 | 0.07 | 0.07 | 0.08 | 0.05 | 0.05 | 0.05 | 0.06 | 0.06 | 0.06 | 0.07 | 0.06 | 0.05 | 0.05 | 0.06 | 0.06 | 0.07 | 0.07 | 0.05 | 0.05 | 0.06 | 0.06 |
| | 720 | 0.06 | 0.05 | 0.06 | 0.07 | 0.07 | 0.08 | 0.05 | 0.05 | 0.05 | 0.06 | 0.06 | 0.06 | 0.07 | 0.06 | 0.05 | 0.05 | 0.06 | 0.06 | 0.07 | 0.07 | 0.05 | 0.05 | 0.06 | 0.06 |
| Electricity Load | 24 | 0.03 | 0.08 | 0.03 | 0.03 | 0.03 | 0.04 | 0.08 | 0.08 | 0.08 | 0.03 | 0.03 | 0.03 | 0.03 | 0.03 | 0.08 | 0.08 | 0.03 | 0.03 | 0.03 | 0.03 | 0.08 | 0.10 | 0.03 | 0.03 |
| | 48 | 0.03 | 0.03 | 0.03 | 0.04 | 0.04 | 0.05 | 0.03 | 0.03 | 0.03 | 0.03 | 0.03 | 0.03 | 0.03 | 0.03 | 0.03 | 0.03 | 0.03 | 0.03 | 0.03 | 0.25 | 0.03 | 0.03 | 0.03 | 0.03 |
| | 96 | 0.03 | 0.03 | 0.03 | 0.04 | 0.14 | 0.20 | 0.03 | 0.03 | 0.03 | 0.03 | 0.03 | 0.03 | 0.03 | 0.03 | 0.03 | 0.03 | 0.03 | 0.03 | 0.03 | 0.16 | 0.03 | 0.03 | 0.03 | 0.03 |
| | 168 | 0.29 | 0.03 | 0.03 | 0.42 | 0.16 | 0.20 | 0.03 | 0.03 | 0.03 | 0.03 | 0.03 | 0.03 | 0.03 | 0.03 | 0.03 | 0.03 | 0.03 | 0.03 | 0.03 | 0.23 | 0.03 | 0.03 | 0.03 | 0.03 |
| | 288 | 0.22 | 0.09 | 0.03 | 0.76 | 0.12 | 0.19 | 0.09 | 0.09 | 0.09 | 0.03 | 0.03 | 0.03 | 0.32 | 0.29 | 0.09 | 0.09 | 0.03 | 0.03 | 0.28 | 0.46 | 0.09 | 0.12 | 0.03 | 0.03 |
| | 336 | 0.20 | 0.10 | 0.03 | 1.07 | 0.13 | 0.18 | 0.10 | 0.10 | 0.10 | 0.03 | 0.03 | 0.03 | 0.31 | 0.27 | 0.10 | 0.10 | 0.03 | 0.03 | 0.26 | 0.35 | 0.10 | 0.13 | 0.03 | 0.03 |
| | 672 | 0.24 | 0.15 | 0.04 | 1.56 | 0.13 | 0.42 | 0.15 | 0.15 | 0.15 | 0.04 | 0.04 | 0.04 | 0.34 | 0.27 | 0.15 | 0.15 | 0.04 | 0.04 | 0.32 | 0.12 | 0.15 | 0.14 | 0.04 | 0.04 |
| | 720 | 0.25 | 0.15 | 0.04 | 1.54 | 0.15 | 0.49 | 0.15 | 0.15 | 0.15 | 0.04 | 0.04 | 0.04 | 0.34 | 0.28 | 0.15 | 0.15 | 0.04 | 0.04 | 0.32 | 0.19 | 0.15 | 0.22 | 0.04 | 0.04 |
| Weather | 24 | 0.05 | 0.02 | 0.03 | 0.05 | 0.04 | 0.17 | 0.02 | 0.02 | 0.02 | 0.03 | 0.03 | 0.03 | 0.02 | 0.02 | 0.02 | 0.02 | 0.03 | 0.03 | 0.02 | 0.02 | 0.02 | 0.02 | 0.03 | 0.03 |
| | 48 | 0.18 | 0.02 | 0.03 | 0.09 | 0.08 | 0.31 | 0.02 | 0.02 | 0.02 | 0.03 | 0.03 | 0.03 | 0.02 | 0.02 | 0.02 | 0.02 | 0.03 | 0.03 | 0.02 | 0.02 | 0.02 | 0.02 | 0.03 | 0.03 |
| | 96 | 0.02 | 0.02 | 0.03 | 0.02 | 0.02 | 0.02 | 0.02 | 0.02 | 0.02 | 0.03 | 0.03 | 0.03 | 0.02 | 0.02 | 0.02 | 0.02 | 0.03 | 0.03 | 0.02 | 0.02 | 0.02 | 0.02 | 0.03 | 0.03 |
| | 168 | 0.02 | 0.02 | 0.03 | 0.02 | 0.02 | 0.02 | 0.02 | 0.02 | 0.02 | 0.03 | 0.03 | 0.03 | 0.02 | 0.02 | 0.02 | 0.02 | 0.03 | 0.03 | 0.02 | 0.02 | 0.02 | 0.02 | 0.03 | 0.03 |
| | 288 | 0.02 | 0.02 | 0.03 | 0.02 | 0.02 | 0.02 | 0.02 | 0.02 | 0.02 | 0.03 | 0.03 | 0.03 | 0.02 | 0.02 | 0.02 | 0.02 | 0.03 | 0.03 | 0.02 | 0.02 | 0.02 | 0.02 | 0.03 | 0.03 |
| | 336 | 0.02 | 0.02 | 0.03 | 0.02 | 0.02 | 0.02 | 0.02 | 0.02 | 0.02 | 0.03 | 0.03 | 0.03 | 0.02 | 0.02 | 0.02 | 0.02 | 0.03 | 0.03 | 0.02 | 0.02 | 0.02 | 0.02 | 0.03 | 0.03 |
| | 672 | 0.02 | 0.02 | 0.03 | 0.02 | 0.02 | 0.02 | 0.02 | 0.02 | 0.02 | 0.03 | 0.03 | 0.03 | 0.02 | 0.02 | 0.02 | 0.02 | 0.03 | 0.03 | 0.02 | 0.02 | 0.02 | 0.02 | 0.03 | 0.03 |
| | 720 | 0.02 | 0.02 | 0.03 | 0.02 | 0.02 | 0.02 | 0.02 | 0.02 | 0.02 | 0.03 | 0.03 | 0.03 | 0.02 | 0.02 | 0.02 | 0.02 | 0.03 | 0.03 | 0.02 | 0.02 | 0.02 | 0.02 | 0.03 | 0.03 |

Table 8: Detailed results of the multivariate forecasting task across all conducted experiments. Each row corresponds to a specific dataset and forecast horizon (h), each column represents an experiment ID, and each cell contains the forecasting accuracy for the respective dataset, forecast horizon, and experiment.

| ExpID Dataset | h | b1 | b2 | b3 | r1-1 | r1-2 | r1-3 | r2-1 | r2-2 | r2-3 | r3-1 | r3-2 | r3-3 | s1 | s11 | s2 | s21 | s3 | s31 | rs1 | rs11 | rs2 | rs21 | rs3 | rs31 |
|---|---|---|---|---|---|---|---|---|---|---|---|---|---|---|---|---|---|---|---|---|---|---|---|---|---|
| ETTh1 | 24 | 0.02 | 0.02 | 0.03 | 0.02 | 0.02 | 0.03 | 0.02 | 0.02 | 0.02 | 0.03 | 0.03 | 0.03 | 0.02 | 0.02 | 0.02 | 0.02 | 0.03 | 0.03 | 0.02 | 0.02 | 0.02 | 0.02 | 0.03 | 0.03 |
| | 48 | 0.02 | 0.02 | 0.03 | 0.02 | 0.02 | 0.19 | 0.02 | 0.02 | 0.02 | 0.03 | 0.03 | 0.03 | 0.02 | 0.02 | 0.02 | 0.02 | 0.03 | 0.03 | 0.02 | 0.02 | 0.02 | 0.02 | 0.03 | 0.03 |
| | 96 | 0.16 | 0.04 | 0.03 | 0.11 | 0.13 | 0.20 | 0.04 | 0.04 | 0.04 | 0.03 | 0.03 | 0.03 | 0.16 | 0.17 | 0.04 | 0.04 | 0.03 | 0.03 | 0.06 | 0.25 | 0.04 | 0.04 | 0.03 | 0.03 |
| | 168 | 0.16 | 0.05 | 0.03 | 0.16 | 0.15 | 0.20 | 0.05 | 0.05 | 0.05 | 0.03 | 0.03 | 0.03 | 0.19 | 0.18 | 0.05 | 0.05 | 0.03 | 0.03 | 0.31 | 0.27 | 0.05 | 0.05 | 0.03 | 0.03 |
| | 288 | 0.17 | 0.05 | 0.03 | 0.19 | 0.18 | 0.23 | 0.05 | 0.05 | 0.05 | 0.03 | 0.03 | 0.03 | 0.20 | 0.19 | 0.05 | 0.05 | 0.03 | 0.03 | 0.37 | 0.29 | 0.05 | 0.05 | 0.03 | 0.03 |
| | 336 | 0.18 | 0.05 | 0.03 | 0.19 | 0.18 | 0.23 | 0.05 | 0.05 | 0.05 | 0.03 | 0.03 | 0.03 | 0.20 | 0.19 | 0.05 | 0.05 | 0.03 | 0.03 | 0.38 | 0.33 | 0.05 | 0.05 | 0.03 | 0.03 |
| | 672 | 0.16 | 0.06 | 0.03 | 0.16 | 0.18 | 0.28 | 0.06 | 0.06 | 0.06 | 0.03 | 0.03 | 0.03 | 0.20 | 0.18 | 0.06 | 0.06 | 0.03 | 0.03 | 0.47 | 0.32 | 0.07 | 0.06 | 0.03 | 0.03 |
| | 720 | 0.16 | 0.07 | 0.03 | 0.16 | 0.18 | 0.29 | 0.07 | 0.07 | 0.07 | 0.03 | 0.03 | 0.03 | 0.19 | 0.19 | 0.07 | 0.07 | 0.03 | 0.03 | 0.44 | 0.31 | 0.07 | 0.07 | 0.03 | 0.03 |
| ETTh2 | 24 | 0.08 | 0.04 | 0.04 | 0.03 | 0.04 | 0.05 | 0.04 | 0.04 | 0.04 | 0.04 | 0.04 | 0.04 | 0.03 | 0.03 | 0.04 | 0.04 | 0.04 | 0.04 | 0.04 | 0.04 | 0.04 | 0.04 | 0.04 | 0.04 |
| | 48 | 0.10 | 0.04 | 0.04 | 0.07 | 0.04 | 0.12 | 0.04 | 0.04 | 0.04 | 0.04 | 0.04 | 0.04 | 0.03 | 0.03 | 0.04 | 0.04 | 0.04 | 0.04 | 0.04 | 0.04 | 0.04 | 0.04 | 0.04 | 0.04 |
| | 96 | 0.14 | 0.04 | 0.04 | 0.04 | 0.04 | 0.15 | 0.04 | 0.04 | 0.04 | 0.04 | 0.04 | 0.04 | 0.09 | 0.09 | 0.04 | 0.04 | 0.04 | 0.04 | 0.04 | 0.04 | 0.04 | 0.04 | 0.04 | 0.04 |
| | 168 | 0.05 | 0.07 | 0.04 | 0.05 | 0.04 | 0.04 | 0.07 | 0.07 | 0.07 | 0.04 | 0.04 | 0.04 | 0.11 | 0.11 | 0.07 | 0.07 | 0.04 | 0.04 | 0.04 | 0.04 | 0.07 | 0.07 | 0.04 | 0.04 |
| | 288 | 0.04 | 0.05 | 0.04 | 0.06 | 0.04 | 0.04 | 0.05 | 0.05 | 0.05 | 0.04 | 0.04 | 0.04 | 0.05 | 0.05 | 0.05 | 0.05 | 0.04 | 0.04 | 0.04 | 0.04 | 0.05 | 0.05 | 0.04 | 0.04 |
| | 336 | 0.04 | 0.05 | 0.04 | 0.05 | 0.04 | 0.04 | 0.05 | 0.05 | 0.05 | 0.04 | 0.04 | 0.04 | 0.05 | 0.05 | 0.05 | 0.05 | 0.04 | 0.04 | 0.04 | 0.04 | 0.05 | 0.05 | 0.04 | 0.04 |
| | 672 | 0.04 | 0.25 | 0.04 | 0.14 | 0.49 | 0.21 | 0.25 | 0.25 | 0.25 | 0.04 | 0.04 | 0.04 | 0.11 | 0.15 | 0.25 | 0.25 | 0.04 | 0.04 | 0.06 | 0.07 | 0.27 | 0.25 | 0.04 | 0.04 |
| | 720 | 0.04 | 0.28 | 0.04 | 0.15 | 0.50 | 0.20 | 0.28 | 0.28 | 0.28 | 0.04 | 0.04 | 0.04 | 0.12 | 0.15 | 0.28 | 0.29 | 0.04 | 0.04 | 0.08 | 0.10 | 0.27 | 0.32 | 0.04 | 0.04 |
| ETTm1 | 24 | 0.02 | 0.02 | 0.02 | 0.02 | 0.21 | 0.68 | 0.02 | 0.02 | 0.02 | 0.02 | 0.02 | 0.02 | 0.02 | 0.02 | 0.02 | 0.02 | 0.02 | 0.03 | 0.02 | 0.02 | 0.02 | 0.02 | 0.02 | 0.02 |
| | 48 | 0.02 | 0.02 | 0.02 | 0.02 | 0.33 | 0.57 | 0.02 | 0.02 | 0.02 | 0.02 | 0.20 | 0.02 | 0.02 | 0.02 | 0.02 | 0.02 | 0.02 | 0.03 | 1.29 | 0.02 | 0.02 | 0.02 | 0.02 | 0.02 |
| | 96 | 0.02 | 0.02 | 0.02 | 0.21 | 0.28 | 0.43 | 0.02 | 0.02 | 0.02 | 0.02 | 0.41 | 0.02 | 0.02 | 0.02 | 0.02 | 0.02 | 0.02 | 0.03 | 0.70 | 0.02 | 0.02 | 0.02 | 0.02 | 0.02 |
| | 168 | 0.02 | 0.02 | 0.02 | 0.23 | 0.25 | 0.41 | 0.02 | 0.02 | 0.02 | 0.02 | 0.47 | 0.02 | 0.02 | 0.02 | 0.02 | 0.02 | 0.02 | 0.03 | 0.70 | 0.02 | 0.02 | 0.02 | 0.02 | 0.02 |
| | 288 | 0.31 | 0.02 | 0.02 | 0.25 | 0.27 | 0.50 | 0.02 | 0.02 | 0.02 | 0.02 | 0.69 | 0.34 | 0.02 | 0.02 | 0.02 | 0.02 | 0.03 | 0.02 | 0.02 | 0.02 | 0.02 | 0.02 | 0.02 | 0.02 |
| | 336 | 0.34 | 0.02 | 0.02 | 0.28 | 0.30 | 0.53 | 0.02 | 0.02 | 0.02 | 0.02 | 0.77 | 0.38 | 0.02 | 0.02 | 0.02 | 0.02 | 0.03 | 0.02 | 0.02 | 0.02 | 0.02 | 0.02 | 0.02 | 0.02 |

Table 8 – *Continued from previous page*

| ExpID / Dataset | h | b1 | b2 | b3 | r1-1 | r1-2 | r1-3 | r2-1 | r2-2 | r2-3 | r3-1 | r3-2 | r3-3 | s1 | s11 | s2 | s21 | s3 | s31 | rs1 | rs11 | rs2 | rs21 | rs3 | rs31 |
|---|---|---|---|---|---|---|---|---|---|---|---|---|---|---|---|---|---|---|---|---|---|---|---|---|---|
| | 672 | 0.52 | 0.07 | 0.03 | 0.56 | 0.65 | 0.86 | 0.07 | 0.07 | 0.07 | 0.03 | 0.03 | 0.03 | 0.87 | 0.53 | 0.07 | 0.07 | 0.03 | 0.03 | 0.67 | 0.70 | 0.09 | 0.05 | 0.03 | 0.03 |
| | 720 | 0.54 | 0.07 | 0.03 | 0.62 | 0.70 | 0.89 | 0.07 | 0.07 | 0.07 | 0.03 | 0.03 | 0.03 | 0.87 | 0.55 | 0.07 | 0.07 | 0.03 | 0.03 | 0.67 | 0.74 | 0.12 | 0.11 | 0.03 | 0.03 |
| ETTm2 | 24 | 0.03 | 0.04 | 0.04 | 0.05 | 0.15 | 0.19 | 0.04 | 0.04 | 0.04 | 0.04 | 0.04 | 0.04 | 0.04 | 0.04 | 0.04 | 0.04 | 0.04 | 0.04 | 0.04 | 0.04 | 0.04 | 0.04 | 0.04 | 0.04 |
| | 48 | 0.03 | 0.04 | 0.04 | 0.87 | 0.24 | 0.23 | 0.04 | 0.04 | 0.04 | 0.04 | 0.04 | 0.04 | 0.04 | 0.04 | 0.04 | 0.04 | 0.04 | 0.04 | 0.04 | 0.04 | 0.04 | 0.04 | 0.04 | 0.04 |
| | 96 | 0.12 | 0.04 | 0.04 | 1.17 | 0.34 | 0.25 | 0.04 | 0.04 | 0.04 | 0.04 | 0.04 | 0.04 | 0.38 | 0.37 | 0.04 | 0.04 | 0.04 | 0.58 | 0.04 | 0.04 | 0.04 | 0.04 | 0.04 | 0.04 |
| | 168 | 0.16 | 0.04 | 0.04 | 1.40 | 0.47 | 0.37 | 0.04 | 0.04 | 0.04 | 0.04 | 0.04 | 0.04 | 0.59 | 0.57 | 0.04 | 0.04 | 0.04 | 0.94 | 0.04 | 0.04 | 0.04 | 0.04 | 0.04 | 0.04 |
| | 288 | 0.18 | 0.04 | 0.04 | 1.45 | 0.83 | 0.62 | 0.04 | 0.04 | 0.04 | 0.04 | 0.04 | 0.04 | 0.73 | 0.75 | 0.04 | 0.04 | 0.04 | 0.04 | 1.06 | 0.04 | 0.04 | 0.22 | 0.04 | 0.04 |
| | 336 | 0.21 | 0.04 | 0.04 | 1.49 | 1.04 | 0.78 | 0.04 | 0.04 | 0.04 | 0.04 | 0.04 | 0.04 | 0.80 | 0.83 | 0.04 | 0.04 | 0.04 | 0.04 | 1.10 | 0.04 | 0.04 | 0.22 | 0.04 | 0.04 |
| | 672 | 0.28 | 0.06 | 0.04 | 0.05 | 0.04 | 0.04 | 0.06 | 0.06 | 0.06 | 0.04 | 0.04 | 0.04 | 0.83 | 0.91 | 0.06 | 0.06 | 0.04 | 0.04 | 0.03 | 0.04 | 0.06 | 0.06 | 0.04 | 0.04 |
| | 720 | 0.28 | 0.05 | 0.04 | 0.05 | 0.04 | 0.04 | 0.05 | 0.05 | 0.05 | 0.04 | 0.04 | 0.04 | 0.83 | 0.90 | 0.05 | 0.05 | 0.04 | 0.04 | 0.03 | 0.05 | 0.05 | 0.04 | 0.04 | |
| Electricity Load | 24 | 0.04 | 10.32 | 0.06 | 0.06 | 0.07 | 0.07 | 8.54 | 242.88 | 133.74 | 7.88 | 6.21 | 7.86 | 0.07 | 0.04 | 0.98 | 1.18 | 0.03 | 0.04 | 0.26 | 0.31 | 0.96 | 2.02 | 0.03 | 0.04 |
| | 48 | 0.04 | 10.29 | 0.06 | 0.05 | 0.07 | 0.07 | 7.05 | 242.89 | 132.16 | 7.80 | 5.12 | 10.92 | 0.06 | 0.04 | 0.88 | 1.09 | 0.03 | 0.04 | 0.28 | 0.34 | 0.87 | 1.81 | 0.03 | 0.04 |
| | 96 | 0.04 | 8.84 | 0.06 | 0.05 | 0.06 | 0.06 | 6.73 | 241.38 | 134.18 | 8.05 | 7.65 | 11.58 | 0.06 | 0.04 | 0.77 | 1.83 | 0.04 | 0.04 | 0.26 | 0.36 | 0.77 | 1.62 | 0.03 | 0.04 |
| | 168 | 0.04 | 9.98 | 0.06 | 0.06 | 0.06 | 0.07 | 8.24 | 243.83 | 134.71 | 7.15 | 6.81 | 8.99 | 0.07 | 0.04 | 0.72 | 1.68 | 0.03 | 0.04 | 0.26 | 0.35 | 0.71 | 1.50 | 0.03 | 0.04 |
| | 288 | 0.04 | 9.05 | 0.05 | 0.05 | 0.06 | 0.06 | 7.71 | 240.48 | 132.38 | 6.98 | 6.82 | 8.37 | 0.06 | 0.04 | 0.65 | 1.59 | 0.03 | 0.04 | 0.26 | 0.34 | 0.66 | 1.36 | 0.03 | 0.04 |
| | 336 | 0.04 | 9.43 | 0.05 | 0.05 | 0.06 | 0.06 | 8.02 | 242.63 | 132.96 | 6.86 | 6.53 | 7.80 | 0.07 | 0.04 | 0.64 | 1.57 | 0.03 | 0.04 | 0.25 | 0.34 | 0.64 | 1.34 | 0.03 | 0.04 |
| | 672 | 0.04 | 8.86 | 0.04 | 0.05 | 0.06 | 0.06 | 7.67 | 241.13 | 130.05 | 6.71 | 6.47 | 6.32 | 0.06 | 0.04 | 0.53 | 1.46 | 0.03 | 0.04 | 0.23 | 0.31 | 0.53 | 1.02 | 0.03 | 0.04 |
| | 720 | 0.04 | 8.79 | 0.04 | 0.05 | 0.06 | 0.06 | 7.48 | 236.69 | 127.49 | 6.82 | 6.40 | 6.25 | 0.06 | 0.04 | 0.52 | 1.35 | 0.03 | 0.04 | 0.23 | 0.31 | 0.52 | 0.97 | 0.03 | 0.04 |
| Weather [5] | 24 | 0.03 | 0.03 | 0.03 | 0.03 | 0.03 | 0.03 | 0.03 | 0.03 | 0.03 | 0.03 | 0.03 | 0.03 | 0.03 | 0.03 | 0.03 | 0.03 | 0.03 | 0.03 | 0.03 | 0.03 | 0.03 | 0.03 | 0.03 | 0.03 |
| | 48 | 0.03 | 0.03 | 0.03 | 0.03 | 0.03 | 0.03 | 0.03 | 0.03 | 0.03 | 0.03 | 0.03 | 0.03 | 0.03 | 0.03 | 0.03 | 0.03 | 0.03 | 0.03 | 0.03 | 0.03 | 0.03 | 0.03 | 0.03 | 0.03 |
| | 96 | 0.03 | 0.03 | 0.03 | 0.03 | 0.03 | 0.03 | 0.03 | 0.03 | 0.03 | 0.03 | 0.03 | 0.03 | 0.03 | 0.03 | 0.03 | 0.03 | 0.03 | 0.03 | 0.03 | 0.03 | 0.03 | 0.03 | 0.03 | 0.03 |
| | 168 | 0.03 | 0.03 | 0.03 | 0.03 | 0.03 | 0.03 | 0.03 | 0.03 | 0.03 | 0.03 | 0.03 | 0.03 | 0.03 | 0.03 | 0.03 | 0.03 | 0.03 | 0.03 | 0.03 | 0.03 | 0.03 | 0.03 | 0.03 | 0.03 |
| | 288 | 0.03 | 0.03 | 0.03 | 0.03 | 0.03 | 0.03 | 0.03 | 0.03 | 0.03 | 0.03 | 0.03 | 0.03 | 0.03 | 0.03 | 0.03 | 0.03 | 0.03 | 0.03 | 0.03 | 0.03 | 0.03 | 0.03 | 0.03 | 0.03 |
| | 336 | 0.03 | 0.03 | 0.03 | 0.03 | 0.03 | 0.03 | 0.03 | 0.03 | 0.03 | 0.03 | 0.03 | 0.03 | 0.03 | 0.03 | 0.03 | 0.03 | 0.03 | 0.03 | 0.03 | 0.03 | 0.03 | 0.03 | 0.03 | 0.03 |
| | 672 | 0.03 | 0.03 | 0.03 | 0.03 | 0.03 | 0.03 | 0.03 | 0.03 | 0.03 | 0.03 | 0.03 | 0.03 | 0.03 | 0.03 | 0.03 | 0.03 | 0.03 | 0.03 | 0.03 | 0.03 | 0.03 | 0.03 | 0.03 | 0.03 |
| | 720 | 0.03 | 0.03 | 0.03 | 0.03 | 0.03 | 0.03 | 0.03 | 0.03 | 0.03 | 0.03 | 0.03 | 0.03 | 0.03 | 0.03 | 0.03 | 0.03 | 0.03 | 0.03 | 0.03 | 0.03 | 0.03 | 0.03 | 0.03 | 0.03 |

Table 9: Detailed results of the univariate time series classification task across experiments related to adaptive search space. Each row corresponds to a specific dataset, each column represents an experiment ID, and each cell contains the classification accuracy for the respective dataset and experiment.

| ExpID / Dataset | b1 | b2 | b3 | r1-1 | r2-1 | r3-1 | ar1 | ar2 | ar3 |
|---|---|---|---|---|---|---|---|---|---|
| ACSF1 | 0.61 | 0.62 | 0.54 | 0.61 | 0.62 | 0.59 | 0.66 | 0.62 | 0.58 |
| Adiac | 0.70 | 0.34 | 0.76 | 0.64 | 0.45 | 0.74 | 0.64 | 0.36 | 0.66 |
| AllGestureWiimoteX | 0.67 | 0.10 | 0.37 | 0.49 | 0.10 | 0.40 | 0.47 | 0.10 | 0.45 |
| AllGestureWiimoteY | 0.70 | 0.10 | 0.38 | 0.56 | 0.10 | 0.39 | 0.55 | 0.10 | 0.45 |
| AllGestureWiimoteZ | 0.59 | 0.10 | 0.31 | 0.49 | 0.10 | 0.37 | 0.50 | 0.10 | 0.44 |
| ArrowHead | 0.91 | 0.63 | 0.89 | 0.89 | 0.75 | 0.85 | 0.87 | 0.74 | 0.85 |
| BME | 0.92 | 0.50 | 0.96 | 1.00 | 0.84 | 0.98 | 0.94 | 0.57 | 0.97 |
| Beef | 0.47 | 0.47 | 0.53 | 0.67 | 0.40 | 0.67 | 0.43 | 0.40 | 0.73 |
| BeetleFly | 0.50 | 0.55 | 0.85 | 0.75 | 0.65 | 0.95 | 0.95 | 0.50 | 0.70 |
| BirdChicken | 0.75 | 0.55 | 0.60 | 0.70 | 0.65 | 0.60 | 0.50 | 0.50 | 0.70 |
| CBF | 0.98 | 0.91 | 1.00 | 1.00 | 1.00 | 1.00 | 0.99 | 0.99 | 1.00 |
| Car | 0.63 | 0.45 | 0.58 | 0.60 | 0.50 | 0.62 | 0.63 | 0.45 | 0.60 |
| Chinatown | 0.96 | 0.96 | 0.97 | 0.97 | 0.96 | 0.97 | 0.91 | 0.96 | 0.96 |
| ChlorineConcentration | 0.99 | 0.87 | 1.00 | 0.97 | 0.68 | 1.00 | 0.98 | 0.76 | 1.00 |
| CinCECGTorso | 0.99 | 0.90 | 1.00 | 0.99 | 0.81 | 0.99 | 0.99 | 0.93 | 0.99 |
| Coffee | 0.86 | 0.43 | 0.96 | 0.96 | 0.64 | 1.00 | 0.86 | 0.79 | 0.96 |
| Computers | 0.62 | 0.52 | 0.56 | 0.60 | 0.62 | 0.56 | 0.62 | 0.58 | 0.55 |
| CricketX | 0.68 | 0.41 | 0.62 | 0.61 | 0.33 | 0.62 | 0.65 | 0.33 | 0.63 |
| CricketY | 0.68 | 0.48 | 0.56 | 0.61 | 0.44 | 0.56 | 0.58 | 0.44 | 0.48 |
| CricketZ | 0.73 | 0.33 | 0.64 | 0.67 | 0.39 | 0.62 | 0.60 | 0.32 | 0.59 |
| Crop | 0.74 | 0.73 | 0.77 | 0.75 | 0.67 | 0.77 | 0.72 | 0.68 | 0.75 |
| DiatomSizeReduction | 0.96 | 0.93 | 0.97 | 0.97 | 0.95 | 1.00 | 0.88 | 0.95 | 0.97 |
| DistalPhalanxOutlineAgeGroup | 0.77 | 0.67 | 0.78 | 0.80 | 0.81 | 0.81 | 0.71 | 0.77 | 0.81 |
| DistalPhalanxOutlineCorrect | 0.83 | 0.75 | 0.79 | 0.79 | 0.78 | 0.80 | 0.81 | 0.81 | 0.83 |
| DistalPhalanxTW | 0.73 | 0.71 | 0.74 | 0.76 | 0.72 | 0.73 | 0.74 | 0.74 | 0.72 |
| DodgerLoopDay | 0.43 | 0.15 | 0.47 | 0.57 | 0.15 | 0.53 | 0.58 | 0.15 | 0.62 |

---

[5] The results look the same because of rounding to two decimals.

Table 9 – *Continued from previous page*

| Exp ID Dataset | b1 | b2 | b3 | r1-1 | r2-1 | r3-1 | ar1 | ar2 | ar3 |
|---|---|---|---|---|---|---|---|---|---|
| DodgerLoopGame | 0.72 | 0.52 | 0.89 | 0.90 | 0.52 | 0.84 | 0.90 | 0.52 | 0.82 |
| DodgerLoopWeekend | 0.92 | 0.71 | 1.00 | 0.99 | 0.71 | 0.97 | 0.97 | 0.71 | 0.99 |
| ECG200 | 0.83 | 0.82 | 0.91 | 0.86 | 0.77 | 0.89 | 0.83 | 0.82 | 0.91 |
| ECG5000 | 0.94 | 0.93 | 0.96 | 0.95 | 0.94 | 0.96 | 0.94 | 0.90 | 0.95 |
| ECGFiveDays | 0.98 | 0.90 | 0.99 | 0.98 | 0.90 | 1.00 | 0.96 | 0.86 | 0.96 |
| EOGHorizontalSignal | 0.54 | 0.61 | 0.60 | 0.65 | 0.49 | 0.62 | 0.58 | 0.46 | 0.59 |
| EOGVerticalSignal | 0.42 | 0.21 | 0.57 | 0.55 | 0.35 | 0.56 | 0.40 | 0.38 | 0.48 |
| Earthquakes | 0.80 | 0.80 | 0.80 | 0.81 | 0.80 | 0.80 | 0.80 | 0.80 | 0.80 |
| ElectricDevices | 0.80 | 0.66 | 0.73 | 0.74 | 0.76 | 0.73 | 0.77 | 0.72 | 0.76 |
| EthanolLevel | 0.38 | 0.31 | 0.72 | 0.43 | 0.31 | 0.69 | 0.39 | 0.30 | 0.61 |
| FaceAll | 0.91 | 0.72 | 0.94 | 0.94 | 0.67 | 0.95 | 0.94 | 0.81 | 0.92 |
| FaceFour | 0.88 | 0.70 | 0.89 | 0.91 | 0.73 | 0.89 | 0.91 | 0.68 | 0.89 |
| FacesUCR | 0.92 | 0.72 | 0.94 | 0.95 | 0.74 | 0.94 | 0.93 | 0.74 | 0.93 |
| FiftyWords | 0.51 | 0.50 | 0.71 | 0.70 | 0.49 | 0.70 | 0.58 | 0.58 | 0.68 |
| Fish | 0.81 | 0.59 | 0.87 | 0.86 | 0.39 | 0.86 | 0.87 | 0.58 | 0.87 |
| FordA | 0.89 | 0.65 | 0.81 | 0.85 | 0.84 | 0.75 | 0.64 | 0.73 | 0.78 |
| FordB | 0.88 | 0.62 | 0.78 | 0.79 | 0.73 | 0.70 | 0.75 | 0.68 | 0.70 |
| FreezerRegularTrain | 0.94 | 0.85 | 0.99 | 0.98 | 0.97 | 0.99 | 0.96 | 0.96 | 0.99 |
| FreezerSmallTrain | 0.95 | 0.93 | 0.99 | 0.99 | 0.98 | 1.00 | 0.96 | 0.95 | 0.99 |
| Fungi | 0.94 | 0.40 | 0.96 | 0.97 | 0.72 | 0.99 | 0.87 | 0.88 | 0.94 |
| GestureMidAirD1 | 0.65 | 0.04 | 0.58 | 0.57 | 0.04 | 0.56 | 0.54 | 0.04 | 0.50 |
| GestureMidAirD2 | 0.44 | 0.04 | 0.53 | 0.45 | 0.04 | 0.45 | 0.43 | 0.04 | 0.59 |
| GestureMidAirD3 | 0.22 | 0.04 | 0.37 | 0.27 | 0.04 | 0.34 | 0.27 | 0.04 | 0.26 |
| GesturePebbleZ1 | 0.87 | 0.17 | 0.84 | 0.93 | 0.18 | 0.81 | 0.86 | 0.18 | 0.82 |
| GesturePebbleZ2 | 0.93 | 0.17 | 0.93 | 0.94 | 0.18 | 0.93 | 0.92 | 0.18 | 0.84 |
| GunPoint | 0.97 | 0.70 | 0.95 | 0.96 | 0.89 | 0.95 | 0.93 | 0.74 | 0.97 |
| GunPointAgeSpan | 0.93 | 0.93 | 0.99 | 0.98 | 0.86 | 0.92 | 0.96 | 0.89 | 0.98 |
| GunPointMaleVersusFemale | 1.00 | 0.97 | 1.00 | 1.00 | 0.96 | 0.99 | 1.00 | 0.98 | 0.99 |
| GunPointOldVersusYoung | 1.00 | 1.00 | 1.00 | 1.00 | 1.00 | 1.00 | 1.00 | 0.98 | 1.00 |
| Ham | 0.64 | 0.51 | 0.82 | 0.81 | 0.56 | 0.81 | 0.79 | 0.56 | 0.81 |
| HandOutlines | 0.89 | 0.82 | 0.88 | 0.91 | 0.73 | 0.89 | 0.91 | 0.70 | 0.90 |
| Haptics | 0.41 | 0.34 | 0.43 | 0.41 | 0.39 | 0.45 | 0.46 | 0.34 | 0.47 |
| Herring | 0.61 | 0.59 | 0.61 | 0.61 | 0.61 | 0.61 | 0.61 | 0.61 | 0.61 |
| HouseTwenty | 0.96 | 0.89 | 0.72 | 0.81 | 0.62 | 0.71 | 0.84 | 0.80 | 0.72 |
| InlineSkate | 0.41 | 0.29 | 0.43 | 0.45 | 0.23 | 0.46 | 0.48 | 0.27 | 0.50 |
| InsectEPGRegularTrain | 1.00 | 1.00 | 1.00 | 1.00 | 1.00 | 1.00 | 1.00 | 1.00 | 1.00 |
| InsectEPGSmallTrain | 1.00 | 1.00 | 1.00 | 1.00 | 1.00 | 1.00 | 1.00 | 1.00 | 1.00 |
| InsectWingbeatSound | 0.60 | 0.46 | 0.67 | 0.68 | 0.58 | 0.67 | 0.66 | 0.60 | 0.67 |
| ItalyPowerDemand | 0.96 | 0.96 | 0.96 | 0.96 | 0.93 | 0.95 | 0.50 | 0.94 | 0.96 |
| LargeKitchenAppliances | 0.72 | 0.54 | 0.50 | 0.48 | 0.57 | 0.50 | 0.62 | 0.60 | 0.47 |
| Lightning2 | 0.67 | 0.67 | 0.62 | 0.69 | 0.64 | 0.72 | 0.74 | 0.57 | 0.67 |
| Lightning7 | 0.69 | 0.50 | 0.69 | 0.65 | 0.44 | 0.65 | 0.62 | 0.50 | 0.62 |
| Mallat | 1.00 | 0.94 | 0.98 | 0.99 | 1.00 | 0.98 | 0.98 | 0.96 | 0.99 |
| Meat | 0.98 | 0.98 | 0.98 | 0.98 | 0.93 | 0.97 | 0.85 | 0.95 | 0.97 |
| MedicalImages | 0.68 | 0.65 | 0.75 | 0.76 | 0.70 | 0.74 | 0.62 | 0.65 | 0.75 |
| MelbournePedestrian | 0.96 | 0.10 | 0.94 | 0.96 | 0.10 | 0.96 | 0.95 | 0.10 | 0.94 |
| MiddlePhalanxOutlineAgeGroup | 0.74 | 0.70 | 0.73 | 0.75 | 0.76 | 0.73 | 0.74 | 0.74 | 0.74 |
| MiddlePhalanxOutlineCorrect | 0.79 | 0.73 | 0.80 | 0.79 | 0.73 | 0.83 | 0.80 | 0.71 | 0.78 |
| MiddlePhalanxTW | 0.61 | 0.50 | 0.61 | 0.61 | 0.58 | 0.61 | 0.57 | 0.61 | 0.62 |
| MixedShapesRegularTrain | 0.93 | 0.87 | 0.93 | 0.95 | 0.86 | 0.94 | 0.91 | 0.86 | 0.93 |
| MixedShapesSmallTrain | 0.92 | 0.88 | 0.95 | 0.95 | 0.84 | 0.94 | 0.93 | 0.87 | 0.95 |
| MoteStrain | 0.96 | 0.90 | 0.94 | 0.95 | 0.91 | 0.95 | 0.96 | 0.93 | 0.96 |
| NonInvasiveFetalECGThorax1 | 0.77 | 0.60 | 0.90 | 0.83 | 0.71 | 0.88 | 0.82 | 0.76 | 0.89 |
| NonInvasiveFetalECGThorax2 | 0.78 | 0.61 | 0.89 | 0.86 | 0.62 | 0.91 | 0.83 | 0.56 | 0.90 |
| OSULeaf | 0.75 | 0.42 | 0.65 | 0.64 | 0.45 | 0.66 | 0.60 | 0.53 | 0.67 |
| OliveOil | 0.63 | 0.40 | 0.87 | 0.93 | 0.67 | 0.83 | 0.73 | 0.90 | 0.83 |
| PLAID | 0.48 | 0.16 | 0.42 | 0.42 | 0.16 | 0.42 | 0.26 | 0.16 | 0.41 |
| PhalangesOutlinesCorrect | 0.81 | 0.71 | 0.80 | 0.81 | 0.80 | 0.83 | 0.79 | 0.78 | 0.80 |
| Phoneme | 0.32 | 0.15 | 0.20 | 0.19 | 0.22 | 0.14 | 0.18 | 0.25 | 0.15 |

Table 9 – *Continued from previous page*

| Exp ID Dataset | b1 | b2 | b3 | r1-1 | r2-1 | r3-1 | ar1 | ar2 | ar3 |
|---|---|---|---|---|---|---|---|---|---|
| PickupGestureWiimoteZ | 0.62 | 0.10 | 0.62 | 0.72 | 0.10 | 0.68 | 0.70 | 0.10 | 0.56 |
| PigAirwayPressure | 0.40 | 0.19 | 0.17 | 0.17 | 0.12 | 0.18 | 0.15 | 0.09 | 0.12 |
| PigArtPressure | 0.81 | 0.32 | 0.25 | 0.42 | 0.17 | 0.22 | 0.50 | 0.28 | 0.22 |
| PigCVP | 0.36 | 0.19 | 0.18 | 0.17 | 0.17 | 0.02 | 0.38 | 0.19 | 0.17 |
| Plane | 0.99 | 0.98 | 0.99 | 0.99 | 0.88 | 0.99 | 0.99 | 0.90 | 1.00 |
| PowerCons | 0.92 | 0.84 | 0.97 | 0.96 | 0.92 | 0.98 | 0.94 | 0.92 | 0.99 |
| ProximalPhalanxOutlineAgeGroup | 0.81 | 0.79 | 0.81 | 0.82 | 0.81 | 0.82 | 0.81 | 0.79 | 0.83 |
| ProximalPhalanxOutlineCorrect | 0.84 | 0.75 | 0.84 | 0.84 | 0.76 | 0.84 | 0.80 | 0.76 | 0.80 |
| ProximalPhalanxTW | 0.78 | 0.76 | 0.83 | 0.81 | 0.80 | 0.83 | 0.77 | 0.77 | 0.82 |
| RefrigerationDevices | 0.62 | 0.42 | 0.42 | 0.50 | 0.49 | 0.42 | 0.63 | 0.46 | 0.38 |
| Rock | 0.71 | 0.54 | 0.71 | 0.69 | 0.71 | 0.69 | 0.71 | 0.57 | 0.74 |
| ScreenType | 0.47 | 0.34 | 0.46 | 0.45 | 0.42 | 0.45 | 0.42 | 0.39 | 0.42 |
| SemgHandGenderCh2 | 0.62 | 0.64 | 0.91 | 0.94 | 0.85 | 0.94 | 0.93 | 0.85 | 0.93 |
| SemgHandMovementCh2 | 0.43 | 0.33 | 0.67 | 0.85 | 0.73 | 0.74 | 0.87 | 0.74 | 0.72 |
| SemgHandSubjectCh2 | 0.44 | 0.29 | 0.76 | 0.92 | 0.67 | 0.84 | 0.92 | 0.79 | 0.88 |
| ShakeGestureWiimoteZ | 0.76 | 0.10 | 0.64 | 0.70 | 0.10 | 0.66 | 0.80 | 0.10 | 0.66 |
| ShapeletSim | 0.99 | 0.54 | 0.45 | 0.50 | 0.50 | 0.57 | 0.50 | 0.44 | 0.50 |
| ShapesAll | 0.73 | 0.59 | 0.76 | 0.77 | 0.59 | 0.76 | 0.71 | 0.57 | 0.79 |
| SmallKitchenAppliances | 0.60 | 0.54 | 0.57 | 0.33 | 0.66 | 0.58 | 0.54 | 0.55 | 0.57 |
| SmoothSubspace | 0.95 | 0.93 | 0.92 | 0.94 | 0.97 | 0.92 | 0.93 | 0.94 | 0.92 |
| SonyAIBORobotSurface1 | 0.97 | 0.95 | 0.99 | 0.99 | 0.95 | 0.99 | 0.99 | 0.99 | 0.99 |
| SonyAIBORobotSurface2 | 0.92 | 0.93 | 0.97 | 0.97 | 0.95 | 0.98 | 0.97 | 0.90 | 0.98 |
| StarLightCurves | 0.97 | 0.92 | 0.97 | 0.97 | 0.93 | 0.97 | 0.96 | 0.95 | 0.97 |
| Strawberry | 0.98 | 0.83 | 0.93 | 0.96 | 0.90 | 0.98 | 0.88 | 0.92 | 0.97 |
| SwedishLeaf | 0.88 | 0.74 | 0.89 | 0.88 | 0.79 | 0.91 | 0.72 | 0.39 | 0.88 |
| Symbols | 0.97 | 0.92 | 0.97 | 0.97 | 0.90 | 0.96 | 0.97 | 0.90 | 0.96 |
| SyntheticControl | 0.98 | 0.80 | 0.97 | 0.98 | 0.96 | 0.97 | 0.99 | 0.98 | 0.96 |
| ToeSegmentation1 | 0.94 | 0.52 | 0.70 | 0.68 | 0.56 | 0.67 | 0.90 | 0.63 | 0.65 |
| ToeSegmentation2 | 0.92 | 0.75 | 0.75 | 0.80 | 0.78 | 0.78 | 0.87 | 0.73 | 0.77 |
| Trace | 0.98 | 0.80 | 0.83 | 0.91 | 0.74 | 0.91 | 0.99 | 0.74 | 0.86 |
| TwoLeadECG | 0.99 | 0.92 | 0.99 | 0.97 | 0.94 | 0.99 | 0.94 | 0.93 | 0.96 |
| TwoPatterns | 0.90 | 0.70 | 0.98 | 1.00 | 1.00 | 0.96 | 1.00 | 1.00 | 0.98 |
| UMD | 0.78 | 0.59 | 0.78 | 0.92 | 0.73 | 0.93 | 0.90 | 0.76 | 0.86 |
| UWaveGestureLibraryAll | 0.95 | 0.82 | 0.97 | 0.97 | 0.94 | 0.97 | 0.96 | 0.90 | 0.97 |
| UWaveGestureLibraryX | 0.79 | 0.72 | 0.81 | 0.82 | 0.77 | 0.80 | 0.82 | 0.76 | 0.80 |
| UWaveGestureLibraryY | 0.70 | 0.57 | 0.76 | 0.76 | 0.69 | 0.74 | 0.75 | 0.67 | 0.74 |
| UWaveGestureLibraryZ | 0.70 | 0.65 | 0.75 | 0.75 | 0.69 | 0.76 | 0.72 | 0.65 | 0.76 |
| Wafer | 0.99 | 0.99 | 1.00 | 1.00 | 0.99 | 1.00 | 1.00 | 0.99 | 1.00 |
| Wine | 0.88 | 0.82 | 0.93 | 0.86 | 0.75 | 0.89 | 0.88 | 0.79 | 0.91 |
| WordSynonyms | 0.54 | 0.41 | 0.65 | 0.71 | 0.49 | 0.68 | 0.55 | 0.57 | 0.66 |
| Worms | 0.70 | 0.53 | 0.54 | 0.63 | 0.50 | 0.50 | 0.64 | 0.57 | 0.54 |
| WormsTwoClass | 0.74 | 0.57 | 0.50 | 0.58 | 0.53 | 0.57 | 0.71 | 0.57 | 0.57 |
| Yoga | 0.87 | 0.86 | 0.95 | 0.95 | 0.88 | 0.94 | 0.93 | 0.77 | 0.95 |

Table 10: Detailed results of the multivariate time series classification task across experiments related to adaptive search space. Each row corresponds to a specific dataset, each column represents an experiment ID, and each cell contains the classification accuracy for the respective dataset and experiment.

| ExpID Dataset | b1 | b2 | b3 | r1-1 | r2-1 | r3-1 | ar1 | ar2 | ar3 |
|---|---|---|---|---|---|---|---|---|---|
| ArticularyWordRecognition | 0.95 | 0.92 | 0.97 | 0.97 | 0.89 | 0.96 | 0.87 | 0.93 | 0.97 |
| AtrialFibrillation | 0.20 | 0.33 | 0.33 | 0.33 | 0.33 | 0.27 | 0.33 | 0.33 | 0.33 |
| BasicMotions | 1.00 | 0.95 | 0.95 | 0.95 | 1.00 | 0.93 | 0.90 | 0.93 | 0.95 |
| CharacterTrajectories | 0.98 | 0.07 | 0.93 | 0.97 | 0.07 | 0.88 | 0.69 | 0.07 | 0.68 |
| Cricket | 1.00 | 0.94 | 0.94 | 0.80 | 0.86 | 0.96 | 0.87 | 0.48 | 0.92 |

Table 10 – *Continued from previous page*

| dataset | b1 | b2 | b3 | r1-1 | r2-1 | r3-1 | ar1 | ar2 | ar3 |
|---|---|---|---|---|---|---|---|---|---|
| DuckDuckGeese | 0.50 | 0.34 | 0.32 | 0.28 | 0.30 | 0.40 | 0.42 | 0.28 | 0.32 |
| ERing | 0.94 | 0.89 | 0.95 | 0.95 | 0.85 | 0.95 | 0.85 | 0.88 | 0.93 |
| EigenWorms | 0.76 | 0.45 | 0.56 | 0.65 | 0.55 | 0.56 | 0.68 | 0.41 | 0.55 |
| Epilepsy | 0.96 | 0.83 | 0.80 | 0.89 | 0.79 | 0.86 | 0.92 | 0.85 | 0.89 |
| EthanolConcentration | 0.32 | 0.39 | 0.34 | 0.32 | 0.27 | 0.36 | 0.29 | 0.21 | 0.33 |
| FaceDetection | 0.50 | 0.50 | 0.56 | 0.58 | 0.52 | 0.61 | 0.55 | 0.51 | 0.57 |
| FingerMovements | 0.58 | 0.49 | 0.44 | 0.53 | 0.50 | 0.50 | 0.51 | 0.50 | 0.53 |
| HandMovementDirection | 0.30 | 0.30 | 0.34 | 0.30 | 0.30 | 0.43 | 0.30 | 0.30 | 0.38 |
| Handwriting | 0.64 | 0.12 | 0.45 | 0.46 | 0.18 | 0.55 | 0.21 | 0.12 | 0.26 |
| Heartbeat | 0.74 | 0.70 | 0.68 | 0.68 | 0.70 | 0.72 | 0.73 | 0.72 | 0.72 |
| InsectWingbeat | 0.20 | 0.10 | 0.17 | 0.17 | 0.10 | 0.23 | 0.18 | 0.10 | 0.17 |
| JapaneseVowels | 0.95 | 0.18 | 0.91 | 0.92 | 0.18 | 0.88 | 0.90 | 0.18 | 0.91 |
| LSST | 0.49 | 0.38 | 0.42 | 0.42 | 0.41 | 0.43 | 0.44 | 0.41 | 0.42 |
| Libras | 0.88 | 0.75 | 0.84 | 0.76 | 0.61 | 0.76 | 0.74 | 0.60 | 0.77 |
| MotorImagery | 0.52 | 0.46 | 0.50 | 0.51 | 0.56 | 0.50 | 0.56 | 0.51 | 0.50 |
| NATOPS | 0.85 | 0.80 | 0.78 | 0.80 | 0.82 | 0.89 | 0.81 | 0.79 | 0.79 |
| PEMS-SF | 0.76 | 0.74 | 0.83 | 0.70 | 0.62 | 0.76 | 0.80 | 0.72 | 0.77 |
| PenDigits | 0.98 | 0.97 | 0.99 | 0.99 | 0.99 | 0.99 | 0.99 | 0.98 | 0.99 |
| PhonemeSpectra | 0.24 | 0.14 | 0.12 | 0.13 | 0.10 | 0.10 | 0.16 | 0.10 | 0.12 |
| RacketSports | 0.86 | 0.81 | 0.82 | 0.81 | 0.84 | 0.75 | 0.85 | 0.83 | 0.84 |
| SelfRegulationSCP1 | 0.77 | 0.84 | 0.84 | 0.83 | 0.75 | 0.82 | 0.73 | 0.74 | 0.81 |
| SelfRegulationSCP2 | 0.54 | 0.51 | 0.53 | 0.49 | 0.53 | 0.52 | 0.48 | 0.52 | 0.55 |
| SpokenArabicDigits | 0.98 | 0.10 | 0.97 | 0.99 | 0.10 | 0.97 | 0.85 | 0.10 | 0.85 |
| StandWalkJump | 0.29 | 0.29 | 0.29 | 0.29 | 0.29 | 0.36 | 0.29 | 0.43 | 0.29 |
| UWaveGestureLibrary | 0.94 | 0.78 | 0.90 | 0.93 | 0.83 | 0.90 | 0.94 | 0.87 | 0.91 |

