# OpenReview forum: "AutoTSAugment: Model-Agnostic Automated Data Augmentation for Unsupervised Contrastive-based Time Series Representation Learning"
_ICLR.cc/2026/Conference — ICLR 2026 Conference Withdrawn Submission_

### Official Review · Reviewer_EXGo · 2025-10-17

**Soundness:** 2
**Presentation:** 2
**Contribution:** 1
**Rating:** 2
**Confidence:** 4

**Summary:**

This paper introduces AutoTSAugment, a modular and model-agnostic framework for automatically discovering data augmentation strategies for unsupervised, contrastive-based time series representation learning. The authors identify that manually designing augmentation policies is a major bottleneck, requiring significant effort and domain expertise. The proposed AutoML framework consists of a diverse search space of common time series transformations and a search strategy guided by a novel, distance-based objective function designed for unsupervised settings.  The primary strengths lie in its extensive and rigorous empirical evaluation, which spans multiple models, tasks, and a very large number of datasets, lending strong support to its conclusions. However, the technical novelty of the proposed search objective appears somewhat limited.

**Strengths:**

1. The paper's claims are supported by an exceptionally thorough set of experiments, which strongly validates the proposed framework's effectiveness and generalizability.
- The evaluation spans four distinct downstream tasks: univariate and multivariate time series classification, as well as univariate and multivariate forecasting, demonstrating the versatility of the learned representations.
- The framework is tested on 164 widely used datasets (128 from UCR, 30 from UEA, and several standard forecasting benchmarks), which provides robust evidence for the general applicability of the findings.

2. The work addresses a well-recognized and significant pain point in applying deep learning to time series, the manual selection of data augmentations.

3. The authors provide substantial detail about their framework and experimental setup, which is commendable and facilitates the verification and extension of their work.

**Weaknesses:**

1. Limited contribution. While the framework as a whole is novel, the core technical contribution of the unsupervised search objective could be better motivated and clarified.
- The paper cites [Baratchi et al., 2024] for the general form of the search objective, which may lead readers to question how much of the formulation is a novel contribution of this specific work versus an adaptation of an existing AutoML pattern.
- The score function, Eq. 2 $\mathcal{P}(f;\mathcal{X})$,  is hard to understand.   The scoring function $\Phi_{\tau}(d) = s \cdot |d - \tau|$ is a simple absolute distance from a pre-defined threshold $\tau$. Its conceptual novelty is limited.

2. Unclear writing.  For example, in the contribution, the authors claim this module is designed for time series methods. Why time series is unclear?  I am not sure it is because of the search space. The second contribution, "beyond that of standard AutoML", lacks evidence.

3. Unreasonable experiment setting. This paper's main contribution should be the augmentation framework. So the baseline should contain the basic AutoML method and explain why this method is better than the normal AutoML. Since the title claims "model-agnostic", there should be other architectures for time series, such as transformer, linear, and so on. As far as I know, TS2Vec, CoST, and AutoTCL are based on 1d-CNN.

**Questions:**

1. Why was a distance-based objective in the input space (Eq. 2) chosen? Were other unsupervised objectives, such as metrics in the embedding space (e.g., alignment and uniformity), considered?
2.  How sensitive is the framework's performance to the choice of the distance threshold $\tau$?
3. The search space includes six specific augmentation methods (Sec. 3.1). What was the rationale for selecting these six? Were other types, such as frequency-domain augmentations, considered and excluded?
4. What exact rank r (low-rank dimension) was used for the controller in the main experiments (Sec. 3.2; Eq. 5)?

---

### Official Review · Reviewer_A5NT · 2025-10-19

**Soundness:** 3
**Presentation:** 1
**Contribution:** 2
**Rating:** 4
**Confidence:** 3

**Summary:**

This paper presents a framework to automate the selection of data augmentation strategies for unsupervised, contrastive-based time series representation learning. The proposed method is designed to be modular and model-agnostic, decoupling the augmentation process from the model architecture. The authors claim two main contributions: (1) the framework itself, which includes a defined search space and search strategies, and (2) a novel search objective for evaluating augmentation quality in an unsupervised setting. The results show that the framework can achieve competitive performance compared to manually-tuned augmentation strategies in existing baseline models.

**Strengths:**

1.  **Empirical Evaluation:** The authors evaluate the proposed framework by integrating it into three different baseline models and testing it on 164 datasets across both classification and forecasting tasks. This comprehensive evaluation provides strong evidence for the general applicability and robustness of the framework.

2.  **Model-Agnostic and Modular Approach:** The proposed framework decouples augmentation from the model. It addresses a limitation in existing monolithic models where augmentation strategies are integrated.

**Weaknesses:**

1.  **Incremental Novelty:** The work feels more like an incremental application of existing ideas rather than a novel contribution.
    * The **search objective**, claimed as a primary novelty, appears to be an application of the standard AutoML problem formalization [1] to this specific problem. The performance metric itself is a straightforward implementation of the core heuristic in contrastive learning, that positive pairs should be close but not identical, rather than a novel mechanism.
    * The **framework**, while useful, is an assembly of existing concepts (AutoML, model-agnostic design, unsupervised criteria) applied to the time series domain. It is a valuable engineering contribution but does not introduce a new algorithmic mechanism.

2.  **Limited Search Space:** The search space is constructed from only six common and widely-used augmentation methods. While the goal is automation, the framework's potential is inherently limited by this small set of transformations.

3.  **Overstated Claims and Presentation:** The paper's presentation feels somewhat overstated. The framing of the framework's novelty and efficiency can be misleading, for example, the reported 63.25 percent speed up is compared within the proposed framework, not comparing with optimized baselines.

[1] Mitra Baratchi, Can Wang, Steffen Limmer, Jan N. van Rijn, Holger Hoos, Thomas Bäck, and
Markus Olhofer. Automated machine learning: past, present and future. Artificial Intelligence
Review, 57(122), 2024.

**Questions:**

1.  The scenario described in Figure 1, which highlights the framework's efficiency, seems practical only when training multiple different models on the exact same, pre-generated set of augmented data. In most common use cases, data augmentation is done "on-the-fly" while training a single model. Could you clarify the practical importance and applicability of this "compute once, reuse often" paradigm in typical research or production workflows?

2.  For the adaptive search space experiment, you used `RandSampling(n=1)`. This seems to reduce the "search" problem to simply picking a single, pre-recommended option. Why was this simplistic setting chosen instead of using the recommendations to guide a more complex search (e.g., by creating a weighted sampling distribution for Random Search or a more informed prior for Bayesian Optimization)?

3.  The paper claims a significant speed-up of "up to 63.25% faster," but this metric compares the different search strategies *within* the AutoTSAugment framework. It doesn't compare the framework's end-to-end runtime against the original, highly optimized monolithic baselines. What is the practical implication of this internal comparison, and how does the total runtime (including search) of your framework with RandSampling compare to a baseline like TS2Vec using its native on-the-fly augmentation?

**Typo:** Line 238: "the search objective of is " $\rightarrow$ "the search objective is".

---

### Official Review · Reviewer_nGRv · 2025-10-30

**Soundness:** 3
**Presentation:** 2
**Contribution:** 2
**Rating:** 2
**Confidence:** 4

**Summary:**

This paper introduces AutoTSAugment, a framework for automatically selecting augmentations in unsupervised contrastive time series learning. The idea is to remove manual tuning through a distance-based search objective. Experiments on 164 datasets show strong and sometimes better performance than manually chosen strategies.

**Strengths:**

- The paper addresses an important and practical problem in time-series representation learning: how to automatically select effective data augmentation strategies. This motivation is timely and valuable, as existing contrastive learning approaches still rely on manual augmentation design.

- The framework design is clear. AutoTSAugment aims to decouple the augmentation generation from model training, enabling augmented views to be computed once and reused across models.

- The empirical evaluation is extensive, covering 164 datasets for classification and forecasting tasks.

**Weaknesses:**

- The abstract is overly long and lacks focus, making it difficult for me to quickly grasp the paper’s main contributions. It spends too much space on describing background and formulation, rather than emphasizing the novel aspects and key takeaways.

- The proposed distance-based search objective is not convincing. The idea of making augmented views “similar but not identical” is well established in contrastive learning. The implementation using a fixed distance threshold  feels heuristic and lacks theoretical grounding.
- The optimization target  𝒫(f; 𝒳)  is a proxy metric that is fully detached from real downstream performance, yet the paper assumes they are correlated. There is no analysis showing that improving this proxy actually leads to better representations. In fact, the results in Section 5 suggest that more complex optimization methods do not lead to performance gains, which weakens the fundamental motivation for the proposed objective.
- The main finding somewhat contradicts the purpose of the framework. Random sampling achieves equal or even better results compared to much more elaborate AutoML strategies, while being substantially faster. This makes the whole AutoTSAugment design feel unnecessary and reduces the contribution to an empirical confirmation of prior insights such as those from RandAugment.
- The model-agnostic claim is not convincingly supported by experiments. The paper shows compatibility with several existing contrastive baselines, but does not demonstrate that augmentations found once can transfer across different architectures, which is what Figure 1 seems to claim. Without such evidence, the main advantage of the framework remains speculative.

**Questions:**

1. Given that random sampling yields comparable or better results, what is the real value or necessity of the AutoML framework?
2. Have you analyzed the correlation between 𝒫(f; 𝒳) and downstream task performance to justify using it as a search metric?
3. How do you ensure that the manually chosen threshold τ generalizes across datasets with widely varying scales and distributions?
4. Can you show results where the same augmentation strategy, once found, works effectively across different model architectures?

---

### Official Review · Reviewer_oppW · 2025-11-02

**Soundness:** 2
**Presentation:** 2
**Contribution:** 1
**Rating:** 2
**Confidence:** 4

**Summary:**

This paper introduces AutoTSAugment, a modular and model-agnostic AutoML framework for automated time series data augmentation in unsupervised contrastive representation learning. The framework decouples augmentation from model architecture, defining a search space over common time-series augmentations (e.g., scaling, warping, jitter, permutation) and a search objective based on time-series distance metrics rather than supervised validation scores. Three search strategies are explored, i.e. random sampling, random search, and Bayesian optimization. Results show that AutoTSAugment achieves performance comparable or slightly better than manually designed augmentations, while significantly reducing the manual effort and search time.

**Strengths:**

- Testing on 164 datasets with multiple baselines (TS2Vec, CoST, AutoTCL) and tasks (classification and forecasting).
- Reproducibility is supported via an open GitHub repository, which strengthens the paper’s scientific value.

**Weaknesses:**

- The proposed approach largely integrates existing components, standard augmentations, simple distance-based scoring, and known AutoML search strategies. The conceptual contribution (unsupervised model-agnostic augmentation search) feels incremental relative to prior work such as AutoAugment [1], RandAugment [2], AutoCL [3], AutoTCL [4].
- The work provides minimal analysis on why the proposed distance-based search objective correlates with representation quality. There’s no theoretical or ablation study connecting the proposed metric with downstream performance.
- While results are consistent, the performance improvements are often statistically insignificant or marginal. The claim that the method achieves “similar or better” results than manual augmentation can be interpreted as parity rather than advancement.
- The proposed distance threshold scoring is heuristic and may not generalize well to tasks requiring semantic consistency beyond low-level similarity. The authors do not discuss potential failure cases or sensitivity to $\tau$ and distance metric choices.
- Recent time-series foundation model works (e.g., FMs like Moment [5], or models from [6]) are only mentioned briefly, with no quantitative benchmarking. Including them would provide stronger evidence of relevance.

[1] Cubuk, Ekin D., et al. "Autoaugment: Learning augmentation strategies from data." Proceedings of the IEEE/CVF conference on computer vision and pattern recognition. 2019.

[2] Cubuk, Ekin D., et al. "Randaugment: Practical automated data augmentation with a reduced search space." Proceedings of the IEEE/CVF conference on computer vision and pattern recognition workshops. 2020.

[3] Jing, Baoyu, et al. "Automated contrastive learning strategy search for time series." Proceedings of the 33rd ACM International Conference on Information and Knowledge Management. 2024.

[4] Zheng, Xu, et al. "Auto tcl: Automated time series contrastive learning with adaptive augmentations." Proc. 32nd Int. Joint Conf. Artif. Intell.(IJCAI). 2023.

[5] Goswami, Mononito, et al. "Moment: A family of open time-series foundation models." arXiv preprint arXiv:2402.03885 (2024).

[6] Liang, Yuxuan, et al. "Foundation models for time series analysis: A tutorial and survey." Proceedings of the 30th ACM SIGKDD conference on knowledge discovery and data mining. 2024.

**Questions:**

Minor issues:
- Some notations (e.g., $\phi \tau$) are not fully motivated before being used.
- Discussion of adaptive search space is interesting but limited in depth, the adaptation method (Liu et al., 2024) is borrowed wholesale with minor integration.
- Figures (e.g., CD diagrams) are informative but hard to interpret without corresponding numerical summaries in the main text.

---

### Note · Authors · 2025-11-21

**Comment:**

We decided to withdraw the paper for the following reasons:

1) Reviewer A5NT questions the novelty of our search objective by citing a **previous paper from our lab**, Baratchi et al. (2024) [3], and labeling it a “standard AutoML problem formalization,” without acknowledging the specific differences and contributions of our work. Baratchi et al. (2024) [3] do not define the specific unsupervised distance-based augmentation objective proposed in our work.

2) Several criticisms that are either addressed in the paper or beyond its scope.
Examples:
- The request to include “other AutoML methods” as baseline comparisons, without mentioning any examples of any concrete methods that satisfy the experiment settings (from reviewer EXGo). We explicitly explain in the introduction and the related work sections that, to the best of our knowledge, there is no other AutoML method that has the following properties: (i) tailored to time series, (ii) works in unsupervised settings, (iii) model-agnostic, (iv) modular, and (v) interpretable. Given this, we did not find any other AutoML method that can be directly compared with AutoTSAugment.
- The request to include “other architectures for time series, such as transformer, linear, and so on” (from reviewer EXGo). We explicitly define the purpose of AutoTSAugment as working with unsupervised contrastive-based time series representation learning methods. We selected the methods that are recognized as the SOTA in the literature to do our experiments and did additional experiments on time series foundation models in the appendix. The vast majority of the SOTA unsupervised contrastive methods for time series are based on CNN (e.g., AutoTCL, TS2Vec .. etc) [1][2], so using CNN-based models as the main set of baseline methods was not an arbitrary choice.

3) The reviewers acknowledge the empirical evaluation is extensive and practically relevant; however, they still decide on rejection by labelling the work as “incremental” or “engineering” without offering a precise explanation of what would count as sufficient conceptual novelty in this area, or how we could revise our work to meet that bar.

4) All the 4 reviews are fully or partially generated by AI. According to (source: https://iclr.pangram.com/reviews?query=autotsaugment), our paper received the following reviews: 1 x Fully AI-generated, 1 x Heavily AI-edited, 1 x Moderately AI-edited, and 1 x Lightly AI-edited, with 0 fully human-written reviews.

Given these points, we are not confident that additional author time spent on a rebuttal would lead to a careful, human-led re-evaluation of the work, and therefore decided to withdraw the paper and consider another venue.


References

[1] Qianwen Meng, Hangwei Qian, Yong Liu, Yonghui Xu, Zhiqi Shen, and Lizhen Cui. Unsupervised representation learning for time series: A review, 2023. arXiv:2308.01578v1

[2] Kexin Zhang, Qingsong Wen, Chaoli Zhang, Rongyao Cai, Ming Jin, Yong Liu, James Y. Zhang, Yuxuan Liang, Guansong Pang, Dongjin Song, and Shirui Pan. Self-supervised learning for time series analysis: Taxonomy, progress, and prospects. IEEE Transactions on Pattern Analysis and Machine Intelligence, 2024.

[3] Mitra Baratchi, Can Wang, Steffen Limmer, Jan N. van Rijn, Holger Hoos, Thomas Bäck, and Markus Olhofer. Automated machine learning: past, present, and future. Artificial Intelligence Review, 57(122), 2024.

**Withdrawal Confirmation:**

I have read and agree with the venue's withdrawal policy on behalf of myself and my co-authors.